# NOISE-ROBUST DENSITY ESTIMATION FOR TABULAR DATA ANOMALY DETECTION

## ABSTRACT

Density-based anomaly detection methods often provide accurate and interpretable predictions but their performance can be severely affected by the inherent noise of data. In this paper, we present a noise-robust density estimation (NRDE) method for tabular data anomaly detection. We aim to estimate the density of pure data with the influence of noises isolated, which is a non-trivial task since the data-generating process is completely unknown. NRDE learns a Jacobian-regularized normalizing flow to estimate the sources of data and categorizes sources into two groups, where one group generates pure data and the other generates noise. Then we can estimate the density of pure data and use it to detect anomalies caused by the sources of pure data rather than the changes caused by the sources of noise. Therefore, compared with other density-based methods, our NRDE is much more robust to noise. In addition to the new algorithm, we also provide theoretical results to support the effectiveness of NRDE. We compare NRDE with 15 baselines on 47 benchmark datasets under different settings, including vanilla anomaly detection, anomaly detection with anomaly contamination, anomaly detection on noisy data, and transductive outlier detection. The results demonstrate the effectiveness and superiority of NRDE.

## 1 INTRODUCTION

In an increasingly data-driven world, the problem of identifying unusual patterns or deviations from expected behavior—known as anomaly detection—has become paramount across diverse domains. Anomaly detection (Chandola et al., 2009; Pang et al., 2021; Ruff et al., 2021), sometimes also referred to as novelty or outlier detection (Breunig et al., 2000; Pimentel et al., 2014), involves the identification of data points, events, or observations that significantly differ from the majority of the data. These anomalies can signal critical incidents such as fraud (Ahmed et al., 2016), security breaches (Breier & Branišová, 2017), system failures (Du et al., 2017), or novel insights, making their accurate detection essential for timely intervention and decision-making.

In the past few years, a diverse range of deep learning-based anomaly detection methods have been proposed (Ruff et al., 2018b; Deecke et al., 2019; Ruff et al., 2019; Wang et al., 2021; Pang et al., 2019; Goyal et al., 2020; Qiu et al., 2021; Cai & Fan, 2022; Xu et al., 2023a; Zhang et al., 2024). For instance, DeepSVDD (Ruff et al., 2018b) assumes that representations of normal data can be enclosed within a small hypersphere and representations of anomalous data lie outside the hypersphere, where the representations are given by a neural network. ICL (Shenkar & Wolf, 2022) assumes that a subset of the feature vector is related to the rest and uses self-supervised learning to maximize the mutual information between each sample and the masked-out part. SLAD (Xu et al., 2023b) performs scale learning to embed high-level information into its ranking mechanism. Although these methods often demonstrate impressive performance in various scenarios, several of them require making assumptions on the structure or distribution of normal and anomalous data, which may not hold or are difficult to guarantee by the training process. For instance, Zhang et al. (2024) analyzed the limitations of the hypersphere assumption in high-dimensional spaces and proposed to project normal data into the region bounded by two hyperspheres. Moreover, some of these methods are proposed to solve the one-class classification (OCC) problem, which relies on the assumption that training data originate from a single class or have a single manifold structure. Consequently, these methods can be ineffective when the training data encompasses multiple clusters or lies on multiple disconnected manifolds, as mentioned in (Khayatkhoei et al., 2018).

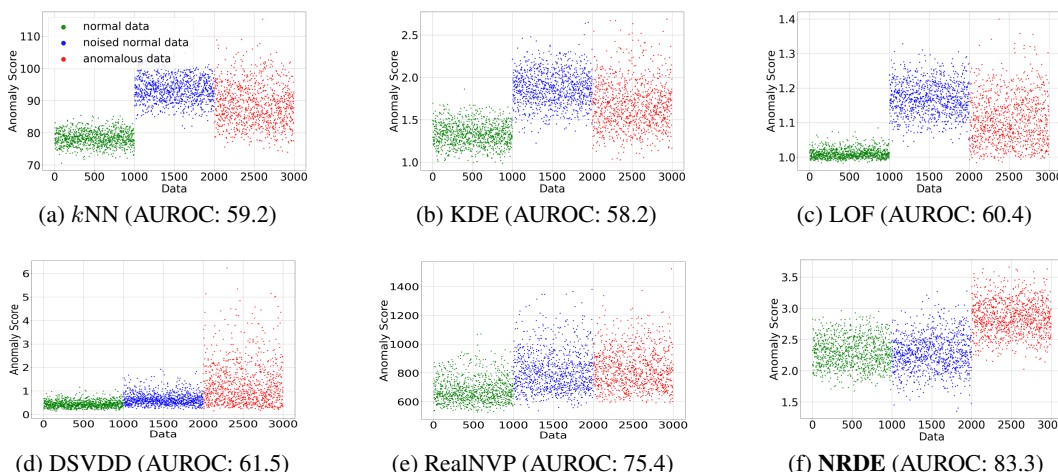

Figure 1: Detection performance on a synthetic dataset. The data were generated from a few data sources and many noise sources. Points marked in green, blue, and red represent **normal data**, **noisy normal data** (caused by noise change), and **anomalous data**, respectively. See (8) for definitions. The five compared methods detect most of the noisy normal samples as anomalies, while our NRDE is robust to the changes of noise. More details about this experiment are in Appendix F.

It should be noted that density-based methods make no assumptions about the shape or distribution of the data and are capable of modeling complex data structures. This flexibility allows them to be effective even when the training data encompasses multiple classes, and these methods use the local or global density of the data as an anomaly score. Traditional density-based methods include Kernel Density Estimation (KDE) (Parzen, 1962), Gaussian Mixture Models (GMM), etc. These methods often suffer from the curse of dimensionality and are not effective in modeling complex data. To address the problem, several deep learning based density estimation methods have been proposed. For instance, DAGMM (Zong et al., 2018) combines deep auto-encoders with GMM, utilizing the output density given by GMM in a low-dimensional space to detect anomalies. Normalizing flow (Kobyzev et al., 2020), an effective generative model, is also effective in estimating the density of complex data, and hence is useful for anomaly detection. Some flow-based image anomaly detection methods (Gudovskiy et al., 2022; Kim et al., 2023) first employ feature extractors to derive semantic representations of images and then implement normalizing flow to detect anomalies. In this work, we focus on tabular data since data of other types can be converted into tabular formats using some feature extractors or pre-trained deep models.

For standard anomaly detection, density-based methods, including normalizing flow and other shallow and deep models, are sensitive to the changes of inherent noise in the data, yielding high false-positive rates. It is noteworthy that such inherent noise can be largely different from artificial noise like Gaussian noise, since they could represent minor changes from equipment or environment for data collection. More specifically, real data have inherent noise and can be described by the model $\mathbf{x} = G(\mathbf{s}_D, \mathbf{s}_N)$, where $\mathbf{s}_D$ and $\mathbf{s}_N$ denote the pure data source and noise source respectively, and $G$ is the observation generating function. The changes of $\mathbf{x}$ caused by $\mathbf{s}_N$ should not be treated as anomalies, or at least should be distinguished from the concerned anomalies, and we call such data noisy normal data for convenience. For instance, in a vehicle monitoring system, changes in background noise may alter the observed data, but we are only concerned with the status of the vehicle itself. Similarly, in medical diagnosis, we hope that changes in instruments and equipment or the occasional noise do not affect the diagnostic results for diseases. In Figure 1, we use a synthetic dataset to show the influence of inherent noise on the performance of five anomaly detection methods and our proposed method **N**oise-**R**obust **D**ensity **E**stimation (NRDE). We observe that the five methods fail to distinguish between noisy normal data and real anomalies, exhibiting high false positive rates and low AUROC values, whereas our NRDE is robust to changes in the inherent noise in the data and performs the best.

Our NRDE trains a neural network to estimate the density of pure data with the influence of noise isolated. Specifically, we propose a Jacobian-regularized normalizing flow to estimate the density of

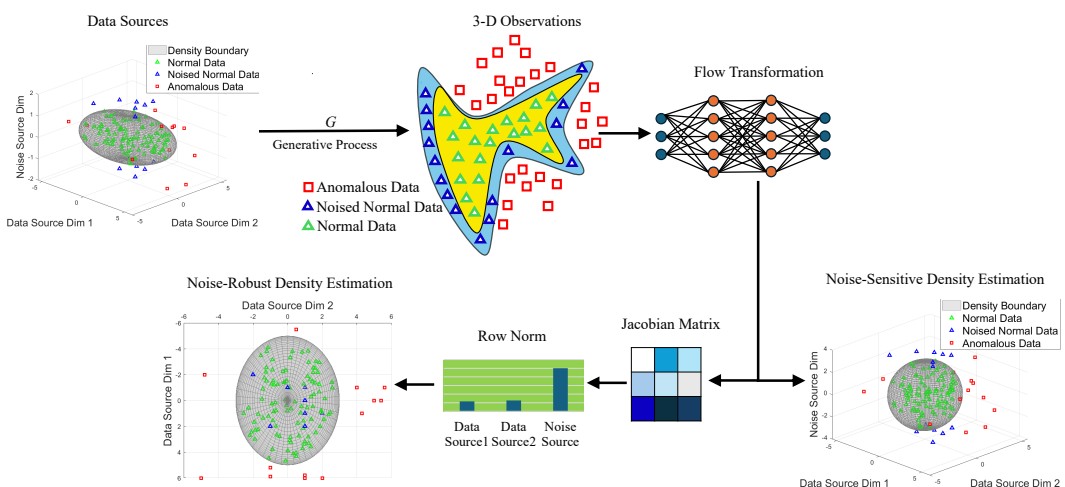

Figure 2: Architecture of the proposed method NRDE. NRDE estimates the density of pure data by utilizing a normalizing flow with Jacobian regularization, where the influence of noise sources is isolated. Therefore, NRDE is robust to the changes of inherent noise in the data.

data and categorize the sources of data into two distinct groups: those that generate pure data and those that produce noise. As a result, we can detect anomalies that are caused by pure data sources without being affected by the noise. The framework of NRDE is shown in Figure 2. Our contributions are summarized as follows:

- We propose a novel density-based AD method, NRDE, for tabular data based on a Jacobian-regularized normalizing flow.
- NRDE categorize data sources into pure-data sources and noise sources and performs density estimation for the pure data only, making it robust to the changes of noise.
- We provide some theoretical analysis for NRDE to support its effectiveness.

We conduct experiments on 47 tabular datasets to compare NRDE against 15 baseline methods. While the primary evaluation is performed under the standard anomaly detection setting, our experimental setup also includes anomaly detection with noise, anomaly detection with contaminated data, and (transductive) outlier detection.

## 2 RELATED WORK AND PRELIMINARY KNOWLEDGE

### 2.1 GENERATIVE MODELS FOR ANOMALY DETECTION

Deep generative models (Schlegl et al., 2019; Kirichenko et al., 2020; Xia et al., 2022; Liu et al., 2025) are useful in anomaly detection due to their ability to model complex data. For instance, OCGAN (Perera et al., 2019) trains a generative adversarial network (Goodfellow et al., 2014) using only normal data for one-class novelty detection. It constrains the latent space of an auto-encoder to represent only the given class by bounding the space and using adversarial discriminators to ensure latent codes and generated samples resemble the in-class data. The model is further refined by exploring latent points that produce out-of-class samples, strengthening its ability to reject novelties. (Yu et al., 2021) learns to transform the visual feature by deep feature extractors into a tractable distribution and obtains the likelihood to recognize anomalies in the inference phase. RobustRealNVP (Liu et al., 2022) ignores low-density points that are likely to be anomalies, by discarding the gradient produced by these points in the training stage, and therefore obtains a robust density function. Kim et al. (2023) trains a normalizing flow to map the feature distributions of each location in normal data to distinct distributions, while mapping the distribution of abnormal data to one that is significantly different from that of normal data, thereby enhancing discriminability. DTE (Livernoche et al., 2023) estimates the distribution over diffusion time for a given input and uses the mode or mean of this distribution as the anomaly score. In (Rozner et al., 2023), the authors found that density functions around normal samples are relatively stable and proposed to use an autoregressive probabilistic model to maximize the density of training samples while minimizing their density variance. Unfortunately,

these works do not address the problem of sensitivity to inherent noise change in standard anomaly detection shown by Figure 1.

## 2.2 Independent Component Analysis and Normalizing Flow

ICA (Hyvärinen & Oja, 2000) assumes that observed data is generated by an unknown mixing process of several independent components (sources) which are from simple distributions, and tries to obtain these components. By categorizing the mixing process, we can divide ICA methods into linear ICA and nonlinear ICA. Linear ICA assumes that the mixing process is linear and the sources are non-Gaussian, and often solves the problem by maximizing the non-Gaussianity. As for nonlinear ICA, the mixing process is assumed to be nonlinear, and the main problem faced by the field is that the model is unidentifiable or the sources are inseparable. In other words, there are infinitely many ways to transform the data into independent components, which is still a mixture of underlying sources. By utilizing additional structure in the data or introducing auxiliary variables, many methods (Hyvärinen & Pajunen, 1999; Hyvarinen & Morioka, 2016; Zheng et al., 2022) have been developed.

Here, we briefly review the foundational concept of normalizing flows. Given a set of observations, each of which, denoted as $\mathbf{x}$, is drawn from some complex distribution $\mathcal{X}$ in $\mathbb{R}^d$, normalizing flow aims to learn a function $F_{\mathcal{W}} : \mathbb{R}^d \to \mathbb{R}^d$ composed of a sequence of invertible mappings $\{f_{\mathcal{W}_t}\}_{t=1}^T$, i.e., $F_{\mathcal{W}} = f_{\mathcal{W}_T} \circ \cdots \circ f_{\mathcal{W}_2} \circ f_{\mathcal{W}_1}$, that transforms complex distribution $\mathcal{X}$ into a simpler one, denoted as $\mathcal{Z}$, such as a standard Gaussian $\mathcal{N}(\mathbf{0}, \mathbf{I})$. Here, $T$ is the number of mappings and $\mathcal{W} = \{\mathcal{W}_1, \ldots, \mathcal{W}_T\}$ denotes the set of all neural network parameters. Because $F_{\mathcal{W}}$ is invertible, the density $p_{\mathcal{X}}(\mathbf{x})$ of $\mathbf{x}$ can be computed using the change-of-variables formula:

$$p_{\mathcal{X}}(\mathbf{x}) = p_{\mathcal{Z}}(F_{\mathcal{W}}(\mathbf{x}))|\det(\nabla_{\mathbf{x}} F_{\mathcal{W}}(\mathbf{x}))|, \tag{1}$$

where $\det(\nabla_{\mathbf{x}} F_{\mathcal{W}}(\mathbf{x}))$ is the determinant of the Jacobian matrix of $F_{\mathcal{W}}$ evaluated at $\mathbf{x}$. One of the coupling normalizing flows is the RealNVP proposed by (Dinh et al., 2016), where $f_{\mathcal{W}_i}$ is called the coupling transformation. Denoting $\mathbf{x}^{(i)} \in \mathbb{R}^d$ the input of $f_{\mathcal{W}_i}$, $\mathbf{x}^{(i)}$ is usually split into two parts, i.e., $\mathbf{x}_\alpha^{(i)} = [x_{\alpha_1}^{(i)}, x_{\alpha_2}^{(i)}, \ldots, x_{\alpha_{q_i}}^{(i)}]^\top$ and $\mathbf{x}_\beta^{(i)} = [x_{\beta_1}^{(i)}, x_{\beta_2}^{(i)}, \ldots, x_{\beta_{d-q_i}}^{(i)}]^\top$, where $1 < q_i < d$. Then the output $\mathbf{y}^{(i)}$ of $f_{\mathcal{W}_i}$ is given as

$$\mathbf{y}_\alpha^{(i)} = \mathbf{x}_\alpha^{(i)}, \quad \mathbf{y}_\beta^{(i)} = \mathbf{x}_\beta^{(i)} \odot \exp(h_{i1}(\mathbf{x}_\alpha^{(i)})) + h_{i2}(\mathbf{x}_\alpha^{(i)}), \tag{2}$$

where $h_{i1} : \mathbb{R}^{q_i} \to \mathbb{R}^{d-q_i}$ and $h_{i2} : \mathbb{R}^{q_i} \to \mathbb{R}^{d-q_i}$ are two multilayer neural networks.

## 3 Proposed Method

### 3.1 Formulation of Noise-Robust Anomaly Detection

Let $\mathcal{D} = \{\mathbf{x}^{(1)}, \mathbf{x}^{(2)}, \ldots, \mathbf{x}^{(n)}\}$ be a set of $d$-dimensional training data, which is drawn from an unknown distribution $\mathcal{X}$. The primary goal of anomaly detection (AD) is to learn a model $\Phi : \mathbb{R}^d \to \mathbb{R}$ from the training set $\mathcal{D}$, which can quantify the degree of anomaly or the dissimilarity of a new sample $\mathbf{x}_{\text{new}}$ relative to the distribution $\mathcal{X}$.

As mentioned in the technique of independent components analysis (ICA) (Hyvärinen & Oja, 2000; Hyvärinen et al., 2009), an observation $\mathbf{x}$ can be regarded as given by an unknown invertible linear or nonlinear transformation, denoted as $G : \mathbb{R}^d \to \mathbb{R}^d$, on some unknown source $\mathbf{s} \in \mathbb{R}^d$, i.e.,

$$\mathbf{x} = G(\mathbf{s}), \tag{3}$$

where $\mathbf{s} \sim \mathcal{S}$. It is natural to assume that the source distribution $\mathcal{S}$ is simple and each dimension of $\mathcal{S}$ is independent. For instance[1], consider $\mathcal{S} = \mathcal{N}(\boldsymbol{\mu}, \boldsymbol{\Sigma})$, where $\boldsymbol{\mu} = [\mu_1, \mu_2, \ldots, \mu_d]^\top$, $\boldsymbol{\Sigma} = \text{diag}(\sigma_1^2, \sigma_2^2, \ldots, \sigma_d^2)$, and $\sigma_1 \geq \sigma_2 \geq \cdots \geq \sigma_d$. For convenience, we consider that the primary distinction among these sources resides in their variances, leading to the specification $\mathcal{S} = \mathcal{N}(\mathbf{0}, \boldsymbol{\Sigma})$. Based on $G$, the ideal normalizing flow can be formulated as:

$$F_{\mathcal{W}}^*(\mathbf{x}) := \boldsymbol{\Sigma}^{-\frac{1}{2}} G^{-1}(\mathbf{x}), \tag{4}$$

---

[1]Although the standard ICA requires an assumption that the sources are non-Gaussian, the Gaussian assumption in this work makes sense because $G$ may first convert each source to non-Gaussian and then perform mixing.

where $\mathbf{z} = F^*_{\mathcal{W}}(\mathbf{x}) = \mathbf{\Sigma}^{-\frac{1}{2}}\mathbf{s} \sim \mathcal{N}(\mathbf{0}, \mathbf{I})$.

We split the source $\mathbf{s}$ into two distinct parts:

$$\mathbf{s} = [\mathbf{s}_D; \mathbf{s}_N], \tag{5}$$

where $\mathbf{s}_D \in \mathbb{R}^m$ denotes the pure data (or signal) source and $\mathbf{s}_N \in \mathbb{R}^{d-m}$ denotes the noise source. It is natural to assume that the variances of $\mathbf{s}_D$ are much greater than those of $\mathbf{s}_N$, namely,

$$\sigma_1 \geq \sigma_2 \cdots \geq \sigma_m > c\sigma_{m+1} \geq c\sigma_{m+2} \cdots \geq c\sigma_d, \tag{6}$$

where $c$ is some constant much greater than 1. The data with noise removed, i.e., pure data, is

$$\mathbf{x}_{\text{pure}} = G([\mathbf{s}_D; \mathbf{0}]). \tag{7}$$

Thus, the inherent noise in data is $\boldsymbol{\epsilon} := \mathbf{x} - \mathbf{x}_{\text{pure}}$. Letting $\mathcal{T}$ denote the signal source distribution deemed as normal, we have the following categorization for the data:

$$\begin{aligned}
\text{pure normal data}: \ & \mathbf{x}_{\text{pure}} = G([\mathbf{s}_D; \mathbf{0}]), \ \mathbf{s}_D \sim \mathcal{T} \\
\text{noisy normal data}: \ & \mathbf{x}_{\text{norm}} = G([\mathbf{s}_D; \mathbf{s}_N]), \ \mathbf{s}_D \sim \mathcal{T}, \ \mathbf{s}_N \neq \mathbf{0} \\
\text{anomalous data}: \ & \mathbf{x}_{\text{anom}} = G([\mathbf{s}_D; \mathbf{s}_N]), \ \mathbf{s}_D \not\sim \mathcal{T}
\end{aligned} \tag{8}$$

In this work, given the observation $\mathbf{x}$, we want to recover $\mathbf{x}_{\text{pure}}$, and evaluate whether $\mathbf{x}_{\text{pure}}$ is normal or anomalous, which is determined by $\mathbf{s}_D$ only and is irrelevant to $\mathbf{s}_N$.

**Rationality of the assumption in (6)**: This assumption is rational because a meaningful signal, by definition, should contain structured information and variation that differentiates it from the background. Noise, often arising from random and uncorrelated processes, tends to have its energy dispersed without a dominant structure. Therefore, the variance of the signal, which captures its total power and variability, is expected to be higher than that of the noise. This is a common and often necessary condition for the signal to be detectable and analyzable amidst the random fluctuations. For instance, in machine learning and statistics, PCA (Jolliffe & Cadima, 2016) assumes the most important data patterns are the directions with the highest variance, effectively treating them as the "signal" and discarding low-variance "noise." In signal processing, denoising filters work by removing low-power (low-variance) frequencies assumed to be noise, while preserving high-power (high-variance) frequencies considered to be the signal.

## 3.2 Signal and Noise Isolation

To realize the aforementioned noise-robust anomaly detection, we need to calculate $p_{\mathcal{X}}(\mathbf{x}_{\text{pure}})$ or $p_{\bar{\mathcal{X}}}(\mathbf{x}_{\text{pure}})$, where $\mathcal{X}$ denotes the distribution of $\mathbf{x}$ and $\bar{\mathcal{X}}$ denotes the distribution of $\mathbf{x}_{\text{pure}}$ defined on the $m$-dimensional manifold embedded in $\mathbb{R}^d$. When $p_{\mathcal{X}}(\mathbf{x}_{\text{pure}})$ or $p_{\bar{\mathcal{X}}}(\mathbf{x}_{\text{pure}})$ are smaller, $\mathbf{x}_{\text{pure}}$, as well as the corresponding noisy counterpart $\mathbf{x}$, is more likely to be anomalous.

As the $p_{\mathcal{X}}(\mathbf{x}_{\text{pure}})$ and $p_{\bar{\mathcal{X}}}(\mathbf{x}_{\text{pure}})$ are closely related (see Appendix A.4) and they have very similar performance in our experiments (see Appendix H.11), we here focus on $p_{\mathcal{X}}(\mathbf{x}_{\text{pure}})$. Let $F_{\mathcal{W}}$ be the flow model learned from $\mathcal{D}$ and suppose $\mathbf{x}_{\text{pure}}$ can be identified from $\mathbf{x}$, we can obtain

$$p_{\mathcal{X}}(\mathbf{x}_{\text{pure}}) = p_{\mathcal{Z}}(F_{\mathcal{W}}(\mathbf{x}_{\text{pure}})) \left| \det(\nabla_{\mathbf{x}_{\text{pure}}} F_{\mathcal{W}}(\mathbf{x}_{\text{pure}})) \right|. \tag{9}$$

Using (4), we have the ideal case for $p_{\mathcal{X}}(\mathbf{x}_{\text{pure}})$, i.e.,

$$\begin{aligned}
\log p^*_{\mathcal{X}}(\mathbf{x}_{\text{pure}}) &= \log p_{\mathcal{Z}}(F^*_{\mathcal{W}}(\mathbf{x}_{\text{pure}})) + \log |\det(\nabla_{\mathbf{x}_{\text{pure}}} F^*_{\mathcal{W}}(\mathbf{x}_{\text{pure}}))| \\
&= \log\left((2\pi)^{-\frac{d}{2}} \exp(-\tfrac{1}{2}\mathbf{s}^\top \mathbf{\Sigma}^{-1}\mathbf{s})\right) + \log |\det(\nabla_{\mathbf{x}_{\text{pure}}} F^*_{\mathcal{W}}(\mathbf{x}_{\text{pure}}))| \\
&= \log |\det(\nabla_{\mathbf{x}_{\text{pure}}} F^*_{\mathcal{W}}(\mathbf{x}_{\text{pure}}))| - \sum_{i=1}^{m} \frac{s_i^2}{2\sigma_i^2} - \frac{d}{2}\log(2\pi)
\end{aligned} \tag{10}$$

where we have used the fact that $F^*_{\mathcal{W}}(\mathbf{x}_{\text{pure}}) = [\mathbf{z}_D; \mathbf{0}]$ and $\mathbf{z} = \mathbf{\Sigma}^{-\frac{1}{2}}\mathbf{s}$. The challenge is that we may never obtain $F^*_{\mathcal{W}}$. The learned $F_{\mathcal{W}}$ from $\mathcal{D}$ can only ensure that $\mathbf{z} = F_{\mathcal{W}}(\mathbf{x}) \sim \mathcal{N}(\mathbf{0}, \mathbf{I})$. It is difficult to determine which of $z_1, \ldots, z_d$ correspond to $\mathbf{s}_D$ and which of $z_1, \ldots, z_d$ correspond to $\mathbf{s}_N$. Moreover, the number of data sources $m$ is unknown and is not easy to estimate. In the following context, we show how to address these problems.

Note that (4) indicates that

$$\frac{\partial z_j}{\partial \mathbf{x}} = \sigma_j^{-1} \times \frac{\partial G_j^{-1}(\mathbf{x})}{\partial \mathbf{x}}. \tag{11}$$

We assume that

$$\gamma - \delta \leq \left\| \frac{\partial G_j^{-1}(\mathbf{x})}{\partial \mathbf{x}} \right\| \leq \gamma + \delta, \quad \forall j \in [d], \tag{12}$$

where $\gamma$ and $\delta$ are some positive constants and $\delta \ll \gamma$. This assumption is reasonable because $G$ usually mixes the sources randomly and uniformly. Moreover, it is more general than the assumption used in linear ICA (Hyvärinen et al., 2001), which assumes $\mathbf{W}^\top \mathbf{W} = \mathbf{I}$ in $G(\mathbf{s}) = \mathbf{W}\mathbf{s}$, meaning $\gamma = 1$ and $\delta = 0$. Combining (11) and (12), we have

$$(\gamma - \delta) \left\| \frac{\partial z_j}{\partial \mathbf{x}} \right\|^{-1} \leq \sigma_j \leq (\gamma + \delta) \left\| \frac{\partial z_j}{\partial \mathbf{x}} \right\|^{-1}. \tag{13}$$

If $(\gamma - \delta) \left\| \frac{\partial z_j}{\partial \mathbf{x}} \right\|^{-1} > c(\gamma + \delta) \left\| \frac{\partial z_{j'}}{\partial \mathbf{x}} \right\|^{-1}$ or $\left\| \frac{\partial z_{j'}}{\partial \mathbf{x}} \right\| > c\frac{\gamma+\delta}{\gamma-\delta} \left\| \frac{\partial z_j}{\partial \mathbf{x}} \right\|$ equivalently, then $\sigma_j > c\sigma_{j'}$. This means we may compare $\left\| \frac{\partial z_1}{\partial \mathbf{x}} \right\|, \ldots, \left\| \frac{\partial z_d}{\partial \mathbf{x}} \right\|$ to distinguish between $\mathbf{s}_D$ and $\mathbf{s}_N$. However, a clear gap may not exist between $\left\| \frac{\partial z_1}{\partial \mathbf{x}} \right\|, \ldots, \left\| \frac{\partial z_d}{\partial \mathbf{x}} \right\|$. An intuitive example is shown in Figure 3. The reason is that the source $\mathbf{s}$ in (3) is not identifiable and there are many equivalent problems (Hyvärinen & Pajunen, 1999; Hyvarinen et al., 2019; Zheng et al., 2022). For instance, let $\mathbf{R}$ be an orthonormal matrix and $F_{\mathcal{W}}(\mathbf{x}) = \mathbf{R}F_{\mathcal{W}}^*(\mathbf{x})$ is a normalizing flow learned from $\mathcal{D}$. In this case, $F_{\mathcal{W}}(\mathbf{x}) \sim \mathcal{N}(\mathbf{0}, \mathbf{I})$ and the estimated density remains unchanged. However, $F_{\mathcal{W}}(\mathbf{x})$ becomes a combination of $\mathbf{z}$, and the row norms of the Jacobian matrix do not reflect the variances of sources.

However, we can exploit the prior knowledge (6) to train $F_{\mathcal{W}}$ and may consider the optimization

$$\underset{\mathcal{W},A,B}{\text{maximize}} \sum_{\mathbf{x} \in \mathcal{D}} \log \left( p_{\mathcal{Z}}(F_{\mathcal{W}}(\mathbf{x})) | \det(\nabla_{\mathbf{x}} F_{\mathcal{W}}(\mathbf{x})) | \right)$$

$$\text{subject to } \min_{j \in A} \left\| \frac{\partial z_j}{\partial \mathbf{x}} \right\|^{-1} > c' \max_{j \in B} \left\| \frac{\partial z_j}{\partial \mathbf{x}} \right\|^{-1}, \ \forall \mathbf{x} \in \mathcal{D} \tag{14}$$

$$A \cup B = [d], \ A \cap B = \emptyset, \ |A| = m$$

where $c' = c\frac{\gamma+\delta}{\gamma-\delta}$ and $A$ corresponds to $\mathbf{s}_D$ and $B$ corresponds to $\mathbf{s}_N$. It is very difficult to solve (14) because $c, \gamma, \delta$ are unknown and the constraints are related to every $\mathbf{x}$ and min and max operations. We also need to know $m$.

### 3.3 JACOBIAN-REGULARIZED NORMALIZING FLOW

The constraints in (14) indicate that some rows of the Jacobian matrix $\nabla_{\mathbf{x}} F_{\mathcal{W}}(\mathbf{x})$ have much smaller norms than other rows, which is a kind of sparseness. Therefore, we propose to regularize $\nabla_{\mathbf{x}} F_{\mathcal{W}}(\mathbf{x})$ during the optimization of $F_{\mathcal{W}}$ and hence solve

$$\underset{\mathcal{W}}{\text{minimize}} \ \frac{1}{n} \sum_{\mathbf{x} \in \mathcal{D}} -\log \left( p_{\mathcal{Z}}(F_{\mathcal{W}}(\mathbf{x})) | \det(\nabla_{\mathbf{x}} F_{\mathcal{W}}(\mathbf{x})) | \right) + \lambda \mathcal{R} \left( \frac{1}{n} \sum_{\mathbf{x} \in \mathcal{D}} |\nabla_{\mathbf{x}} F_{\mathcal{W}}(\mathbf{x})| \right), \tag{15}$$

where $\mathcal{R}$ denotes a sparse regularizer on matrix and $\lambda > 0$ is a hyperparameter. Instead of regularizing for each $\mathbf{x}$ of $\mathcal{D}$, we regularize the average of absolute Jacobian matrices. We use the following $\mathcal{R}$:

$$\mathcal{R}(\mathbf{Q}) = \sum_{i=1}^{d} \sqrt{\|\mathbf{q}_{i:}\|_1}, \tag{16}$$

where $\mathbf{q}_{i:}$ denotes the $i$th row of $\mathbf{Q} \in \mathbb{R}^{d \times d}$. Note that $\mathcal{R}^2(\mathbf{Q})$ is the $\ell_{1,1/2}$ quasi-norm, which is sharper than $\ell_{2,1}$ norm widely used in sparse optimization. Figure 3 illustrates the effect of $\mathcal{R}$. More details about $\mathcal{R}$ is provided in Appendix C.

An alternative to (16) is using $\mathbf{R}(\mathbf{Q}) = \sum_{j \in [B]} \|\mathbf{q}_{j:}\| - \sum_{j \in [A]} \|\mathbf{q}_{j:}\|$, where $A$ is the index set of the $m$ rows of $\mathbf{Q}$ with smaller norms and $B$ is the index set of the $d - m$ rows of $\mathbf{Q}$ with larger

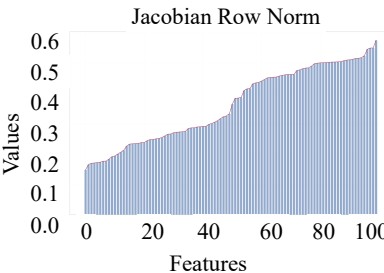 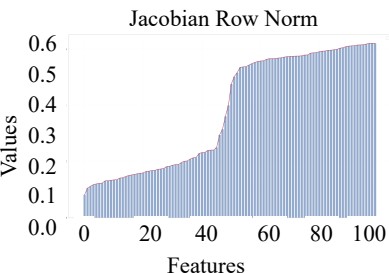

Figure 3: Visualization of row norms of the Jacobian matrix on a synthetic dataset with 50 pure data sources and 50 noise sources. The left one shows the unregularized case, while the right one shows the regularized case. More visualization results on real datasets are shown in Appendix G.1.

norms determined in each iteration. This method requires a good estimate of $m$ and is sensitive to the initialization. The performance is not as good as (16).

Although solving (15) makes sense, in real scenarios, $m$ is hard to estimate, the obtained $F_{\mathcal{W}}$ may not provide a very clear gap between the norms of rows of the Jacobian matrix, and the gap depends on $\lambda$. Therefore, we use a weighted log density $u(\mathbf{x})$ to approximate $\log p_{\mathcal{X}}^*(\mathbf{x}_{\text{pure}})$, which can be regarded as an anomaly score. To be more precise, given a test sample $\mathbf{x}_{\text{new}}$, we compute the anomaly score as $u(\mathbf{x}_{\text{new}})$ where a lower anomaly score indicates a higher probability of being an anomaly:

$$u(\mathbf{x}_{\text{new}}) = \log \left| \det \left( \nabla_{\mathbf{x}_{\text{new}}} F_{\mathcal{W}}(\mathbf{x}_{\text{new}}) \right) \right| - \frac{d}{2} \log 2\pi - \frac{1}{2} \sum_{i=1}^{d} w_i F_{\mathcal{W}}(\mathbf{x}_{\text{new}})_i^2, \quad (17)$$

where the weights $w_i$ are

$$w_i = \exp \left( \frac{1}{\|(\frac{1}{n} \sum_{\mathbf{x} \in \mathcal{D}} |\nabla_{\mathbf{x}} F_{\mathcal{W}}(\mathbf{x})|)_i\| + 1} \right) \bigg/ \sum_{j=1}^{d} \exp \left( \frac{1}{\|(\frac{1}{n} \sum_{\mathbf{x} \in \mathcal{D}} |\nabla_{\mathbf{x}} F_{\mathcal{W}}(\mathbf{x})|)_j\| + 1} \right), \quad (18)$$

and

$$\frac{1}{\|(\frac{1}{n} \sum_{\mathbf{x} \in \mathcal{D}} |\nabla_{\mathbf{x}} F_{\mathcal{W}}(\mathbf{x})|)_i\| + 1} \approx \frac{\sigma_i}{\sigma_i + \gamma + \delta}, \quad (19)$$

Note that $w_i$ is larger for the sources with a larger variance $\sigma_i$, which is more likely to be a data source. Using such weighted log density achieves performance comparable to directly computing $p_{\mathcal{X}}(\mathbf{x}_{\text{pure}})$ or $p_{\bar{\mathcal{X}}}(\mathbf{x}_{\text{pure}})$. However, the latter suffers from a practical limitation—its computation depends explicitly on the number of data sources $m$. Detailed results are provided in Appendix H.11.

In summary, we train a Jacobian-regularized normalizing flow via (15). After the model is well-trained, for any testing data, we can calculate $u(\mathbf{x}_{\text{new}})$ using (17) to approximate the density of pure data and use it as the anomaly score to determine whether $\mathbf{x}_{\text{new}}$ is anomalous or not. More details about the algorithm of NRDE are shown in Appendix F.

### 3.4 THEORETICAL GUANRANTEES

We provide the following theoretical guarantee for $u(\mathbf{x})$ to approximate $\log p_{\mathcal{X}}^*(\mathbf{x}_{\text{pure}})$ and detect anomaly successfully.

**Theorem 3.1.** *Let $\mathbf{x}$ be a normal data point and $\hat{\mathbf{x}}$ be an anomaly. If $\|F_{\mathcal{W}}^*(\mathbf{x})\| = \|F_{\mathcal{W}}^*(\hat{\mathbf{x}})\|$ and $p_{\mathcal{X}}(\mathbf{x}) = p_{\mathcal{X}}(\hat{\mathbf{x}})$, then the weighted log-density $u^*$ based on $F_{\mathcal{W}}^*$ satisfies $u^*(\mathbf{x}) > u^*(\hat{\mathbf{x}})$.*

Theorem 3.1 provides a guarantee for our proposed method to identify the anomalies and noisy normal data that normalizing flow is unable to identify. When $\mathbf{x}$ and $\hat{\mathbf{x}}$ share the same estimated density, normalizing flow is unable to detect such anomalies. Moreover, if their output norms are also the same, the resulting misclassification is due to the presence of noise sources, implying that $\mathbf{x}$ is a noisy normal data. By determining the weight $w_i^*$ for each source, where noise sources naturally receive much smaller weights, the influence of noise sources is minor. Consequently, the weighted log density $u^*$ is dominated by the data sources, enabling reliable discrimination between $\mathbf{x}$ and $\hat{\mathbf{x}}$.

**Theorem 3.2.** *Let the Lipschitz constant of each $f^*_{\mathcal{W}_i}$ and $h^*_{i1}$ be bounded above by $\tau^+_i$ and $\tau_{i\alpha}$ respectively and denote the weight and weighted log density estimated by $F^*_{\mathcal{W}}(\mathbf{x})$ as $\{w^*_i\}^d_{i=1}$ and $u^*(\mathbf{x})$ respectively. Suppose $|u^*(\mathbf{x}) - u(\mathbf{x})| \leq \eta$, then the following inequality holds:*

$$|\log p^*_{\mathcal{X}}(\mathbf{x}_{pure}) - u(\mathbf{x})| \leq \sum_{i \in A} \left(1 - w^*_i\right) F^*_{\mathcal{W}}(\mathbf{x})^2_i + \sum_{j \in B} w^*_j F^*_{\mathcal{W}}(\mathbf{x})^2_j$$

$$+ \sum^T_{i=1} (\tau_{i\alpha} \sqrt{d} \prod^{i-1}_{j=1} \tau^+_j) \|\mathbf{x} - \mathbf{x}_{pure}\| + \eta$$

This theorem indicates that our method can approximate the density of the pure data. Note that when $\sigma_m > c\sigma_{m+1}$, as defined before, the principal estimation error originates from noise $\|\mathbf{x} - \mathbf{x}_{\text{pure}}\|$ and $\eta$, which are intrinsic properties of the data and the regularized normalizing flow respectively.

**Assumption 3.3.** *For any $\mathbf{x}_a, \mathbf{x}_b \in \mathbb{R}^d$, there exists a constant $\varphi$ such that $|u(\mathbf{x}_a) - u(\mathbf{x}_b)| \geq \varphi \|\mathbf{x}_a - \mathbf{x}_b\|$ and if $\hat{\mathbf{x}}$ is an anomaly, $u(\hat{\mathbf{x}}) \leq \max_{\mathbf{x} \in \mathcal{D}} u(\mathbf{x})$.*

This assumption is reasonable since $\varphi$ can be calculated as $\inf_{\mathbf{x} \in \mathbb{R}^d} \|\nabla_{\mathbf{x}} u(\mathbf{x})\|$.

**Theorem 3.4.** *Let $\hat{\mathbf{x}}$ be an anomaly. Suppose that $\mathbf{x}_a, \mathbf{x}_b \in \mathcal{D}$ such that $\arg\max_{\mathbf{x}} u(\mathbf{x}) = \mathbf{x}_a$ and $\arg\min_{\mathbf{x}} u(\mathbf{x}) = \mathbf{x}_b$ and $u(\mathbf{x}_a) = \varsigma_1, u(\mathbf{x}_b) = \varsigma_2$. Then, under the Assumption 3.3, if $\|\hat{\mathbf{x}} - \mathbf{x}_a\| > \frac{\varsigma_1 - \varsigma_2}{\varphi}$, then $\hat{\mathbf{x}}$ can be detected as an anomaly.*

Theorem 3.4 shows that our proposed method can detect anomalies that are significantly distant from normal data. Furthermore, if an anomaly possesses a weighted log-density exceeding the maximum weighted log-density observed in the training set, its detection becomes considerably more challenging or even impossible. The proofs for the theorems are in Appendix A. Also, we compare the time complexity of density-based Methods in Appendix B.

# 4 NUMERICAL RESULTS

## 4.1 EXPERIMENTAL SETTINGS

**Datasets** In our experiments, we evaluate the performance of 15 baseline methods on 47 widely used real-world datasets spanning multiple domains in a popular benchmark for anomaly detection proposed by (Han et al., 2022). Detailed descriptions and statistical information about these datasets are provided in the Appendix E. In anomaly detection tasks, we follow the protocol of (Zong et al., 2018; Bergman & Hoshen, 2020; Shenkar & Wolf, 2022; Xu et al., 2023b) by randomly partitioning normal samples: 50% are training, while the remaining 50% are combined with all anomalous samples to form the test set. For outlier detection, the model is trained on the entire dataset to identify outliers, which is a transductive learning setting.

**Baselines** Our method is compared with 15 baselines, including DTE (Livernoche et al., 2023), MCM (Yin et al., 2024), DPAD (Fu et al., 2024), SLAD (Xu et al., 2023b), ECOD (Li et al., 2022), ICL (Shenkar & Wolf, 2022), NeutralAD (Qiu et al., 2021), DSVDD (Ruff et al., 2018a), RealNVP (Dinh et al., 2016), IF (Liu et al., 2008), AE (Hinton & Salakhutdinov, 2006), LOF (Breunig et al., 2000), $k$NN (Ramaswamy et al., 2000), KDE (Parzen, 1962). For DTE, MCM, DPAD, SLAD, ICL, and NeutralAD, we use the code provided by the authors of the papers. For other methods, we use the code from the Python library PyOD (Chen et al., 2024). All hyperparameters follow the recommended settings.

**Implementation** We use the Area Under the Receiver Operating Characteristic Curve (AUROC) and the Area Under the Precision-Recall Curve (AUPRC) as evaluation metrics, following (Xu et al., 2023b; Han et al., 2022). These two metrics do not rely on specific thresholds of decision and are capable of comprehensively assessing the performance of different methods. All experiments are conducted using the PyTorch framework on a system equipped with an NVIDIA RTX 3090 GPU and an Intel Core i9-12900K CPU. Each experiment is performed five times to obtain the mean value and standard deviation. To ensure a consistent network architecture for fair comparison, we employ two 2-layer multilayer perceptrons (MLPs), corresponding to a parameter setting of $T = 2$ in (1). More details are in Appendix D.

## 4.2 RESULTS OF STANDARD ANOMALY DETECTION

Table 1: AUROC (%) and AUPRC (%) with the standard deviation of each method on several tabular datasets of ADBench. The best results are marked in **bold**.

| AUROC | KDE | KNN | LOF | OC-SVM | IF | AE | DSVDD | RealNVP | NeutralAD | ECOD | ICL | SLAD | DPAD | MCM | DTE-C | Ours |
|---|---|---|---|---|---|---|---|---|---|---|---|---|---|---|---|---|
| annthyroid | 91.4 ± 0.0 | 94.1 ± 0.0 | 92.9 ± 0.0 | 90.9 ± 0.0 | 91.8 ± 1.1 | 83.4 ± 2.0 | 79.4 ± 3.2 | 96.1 ± 0.5 | 78.9 ± 2.8 | 78.7 ± 0.0 | 64.0 ± 6.1 | 90.4 ± 2.9 | 91.2 ± 4.7 | 83.9 ± 0.6 | 97.8 ± 0.0 | **98.4 ± 0.0** |
| breastw | 98.9 ± 0.0 | 99.1 ± 0.0 | 96.7 ± 0.0 | 99.0 ± 0.0 | **99.5 ± 0.0** | 98.4 ± 0.3 | 99.1 ± 0.1 | 98.0 ± 0.0 | 81.4 ± 3.9 | 99.3 ± 0.0 | 90.2 ± 1.3 | 99.2 ± 0.1 | 98.9 ± 0.2 | 99.0 ± 0.0 | 96.3 ± 0.0 | 99.4 ± 0.0 |
| cardio | 95.7 ± 0.0 | 93.4 ± 0.0 | 93.0 ± 0.0 | 96.4 ± 0.0 | 94.9 ± 1.1 | 92.4 ± 3.2 | **96.1 ± 0.3** | 94.1 ± 0.4 | 81.0 ± 1.9 | 93.4 ± 0.0 | 83.9 ± 1.5 | 88.7 ± 3.0 | 89.0 ± 3.3 | 90.4 ± 0.8 | 93.6 ± 0.0 | 95.8 ± 0.6 |
| Cardiotocography | 75.0 ± 0.0 | 71.3 ± 0.0 | 72.7 ± 0.0 | 80.7 ± 0.0 | 79.3 ± 2.5 | 73.4 ± 2.4 | 83.4 ± 2.5 | 77.9 ± 1.8 | 58.2 ± 1.9 | 78.5 ± 0.0 | 54.7 ± 11.9 | 58.4 ± 2.1 | 68.0 ± 3.1 | 70.0 ± 0.9 | 72.4 ± 0.0 | **86.1 ± 2.4** |
| celeba | 70.5 ± 0.0 | 68.0 ± 0.0 | 44.9 ± 0.0 | 79.0 ± 0.0 | 70.8 ± 0.7 | 70.9 ± 1.0 | 48.4 ± 0.8 | 79.4 ± 0.6 | 66.4 ± 9.4 | 75.7 ± 0.0 | 69.5 ± 0.0 | 65.2 ± 2.1 | 56.3 ± 6.1 | 65.3 ± 3.3 | 82.7 ± 0.0 | **87.9 ± 0.7** |
| census | 72.0 ± 0.0 | 71.9 ± 0.0 | 60.5 ± 0.0 | 70.2 ± 0.0 | 62.7 ± 1.9 | 71.8 ± 0.1 | 51.9 ± 3.1 | 72.8 ± 0.3 | 72.9 ± 2.6 | 65.9 ± 0.0 | 66.8 ± 0.0 | 68.9 ± 0.6 | 50.7 ± 0.8 | 68.1 ± 0.2 | 69.6 ± 0.0 | **76.7 ± 3.0** |
| speech | 45.8 ± 0.0 | 48.5 ± 0.0 | 48.9 ± 0.0 | 45.9 ± 0.0 | 46.7 ± 1.2 | 46.8 ± 0.2 | 45.2 ± 1.2 | 50.0 ± 0.0 | 54.3 ± 4.2 | 46.1 ± 0.0 | 49.1 ± 2.9 | 50.7 ± 3.2 | 54.8 ± 4.6 | 49.9 ± 0.3 | 56.1 ± 0.0 | **64.7 ± 1.9** |
| thyroid | 98.3 ± 0.0 | 98.5 ± 0.0 | 94.6 ± 0.0 | 98.2 ± 0.0 | 99.0 ± 0.2 | 98.0 ± 0.3 | 97.5 ± 0.5 | 98.6 ± 0.1 | 65.2 ± 7.9 | 97.7 ± 0.0 | 82.2 ± 5.2 | 94.8 ± 1.8 | 96.1 ± 1.8 | 97.9 ± 0.3 | 99.2 ± 0.0 | **99.2 ± 0.1** |
| vertebral | 43.5 ± 0.0 | 42.5 ± 0.0 | 40.0 ± 0.0 | 52.7 ± 0.0 | 42.6 ± 4.5 | 48.0 ± 4.3 | 43.7 ± 4.5 | 53.6 ± 4.8 | 53.9 ± 3.0 | 41.8 ± 0.0 | 54.2 ± 5.8 | 44.4 ± 4.3 | 46.4 ± 3.5 | 47.2 ± 1.4 | 59.2 ± 0.0 | **72.7 ± 6.0** |
| Waveform | 76.0 ± 0.0 | 76.2 ± 0.0 | 76.6 ± 0.0 | 69.0 ± 0.0 | 72.5 ± 1.4 | 65.8 ± 2.5 | 69.6 ± 3.8 | 72.5 ± 1.6 | 71.5 ± 0.5 | 60.0 ± 0.0 | 59.8 ± 1.1 | 50.2 ± 4.0 | 61.0 ± 3.5 | 69.6 ± 1.2 | 65.6 ± 0.0 | **91.6 ± 1.1** |
| **AUPRC** | | | | | | | | | | | | | | | | |
| annthyroid | 66.2 ± 0.0 | 72.0 ± 0.0 | 66.7 ± 0.0 | 65.2 ± 0.0 | 63.8 ± 2.8 | 60.7 ± 2.1 | 54.8 ± 2.2 | 77.0 ± 2.8 | 29.4 ± 4.0 | 40.8 ± 0.0 | 31.3 ± 9.5 | 63.1 ± 5.8 | 64.5 ± 9.5 | 55.0 ± 0.6 | **84.1 ± 0.0** | 79.7 ± 2.0 |
| breastw | 98.8 ± 0.0 | 99.1 ± 0.0 | 93.7 ± 0.0 | 98.8 ± 0.0 | **99.5 ± 0.0** | 98.1 ± 0.5 | 99.1 ± 0.1 | 96.7 ± 0.1 | 71.2 ± 3.0 | 99.3 ± 0.0 | 86.3 ± 3.0 | 99.2 ± 0.2 | 98.7 ± 0.2 | 99.0 ± 0.1 | 92.1 ± 0.0 | 99.4 ± 0.0 |
| cardio | 84.0 ± 0.0 | 76.8 ± 0.0 | 69.3 ± 0.0 | 82.8 ± 0.0 | 78.4 ± 4.4 | 74.7 ± 5.9 | **83.0 ± 0.9** | 71.0 ± 2.6 | 48.9 ± 4.3 | 70.9 ± 0.0 | 60.7 ± 3.2 | 72.7 ± 3.0 | 73.5 ± 6.6 | 73.1 ± 1.0 | 69.5 ± 0.0 | 75.9 ± 4.3 |
| Cardiotocography | 68.1 ± 0.0 | 62.4 ± 0.0 | 59.9 ± 0.0 | 71.0 ± 0.0 | 67.6 ± 2.4 | 65.0 ± 2.5 | **75.1 ± 2.5** | 62.6 ± 2.1 | 40.3 ± 2.3 | 65.7 ± 0.0 | 45.4 ± 10.1 | 54.7 ± 1.5 | 61.5 ± 2.6 | 61.3 ± 1.0 | 61.1 ± 0.0 | 74.3 ± 2.7 |
| celeba | 8.9 ± 0.0 | 9.8 ± 0.0 | 3.7 ± 0.0 | 20.4 ± 0.0 | 12.5 ± 0.7 | 9.5 ± 0.2 | 4.0 ± 0.1 | 13.1 ± 0.6 | 6.6 ± 1.5 | 17.2 ± 0.0 | 8.9 ± 0.0 | **76.1 ± 0.4** | 5.8 ± 1.3 | 7.3 ± 1.0 | 15.7 ± 0.0 | 20.1 ± 1.6 |
| census | 21.6 ± 0.0 | 21.2 ± 0.0 | 14.3 ± 0.0 | 20.5 ± 0.0 | 14.2 ± 0.8 | 21.6 ± 0.1 | 11.9 ± 0.8 | 20.5 ± 0.6 | 23.3 ± 0.0 | 15.5 ± 0.0 | 17.4 ± 0.0 | 19.8 ± 0.2 | 12.1 ± 0.5 | 18.8 ± 0.2 | 18.0 ± 0.0 | **24.7 ± 1.6** |
| speech | 3.7 ± 0.0 | 3.7 ± 0.0 | 4.5 ± 0.0 | 3.6 ± 0.0 | 3.5 ± 0.2 | 3.6 ± 0.4 | 3.0 ± 0.2 | 3.2 ± 0.0 | 4.0 ± 0.0 | 3.8 ± 0.0 | 3.3 ± 0.2 | 3.8 ± 0.5 | 4.4 ± 1.1 | 4.4 ± 0.2 | 4.9 ± 0.0 | **5.3 ± 0.7** |
| thyroid | 73.8 ± 0.0 | 77.4 ± 0.0 | 58.8 ± 0.0 | 73.9 ± 0.0 | 83.7 ± 1.6 | 78.3 ± 5.5 | 78.9 ± 2.1 | 76.4 ± 1.9 | 6.2 ± 3.2 | 62.9 ± 0.0 | 28.8 ± 12.6 | 67.6 ± 7.7 | 60.6 ± 5.0 | 71.9 ± 2.8 | 86.4 ± 0.0 | **86.8 ± 1.4** |
| vertebral | 19.7 ± 0.0 | 20.3 ± 0.0 | 19.6 ± 0.0 | 23.1 ± 0.0 | 19.4 ± 1.6 | 21.6 ± 2.7 | 25.3 ± 2.2 | 29.8 ± 1.4 | 29.8 ± 1.4 | 19.5 ± 0.0 | 26.2 ± 4.0 | 21.4 ± 3.3 | 21.2 ± 1.3 | 20.9 ± 0.1 | 27.1 ± 0.0 | **41.0 ± 6.6** |
| Waveform | 27.6 ± 0.0 | 27.0 ± 0.0 | 31.7 ± 0.0 | 10.7 ± 0.0 | 10.8 ± 0.5 | 11.1 ± 1.3 | 9.5 ± 1.3 | 11.3 ± 0.6 | 47.4 ± 2.5 | 7.6 ± 0.0 | 29.6 ± 2.2 | 5.7 ± 1.0 | 12.0 ± 2.1 | 20.0 ± 0.8 | 10.3 ± 0.0 | **34.8 ± 3.5** |

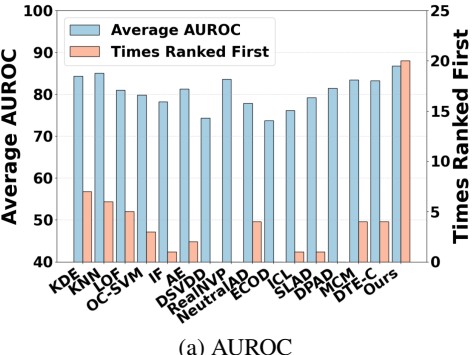

(a) AUROC

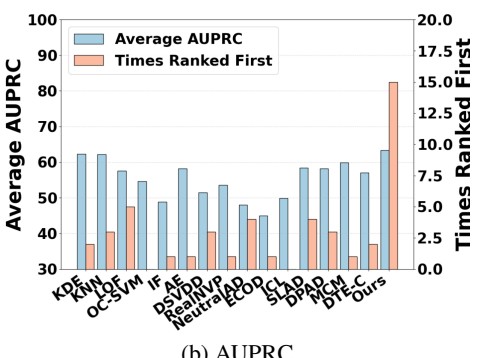

(b) AUPRC

Figure 4: The average AUROC and AUPRC performance of different methods on anomaly detection across 47 datasets, along with the number of datasets where each method is ranked first. Higher values of these metrics indicate better detection performance.

Table 1 reports the performance of different methods on several datasets, while Figure 4 reports the average AUROC and AUPRC results across 47 datasets, with detailed results for each dataset available in Appendix G.2. Our method achieves the best performance, outperforming the second-best method by more than 2%. Compared to RealNVP, NRDE demonstrates a significant improvement, particularly in terms of AUPRC. Additionally, NRDE outperforms other baseline methods on a larger number of datasets. For example, on the Speech, Vertebral, and WPBC, other methods attain AUROC scores around 50%, indicating anomaly detection is particularly challenging for these approaches. In contrast, our method significantly outperforms baselines, highlighting its effectiveness in complex datasets. Notably, density-based methods outperform many deep learning-based approaches, highlighting their effectiveness in anomaly detection. Moreover, KDE and $k$NN—two traditional methods—outperform all deep learning-based baseline methods. We attribute this phenomenon to two main factors. First, as mentioned earlier, tabular data typically consists of features that inherently provide excellent representations of semantic differences. As a result, even the simple Euclidean distance can capture meaningful distinctions between samples. This is also consistent with the results shown in Figure 1, which illustrates the performance of these methods in challenging noisy scenarios. Second, as demonstrated in (Jiang, 2017; Gu et al., 2019), these two methods provide more explicit predictions for datasets with lower dimensions and more samples, which aligns with the experimental results and the curse of dimensionality.

## 4.3 RESULTS OF ANOMALY DETECTION WITH ANOMALY CONTAMINATION

In real applications of AD, the training set often contains a small amount of anomalous data due to various reasons. To evaluate the robustness and performance of all methods in this scenario, we add different ratios of anomalies to the training set and conduct experiments on these contaminated

Table 2: AUROC results (%) of the best-performing 5 methods of anomaly detection in noisy data. The best results per dataset are in **bold**.

| Dataset | DSVDD | KPCA | IF | kNN | NRDE (ours) |
|---|---|---|---|---|---|
| Cardiotocography | **83.7** | 75.8 | 80.7 | 71.3 | 82.1 |
| Pima | 72.5 | 77.0 | 75.8 | 78.1 | **79.6** |
| Satellite | 81.5 | 84.1 | 79.6 | **86.9** | 85.1 |
| SpamBase | 80.3 | **86.3** | 82.4 | 83.0 | 79.1 |
| WPBC | 47.5 | 52.2 | 51.7 | 51.5 | **62.9** |
| AVG | 73.1 | 75.1 | 74.0 | 74.2 | **77.2** |

datasets. The contamination ratio ranges from $1\%$ to $10\%$ of the training set size. We report the average performance of all methods in Figure 5, where the detailed experimental results for each dataset are in Appendix G.3. From the figure, we observe that as the anomaly ratio increases, the performance of all methods decreases. In this scenario, our proposed method consistently achieves superior performance over other methods, demonstrating its robustness to anomalies in the training set. It should be noted that the AUROC performance of our proposed method remains unaffected by the anomaly ratio. While its AUPRC performance is influenced, with a performance drop less significant than that observed in other methods. Additionally, ECOD appears to be the baseline method whose performance is least influenced by the anomaly ratio.

### 4.4 Results of Anomaly Detection on Noisy Data

In real-world anomaly detection scenarios, data are often corrupted by noise. To evaluate the performance of all methods in this complex scenario, we perturb training data and anomalous testing data with Gaussian noise drawn from $\mathcal{N}(\mathbf{0}, 0.1\mathbf{I}_d)$ to training data and anomalous testing data, while normal test samples receive stronger noise $\mathcal{N}(\mathbf{0}, 0.2\mathbf{I}_d)$. Note that the data is first normalized and then corrupted by the noise. Table 2 illustrates the experimental results. Our approach consistently outperforms competing methods, underscoring its robustness to noise.

### 4.5 More Results

The time complexity comparison, more detailed results for visualization, standard anomaly detection, anomaly detection with anomaly contamination, outlier detection, ablation studies, hyperparameter analysis, experiments to verify our assumptions and motivation are in *Appendices* B, G.1, G.2, G.3, G.4, G.5 and G.6 respectively.

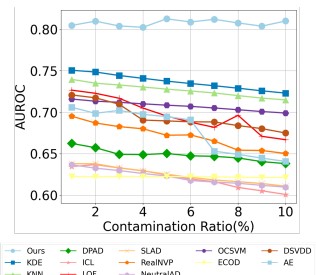

Figure 5: Average AUROC values across 5 datasets of AD experiments with anomaly contamination, contamination ratio ranging from $1\%$ to $10\%$.

## 5 Conclusion

We proposed a novel and effective method NRDE for anomaly detection in tabular data. Our key observation is that data is typically generated by independent sources, which can be categorized into pure data sources and noise sources. By distinguishing these sources using the Jacobian matrix, we can approximate the density of the pure data with a weighted log density that is unaffected by noise. This allows NRDE to be robust to noise and effectively identify both anomalous data and noisy normal data. We provided theoretical analysis on the estimation error, the reliability of our proposed method, and the time complexity of density-based approaches. Numerical experiments demonstrated that NRDE outperforms 15 baseline methods across 47 real-world datasets. Furthermore, NRDE exhibits robustness to anomalies in the training set and noise inside the data.

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

# A    PROOF FOR THEOREMS

## A.1    PROOF FOR THEOREM 3.1

*Proof.* Let $\hat{\mathbf{x}}$ be an anomaly and $\mathbf{x}$ is a normal data point. Suppose their estimated densities are the same, i.e., $p_{\mathcal{X}}(\mathbf{x}) = p_{\mathcal{X}}(\hat{\mathbf{x}})$. In this situation, using merely the density will either detect both of them as normal or anomalous.

We compare their weighted log density:

$$
\begin{aligned}
& u^*(\mathbf{x}) - u^*(\hat{\mathbf{x}}) \\
= & -\sum_{i \in A} w_i^* F_{\mathcal{W}}^*(\mathbf{x})_i^2 - \sum_{j \in B} w_j^* F_{\mathcal{W}}^*(\mathbf{x})_j^2 + \log|\det \nabla_{\mathbf{x}} F_{\mathcal{W}}^*(\mathbf{x})| \\
& + \sum_{i \in A} w_i^* F_{\mathcal{W}}^*(\hat{\mathbf{x}})_i^2 + \sum_{j \in B} w_j^* F_{\mathcal{W}}^*(\hat{\mathbf{x}})_j^2 - \log|\det \nabla_{\hat{\mathbf{x}}} F_{\mathcal{W}}^*(\hat{\mathbf{x}})|
\end{aligned}
\tag{20}
$$

Since $p_{\mathcal{X}}(\mathbf{x}) = p_{\mathcal{X}}(\hat{\mathbf{x}})$, we have:

$$
\log|\det \nabla_{\mathbf{x}} F_{\mathcal{W}}^*(\mathbf{x})| - \log|\det \nabla_{\mathbf{x}} F_{\mathcal{W}}^*(\hat{\mathbf{x}})| = \frac{1}{2}\sum_{i=1}^{d}(F_{\mathcal{W}}^*(\mathbf{x})_i^2 - F_{\mathcal{W}}^*(\hat{\mathbf{x}})_i^2)
\tag{21}
$$

Thus, we have:

$$
\begin{aligned}
& u(\mathbf{x}) - u(\hat{\mathbf{x}}) \\
= & \sum_{i \in A}(1 - w_i^*)F_{\mathcal{W}}^*(\mathbf{x})_i^2 + \sum_{j \in B}(1 - w_j^*)F_{\mathcal{W}}^*(\mathbf{x})_j^2 \\
& - \sum_{i \in A}(1 - w_i^*)F_{\mathcal{W}}^*(\hat{\mathbf{x}})_i^2 - \sum_{j \in B}(1 - w_j^*)F_{\mathcal{W}}^*(\hat{\mathbf{x}})_j^2
\end{aligned}
\tag{22}
$$

$\forall j \in B, i \in A$, we have $F_{\mathcal{W}}^*(\mathbf{x})_j^2 > F_{\mathcal{W}}^*(\hat{\mathbf{x}})_j^2, F_{\mathcal{W}}^*(\mathbf{x})_i^2 < F_{\mathcal{W}}^*(\hat{\mathbf{x}})_i^2$ and $w_i^* > w_j^*$, this is because $\mathbf{x}$ and $\hat{\mathbf{x}}$ have the same probability but $\hat{\mathbf{x}}$ is an anomaly, thus $\mathbf{x}$ contains more noise.

If $\|F_{\mathcal{W}}^*(\mathbf{x})\|_2 = \|F_{\mathcal{W}}^*(\hat{\mathbf{x}})\|_2$, then we have:

$$
\begin{aligned}
\sum_{i=1}^{d} F_{\mathcal{W}}^*(\mathbf{x})_i^2 = \sum_{i=1}^{d} F_{\mathcal{W}}^*(\hat{\mathbf{x}})_i^2 \\
\sum_{j \in B}\left(F_{\mathcal{W}}^*(\mathbf{x})_j^2 - F_{\mathcal{W}}^*(\hat{\mathbf{x}})_j^2\right) = \sum_{i \in A}\left(F_{\mathcal{W}}^*(\hat{\mathbf{x}})_i^2 - F_{\mathcal{W}}^*(\mathbf{x})_i^2\right)
\end{aligned}
\tag{23}
$$

For $\forall j \in B, i \in A$, we have $w_i^* > w_j^* \Rightarrow \min_i w_i^* > \max_j w_j^*$. Then:

$$
\begin{aligned}
& u^*(\mathbf{x}) - u^*(\hat{\mathbf{x}}) \\
\geq & (1 - \max_{j \in B} w_j^*)\sum_{j \in B}\left(F_{\mathcal{W}}^*(\mathbf{x})_j^2 - F_{\mathcal{W}}^*(\hat{\mathbf{x}})_j^2\right) - (1 - \min_{i \in A} w_i^*)\sum_{i \in A}\left(F_{\mathcal{W}}^*(\hat{\mathbf{x}})_i^2 - F_{\mathcal{W}}^*(\mathbf{x})_i^2\right) \\
> & (1 - \min_{i \in A} w_i^*)\Big(\sum_{j \in B}\left(F_{\mathcal{W}}^*(\mathbf{x})_j^2 - F_{\mathcal{W}}^*(\hat{\mathbf{x}})_j^2\right) - \sum_{i \in A}\left(F_{\mathcal{W}}^*(\hat{\mathbf{x}})_i^2 - F_{\mathcal{W}}^*(\mathbf{x})_i^2\right)\Big) \\
= & 0
\end{aligned}
\tag{24}
$$

Thus, the two data points are distinguishable.
This finishes the proof. $\square$

## A.2    PROOF FOR THEOREM 3.2

**Lemma A.1.** *(Behrmann et al., 2021) Let $f_{\mathcal{W}_i}$ be a coupling flow, the Lipschitz constant of the forward $f_{\mathcal{W}_i}$ can be locally bounded for $\mathbf{x} \in [a, b]^d$ as:*

$$
Lip(f_{\mathcal{W}_i}) \leq \max(1, c_g) + M,
\tag{25}
$$

*Where $\exp(h_{i1}(\mathbf{x})) \leq c_g$ and $M = \max(|a|, |b|) \cdot c_{g'} \cdot Lip(h_{i1}) + Lip(h_{i2})$. Similarly, the Lipschitz constant of the reverse $f_{\mathcal{W}_i}^{-1}$ can be locally bounded for $\mathbf{y}_i \in [a^*, b^*]^d$ as:*

$$Lip(f_{\mathcal{W}_i}^{-1}) \leq \max(1, c_{\frac{1}{g}}) + M^*, \tag{26}$$

*Where $M^* = \max(|a^*|, |b^*|) \cdot c_{(\frac{1}{g}')} \cdot Lip(h_{i1}) \cdot c_t + c_{\frac{1}{g}} \cdot Lip(h_{i2})$*

*Proof.* According to LemmaA.1, here we can assume that $\mathbf{x}_i, \mathbf{y}_i$ are both bounded since data is preprocessed and normalized, then we have the bi-Lipschitz constant of $f_{\mathcal{W}_i}^*$ are bounded as:

$$\tau_i^- ||\mathbf{x}^{(i)} - \hat{\mathbf{x}}^{(i)}|| \leq ||f_{\mathcal{W}_i}^*(\mathbf{x}^{(i)}) - f_{\mathcal{W}_i}^*(\hat{\mathbf{x}}^{(i)})|| \leq \tau_i^+ ||\mathbf{x}^{(i)} - \hat{\mathbf{x}}^{(i)}||, \tag{27}$$

The determinant of $\nabla_{\mathbf{x}^{(i)}} f_{\mathcal{W}_i}^*(\mathbf{x}^{(i)})$ can be calculated as:

$$\log |\det \nabla_{\mathbf{x}^{(i)}} f_{\mathcal{W}_i}^*(\mathbf{x}^{(i)})| = h_{i1}^*(\mathbf{x}_\alpha^{(i)}) \cdot \mathbf{1}, \tag{28}$$

Suppose $h_{i1}^*(\mathbf{x}) = \mathbf{W}_{i,L}(\phi(\cdots \phi(\mathbf{W}_{i,2}\phi(\mathbf{W}_{i,1}\mathbf{x}))\cdots))$ and $h_{i2}^*(\mathbf{x}) = \hat{\mathbf{W}}_{i,L}\left(\phi\left(\cdots \phi\left(\hat{\mathbf{W}}_{i,2}\phi\left(\hat{\mathbf{W}}_{i,1}\mathbf{x}\right)\right)\cdots\right)\right)$ are two neural networks comprising $L$ layers and $\phi$ represents the activation function. Consider different $\mathbf{x}^{(i)}, \hat{\mathbf{x}}^{(i)}$, denote $\rho$ the Lipschitz constant of $\phi$, we have:

$$\begin{aligned}
&|\log |\det \nabla_{\mathbf{x}^{(i)}} f_{\mathcal{W}_i}^*(\mathbf{x}^{(i)})| - \log |\det \nabla_{\hat{\mathbf{x}}^{(i)}} f_{\mathcal{W}_i}^*(\hat{\mathbf{x}}^{(i)})||\\
=&\|h_{i1}^*(\mathbf{x}_\alpha^{(i)}) - h_{i1}^*(\hat{\mathbf{x}}_\alpha^{(i)})\|_1\\
\leq&\sqrt{d}\|h_{i1}^*(\mathbf{x}_\alpha^{(i)}) - h_{i1}^*(\hat{\mathbf{x}}_\alpha^{(i)})\|\\
\leq&\sqrt{d}\rho^{L-1}\prod_{l=1}^{L}\|\mathbf{W}_{i,l}\|_2\|\mathbf{x}_\alpha^{(i)} - \hat{\mathbf{x}}_\alpha^{(i)}\|\\
\leq&\sqrt{d}\rho^{L-1}\prod_{l=1}^{L}\|\mathbf{W}_{i,l}\|_2\|\mathbf{x}^{(i)} - \hat{\mathbf{x}}^{(i)}\|\\
=&\tau_{i\alpha}\sqrt{d}\|\mathbf{x}^{(i)} - \hat{\mathbf{x}}^{(i)}\|\\
\leq&\tau_{i\alpha}\sqrt{d}\prod_{j=1}^{i-1}\tau_j^+\|\mathbf{x} - \hat{\mathbf{x}}\|
\end{aligned} \tag{29}$$

Where $\tau_{i\alpha} = \rho^{L-1}\prod_{l=1}^{L}\|\mathbf{W}_{i,l}\|_2$ is the Lipschitz constant of $h_{i1}^*$. Then, we can conclude that $\log |\det(\nabla_x F_{\mathcal{W}}^*(\mathbf{x}))|$ has a Lipschitz constant:

$$\begin{aligned}
&|\log |\det(\nabla_{\mathbf{x}} F_{\mathcal{W}}^*(\mathbf{x}))| - \log |\det(\nabla_{\hat{\mathbf{x}}} F_{\mathcal{W}}^*(\hat{\mathbf{x}}))||\\
=&|\sum_{i=1}^{T}\left(\log |\det \nabla_{\mathbf{x}^{(i)}} f_{\mathcal{W}_i}^*(\mathbf{x}^{(i)})| - \log |\det \nabla_{\hat{\mathbf{x}}^{(i)}} f_{\mathcal{W}_i}^*(\hat{\mathbf{x}}^{(i)})|\right)|\\
\leq&\sum_{i=1}^{T}|\log |\det \nabla_{\mathbf{x}^{(i)}} f_{\mathcal{W}_i}^*(\mathbf{x}^{(i)})| - \log |\det \nabla_{\hat{\mathbf{x}}^{(i)}} f_{\mathcal{W}_i}^*(\hat{\mathbf{x}}^{(i)})||\\
\leq&\sum_{i=1}^{T}\tau_{i\alpha}\sqrt{d}\|\mathbf{x}^{(i)} - \hat{\mathbf{x}}^{(i)}\|\\
\leq&\sum_{i=1}^{T}(\tau_{i\alpha}\sqrt{d}\prod_{j=1}^{i-1}\tau_j^+)\|\mathbf{x} - \hat{\mathbf{x}}\|
\end{aligned} \tag{30}$$

The estimation error between $\log p^*_{\mathcal{X}}(\mathbf{x}_{\text{pure}})$ and $u^*(\mathbf{x})$ is:

$$
\begin{aligned}
&|\log p^*_{\mathcal{X}}(\mathbf{x}_{\text{pure}}) - u^*(\mathbf{x})| \\
=&|\sum_{i \in A}(1-w^*_i)F^*_{\mathcal{W}}(\mathbf{x})^2_i - \sum_{j \in B} w^*_j F^*_{\mathcal{W}}(\mathbf{x})^2_j - \log \frac{|\det \nabla_{\mathbf{x}} F^*_{\mathcal{W}}(\mathbf{x})|}{|\det \nabla_{\mathbf{x}_{\text{pure}}} F^*_{\mathbf{W}}(\mathbf{x}_{\text{pure}})|}| \\
\leq&|\sum_{i \in A}(1-w^*_i)F^*_{\mathcal{W}}(\mathbf{x})^2_i - \sum_{j \in B} w^*_j F^*_{\mathcal{W}}(\mathbf{x})^2_j + |\log \frac{|\det \nabla_{\mathbf{x}} F^*_{\mathcal{W}}(\mathbf{x})|}{|\det \nabla_{\mathbf{x}_{\text{pure}}} F^*_{\mathcal{W}}(\mathbf{x}_{\text{pure}})|}| \\
\leq&\sum_{i \in A}\left(1-w^*_i\right)F^*_{\mathcal{W}}(\mathbf{x})^2_i + \sum_{j \in B} w^*_j F^*_{\mathcal{W}}(\mathbf{x})^2_j + \sum_{i=1}^{T}(\tau_{i\alpha}\sqrt{d}\prod_{j=1}^{i-1}\tau^+_j)\|\mathbf{x} - \mathbf{x}_{\text{pure}}\|
\end{aligned}
\tag{31}
$$

As for the estimation error between $\log p^*_{\mathcal{X}}(\mathbf{x}_{\text{pure}})$ and $u(\mathbf{x})$, we have the following inequality holds:

$$
\begin{aligned}
|\log p^*_{\mathcal{X}}(\mathbf{x}_{\text{pure}}) - u(\mathbf{x})| &= |\log p^*_{\mathcal{X}}(\mathbf{x}_{\text{pure}}) - u^*(\mathbf{x}) + u^*(\mathbf{x}) - u(\mathbf{x})| \\
&\leq |\log p^*_{\mathcal{X}}(\mathbf{x}_{\text{pure}}) - u^*(\mathbf{x})| + |u^*(\mathbf{x}) - u(\mathbf{x})| \\
&\leq \sum_{i \in A}\left(1-w^*_i\right)F^*_{\mathcal{W}}(\mathbf{x})^2_i + \sum_{j \in B} w^*_j F^*_{\mathcal{W}}(\mathbf{x})^2_j \\
&\quad + \sum_{i=1}^{T}(\tau_{i\alpha}\sqrt{d}\prod_{j=1}^{i-1}\tau^+_j)\|\mathbf{x} - \mathbf{x}_{\text{pure}}\| + \eta
\end{aligned}
\tag{32}
$$

This finishes the proof. $\qquad\square$

### A.3 Proof for Theorem 3.4

*Proof.* By Assumption 3.3, we have that:

$$
\begin{aligned}
|u(\hat{\mathbf{x}}) - u(\mathbf{x}_a)| &\geq \varphi\|\mathbf{x}_a - \hat{\mathbf{x}}\| \\
u(\mathbf{x}_a) - u(\hat{\mathbf{x}}) &\geq \varphi\|\mathbf{x}_a - \hat{\mathbf{x}}\|
\end{aligned}
\tag{33}
$$

If $\|\hat{\mathbf{x}} - \mathbf{x}_a\| > \frac{\varsigma_1 - \varsigma_2}{\varphi}$, then we have:

$$
\begin{aligned}
u(\mathbf{x}_a) - u(\hat{\mathbf{x}}) &\geq \varsigma_1 - \varsigma_2 \\
u(\hat{\mathbf{x}}) &\leq u(\mathbf{x}_b)
\end{aligned}
\tag{34}
$$

Now we have the weighted log-density of $\hat{\mathbf{x}}$ is even smaller than the smallest weighted log-density of data from $\mathcal{D}$, thus it can be detected as an anomaly. This finishes the proof. $\qquad\square$

### A.4 Connection between $p_{\mathcal{X}}(\mathbf{x}_{\text{PURE}})$ and $p_{\bar{\mathcal{X}}}(\mathbf{x}_{\text{PURE}})$

The support of $\mathbf{x}_{\text{pure}}$ is an $m$-dimensional manifold $M$ embedded in $\mathbb{R}$. Let $g(\mathbf{z}_D) := F^{-1}(\mathbf{z}_D, \mathbf{0})$ The induced Riemannian metric on the manifold is given by:

$$
\mathbf{M}(\mathbf{z}_D) = J_g(\mathbf{z}_D)^\top J_g(\mathbf{z}_D)
\tag{35}
$$

where $J_g$ denote the Jacobian of $g$, i.e., $\nabla_{\mathbf{z}_D} g(\mathbf{z}_D)$. The volume element on the manifold, relative to the parameter space $\mathbf{s}_D$ is

$$
dV = \sqrt{\det\left[J_g(\mathbf{z}_D)^T J_g(\mathbf{z}_D)\right]}d\mathbf{z}_D.
\tag{36}
$$

The probability in the latent space is:

$$
\mathbb{P}(\mathbf{z}_D \in B) = \int_B p^D_{\mathcal{Z}}(\mathbf{z}_D)\,d\mathbf{z}_D.
\tag{37}
$$

This probability must equal the probability on the manifold $M$. For a measurable set $A \subset M$:

$$
\mathbb{P}(\mathbf{x}_{\text{pure}} \in A) = \int_{g^{-1}(A)} p^D_{\mathcal{Z}}(\mathbf{z}_D)\,d\mathbf{z}_D
\tag{38}
$$

We change the variable of integration from $\mathbf{z}_D$ to $\mathbf{x}_{\text{pure}} \in M$ and use the manifold volume element to obtain

$$d\mathbf{z}_D = \frac{d\mathcal{H}^m(\mathbf{x}_{\text{pure}})}{\sqrt{\det\left[J_g\left(\mathbf{z}_D\right)^\top J_g\left(\mathbf{z}_D\right)\right]}}, \tag{39}$$

where $d\mathcal{H}^m$ is the $m$-dimensional Hausdorff measure on $M$. Substituting this into the integral, we have

$$\mathbb{P}\left(\mathbf{x}_{\text{pure}} \in A\right) = \int_A \frac{p_{\mathcal{Z}}^D\left(\mathbf{z}_D\right)}{\sqrt{\det\left[J_g\left(\mathbf{z}_D\right)^\top J_g\left(\mathbf{z}_D\right)\right]}} d\mathcal{H}^m(\mathbf{x}) \tag{40}$$

Therefore, the probability density function on the manifold $M$ with respect to the Hausdorff measure is as follows

$$p_{\bar{\mathcal{X}}}\left(\mathbf{x}_{\text{pure}}\right) = \frac{p_{\mathcal{Z}}^D\left(\mathbf{z}_D\right)}{\sqrt{\det\left[J_g\left(\mathbf{z}_D\right)^\top J_g\left(\mathbf{z}_D\right)\right]}} = \frac{\mathcal{N}(\mathbf{z}_D; \mathbf{0}, \mathbf{I}_m)}{\sqrt{\det\left[J_g\left(\mathbf{z}_D\right)^\top J_g\left(\mathbf{z}_D\right)\right]}} \tag{41}$$

where the second equality used the fact that $\mathbf{s}_D$ and $\mathbf{s}_N$ are independent.

On the other hand, we have

$$\begin{aligned} p_{\mathcal{X}}\left(\mathbf{x}_{\text{pure}}\right) &= p_{\mathcal{Z}}\left(\mathbf{z}_D, \mathbf{0}\right)\left|\det J_F\left(\mathbf{x}_{\text{pure}}\right)\right| \\ &= c\mathcal{N}(\mathbf{z}_D; \mathbf{0}, \mathbf{I}_m)\left|\det J_F\left(\mathbf{x}_{\text{pure}}\right)\right| \end{aligned} \tag{42}$$

where $c = \frac{1}{2\pi^{(d-m)/2}}$. It follows that

$$\begin{aligned} p_{\mathcal{X}}\left(\mathbf{x}_{\text{pure}}\right) &= p_{\bar{\mathcal{X}}}\left(\mathbf{x}_{\text{pure}}\right) \times c\left|\det J_F\left(\mathbf{x}_{\text{pure}}\right)\right| \sqrt{\det\left[J_g\left(\mathbf{z}_D\right)^\top J_g\left(\mathbf{z}_D\right)\right]} \\ &= p_{\bar{\mathcal{X}}}\left(\mathbf{x}_{\text{pure}}\right) \times c\left|\det J_{F^{-1}}\left(\mathbf{z}_D, \mathbf{0}\right)\right|^{-1} \sqrt{\det\left[J_g\left(\mathbf{z}_D\right)^\top J_g\left(\mathbf{z}_D\right)\right]} \end{aligned} \tag{43}$$

## B    TIME COMPLEXITY OF DENSITY-BASED METHODS

Suppose that $F_{\mathcal{W}}$ is a sequence of $T$ flows defined in (2), and $h_{i1}, h_{i2}$ are two MLPs of $L$ layers parameterized by $\{\mathbf{W}_{i,j}\}_{j=1}^L$, $\{\hat{\mathbf{W}}_{i,j}\}_{j=1}^L$, where $\mathbf{W}_{i,j}, \hat{\mathbf{W}}_{i,j} \in \mathbb{R}^{d_{i,j} \times d_{i,j-1}}$, $j \in [L]$. Consider a batch of $B$ data points, the time complexity of our method per iteration is $\mathcal{O}(B \sum_{i=1}^T (d_{i,L} \sum_{j=0}^{L-2} d_{i,j} d_{i,j+1}))$, and the space complexity is $\mathcal{O}(B \sum_{i=1}^T \sum_{j=0}^L d_{i,j} d_{i,j+1}))$ which primarily arises from the computation of the Jacobian matrix. Here, we also compare the testing time complexity of a few representative density-based methods. We assume that DAGMM (Zong et al., 2018) contains $K$ Gaussians and the encoder and decoder have $\hat{L}$ layers, with $ith$ layer of encoder being $\mathbf{W}_{E,i} \in \mathbb{R}^{d_i \times d_{i-1}}$ and $ith$ layer of decoder being $\mathbf{W}_{D,i} \in \mathbb{R}^{d_{L+1-i} \times d_{L-i}}$. For DPAD (Fu et al., 2024), we assume that the size of its neural network is the same as that of the encoder of DAGMM. Suppose we have one testing data, the time complexity of density-based methods is shown in Table 3.

We notice that traditional density-based methods, such as KNN and KDE, require comparing test data against the entire training set to generate anomaly scores. Consequently, these methods become computationally inefficient as dataset sizes grow, since the time complexity grows linearly with the number of training data. DPAD encounters a similar issue due to its reliance on KNN, although it mitigates this by employing a neural network for dimensionality reduction. In contrast, methods like DAGMM, RealNVP, and our proposed NRDE primarily utilize neural network outputs for anomaly scoring, which do not depend on the training set.

## C    PROPERTY OF THE REGULARIZER $\mathcal{R}()$

Briefly speaking, our objective is to construct a Jacobian matrix in which the row norms exhibit a clear separation—some being significantly larger than others—so that we can distinguish between pure

Table 3: Time complexity comparison of density-based methods in testing stage.

| | Testing Complexity |
|---|---|
| KNN | $\mathcal{O}(nd)$ |
| KDE | $\mathcal{O}(nd)$ |
| LOF | $\mathcal{O}(nd)$ |
| DAGMM | $\mathcal{O}\left(\sum_{l=1}^{\hat{L}} d_{l-1}d_l + d_{\hat{L}}^3\right)$ |
| DPAD | $\mathcal{O}(\sum_{l=1}^{\hat{L}} d_{l-1}d_l + d_{\hat{L}}n)$ |
| RealNVP | $\mathcal{O}\left(\sum_{i=1}^{T}(\sum_{j=0}^{L-1} d_{i,j}d_{i,j+1})\right)$ |
| NRDE | $\mathcal{O}\left(\sum_{i=1}^{T}(\sum_{j=0}^{L-1} d_{i,j}d_{i,j+1})\right)$ |

data sources and noise sources. Consider the derivative $\frac{\partial \mathcal{R}(Q)}{\partial Q_{i,j}} = \frac{sign(Q_{i,j})}{\sqrt{||Q_{i:}||}}$, where $||Q_{i:}||$ is $\ell_1$ norm. In this formulation, rows with larger norms receive a smaller penalty from the regularizer, whereas rows with smaller norms receive a larger penalty. This naturally encourages row-wise sparsity and separation. Moreover, unlike the conventional $\ell_{2,1}$ norm, where smaller entries in the same row receive smaller penalty, our regularizer $\mathcal{R}$ imposes the same penalty on all entries within a given row—avoiding vanishing penalty problem for small entries—thereby enhancing both separation and sparsity. Thus, in theory, the formulation is suitable for our task.

# D  IMPLEMENTATION DETAILS

To ensure a consistent network architecture for a fair comparison, we employ two MLPs with two linear layers, where LeakyReLU is used as the activation function. Note that the outputs of $h_{i1}, hi2$ are actually the split output from the same MLP. The detailed network architecture is shown in Table 5. Additionally, we use Adam as our optimizer and set the batch size to $2048$ for all experiments, while the training epoch is set to 100. Since the scale of the Jacobian norm in different datasets can be largely different, as shown in Figure 7, we use a simple hyperparameter tuning strategy for NRDE: (i) Fixing $\lambda = 0$, decrease learning rate from 0.01 to 0.001 until training becomes stable (i.e., no loss explosion); (ii) Then, based on (15), viewed as $\min_{\mathcal{W}} \mathcal{L}(\lambda, \mathcal{W})$, select $\lambda \in 1, 0.1, 0.01$ such that the regularization term $\lambda \mathcal{R}(\cdot)$ is on a comparable scale with $0.1 \cdot \mathcal{L}(0, \mathcal{W})$. A detailed algorithm for hyperparameter tuning is provided in Algorithm 2.

---

**Algorithm 1** Training and Testing Procedure of NRDE

---

**Training stage of NRDE:**
**Input:** $\mathcal{D} = \{\mathbf{x}_1, \mathbf{x}_2, \ldots, \mathbf{x}_n\}$, $\lambda > 0$, training epoch $B$
  **Output:** $F_{\mathcal{W}}, \{w_i\}_{i=1}^{d}$
  Initialize the parameters of flow network $\mathcal{W}$
  **for** $b = 1, \ldots, B$ **do**
    **for** each batch $\hat{\mathcal{D}}$ **do**
      Obtain the flow output $\{F_{\mathcal{W}}(\mathbf{x})\}_{\mathbf{x} \in \hat{\mathbf{D}}}$
      Update parameters $\mathcal{W}$ using (15)
    **end for**
  **end for**
**Testing stage of NRDE:**
**Input:** $\mathbf{x}_{\text{new}}, F_{\mathcal{W}}, \{w_i\}_{i=1}^{d}$
  **Output:** anomaly score: $u(\mathbf{x}_{\text{new}})$
  Obatain $\mathbf{z}_{\text{new}} = F_{\mathcal{W}}(\mathbf{x}_{\text{new}})$
  Obtain anomaly score $u(\mathbf{x}_{\text{new}})$ using (17)

---

Table 4: Statistics of 47 real-world datasets in ADBench.

| Data | # Samples | # Features | # Anomaly | % Anomaly | Category |
|---|---|---|---|---|---|
| ALOI | 49534 | 27 | 1508 | 3.04 | Image |
| annthyroid | 7200 | 6 | 534 | 7.42 | Healthcare |
| backdoor | 95329 | 196 | 2329 | 2.44 | Network |
| breastw | 683 | 9 | 239 | 34.99 | Healthcare |
| campaign | 41188 | 62 | 4640 | 11.27 | Finance |
| cardio | 1831 | 21 | 176 | 9.61 | Healthcare |
| Cardiotocography | 2114 | 21 | 466 | 22.04 | Healthcare |
| celeba | 202599 | 39 | 4547 | 2.24 | Image |
| census | 299285 | 500 | 18568 | 6.20 | Sociology |
| cover | 286048 | 10 | 2747 | 0.96 | Botany |
| donors | 619326 | 10 | 36710 | 5.93 | Sociology |
| fault | 1941 | 27 | 673 | 34.67 | Physical |
| fraud | 284807 | 29 | 492 | 0.17 | Finance |
| glass | 214 | 7 | 9 | 4.21 | Forensic |
| Hepatitis | 80 | 19 | 13 | 16.25 | Healthcare |
| http | 567498 | 3 | 2211 | 0.39 | Web |
| InternetAds | 1966 | 1555 | 368 | 18.72 | Image |
| Ionosphere | 351 | 32 | 126 | 35.90 | Oryctognosy |
| landsat | 6435 | 36 | 1333 | 20.71 | Astronautics |
| letter | 1600 | 32 | 100 | 6.25 | Image |
| Lymphography | 148 | 18 | 6 | 4.05 | Healthcare |
| magic.gamma | 19020 | 10 | 6688 | 35.16 | Physical |
| mammography | 11183 | 6 | 260 | 2.32 | Healthcare |
| mnist | 7603 | 100 | 700 | 9.21 | Image |
| musk | 3062 | 166 | 97 | 3.17 | Chemistry |
| optdigits | 5216 | 64 | 150 | 2.88 | Image |
| PageBlocks | 5393 | 10 | 510 | 9.46 | Document |
| pendigits | 6870 | 16 | 156 | 2.27 | Image |
| Pima | 768 | 8 | 268 | 34.90 | Healthcare |
| satellite | 6435 | 36 | 2036 | 31.64 | Astronautics |
| satimage-2 | 5803 | 36 | 71 | 1.22 | Astronautics |
| shuttle | 49097 | 9 | 3511 | 7.15 | Astronautics |
| skin | 245057 | 3 | 50859 | 20.75 | Image |
| smtp | 95156 | 3 | 30 | 0.03 | Web |
| SpamBase | 4207 | 57 | 1679 | 39.91 | Document |
| speech | 3686 | 400 | 61 | 1.65 | Linguistics |
| Stamps | 340 | 9 | 31 | 9.12 | Document |
| thyroid | 3772 | 6 | 93 | 2.47 | Healthcare |
| vertebral | 240 | 6 | 30 | 12.50 | Biology |
| vowels | 1456 | 12 | 50 | 3.43 | Linguistics |
| Waveform | 3443 | 21 | 100 | 2.90 | Physics |
| WBC | 223 | 9 | 10 | 4.48 | Healthcare |
| WDBC | 367 | 30 | 10 | 2.72 | Healthcare |
| Wilt | 4819 | 5 | 257 | 5.33 | Botany |
| wine | 129 | 13 | 10 | 7.75 | Chemistry |
| WPBC | 198 | 33 | 47 | 23.74 | Healthcare |
| yeast | 1484 | 8 | 507 | 34.16 | Biology |

# E  STATISTICS OF DATASETS

In our experiments, we evaluate the performance of 14 methods on 47 widely used real-world datasets spanning multiple domains, including healthcare, audio, language processing, and finance, in a popular benchmark for anomaly detection (Han et al., 2022). The statistics of these datasets are

Table 5: Network architecture

| Tabular |
| --- |
| Dimension_input=$2d$ |
| Dimension_firstlayer=$b$ |
| Linear($2d, b$), LeakyReLU() |
| Linear($b, 2d$) |

shown in Table 4. These datasets encompass a range of samples and features, from small to large, providing comprehensive metrics and evaluations for the methods.

---

**Algorithm 2** Hyperparameter tuning Strategy of NRDE

---

**Input:** $\mathcal{D} = \{\mathbf{x}_1, \mathbf{x}_2, \ldots, \mathbf{x}_n\}, F_{\mathcal{W}}$
  **Output:lr**$^*, \lambda^*$
  Initialize the parameters of flow network $\mathcal{W}$
  Obtain the flow output $\{F_{\mathcal{W}}(\mathbf{x})\}_{\mathbf{x} \in \mathcal{D}}$
  Obtain $\mathcal{L}_0 = \mathcal{L}(0, \mathcal{W})$ using (15)
  Obtain $\mathcal{R}\left(\frac{1}{n}\sum_{\mathbf{x} \in \mathcal{D}} |\nabla_{\mathbf{x}} F_{\mathcal{W}}(\mathbf{x})|\right)$
  **for lr** $\in \{10^{-2}, 5*10^{-3}, 10^{-3}\}$ **do**
    Set **lr**$^* = $ **lr**
    Initialize the parameters of flow network $\mathcal{W}$
    **for** $b = 1, \ldots, 10$ **do**
      Obtain the flow output $\{F_{\mathcal{W}}(\mathbf{x})\}_{\mathbf{x} \in \mathcal{D}}$
      Obtain loss $\mathcal{L}_b = \mathcal{L}(0, \mathcal{W})$ using (15)
      Update parameters $\mathcal{W}$ with step size **lr** using $\mathcal{L}_b$
      **if** $\mathcal{L}_b > \mathcal{L}_0$ **then**
        **lr**$^* = 10^{-3}$
      **end if**
    **end for**
    **if lr**$^* \neq 10^{-3}$ **then**
      break the loop
    **end if**
  **end for**
  Set $\lambda^* = 0.01$
  **for** $\lambda \in \{1, 0.1, 0.01\}$ **do**
    **if** $1 \leq \frac{0.1\mathcal{L}_0}{\lambda \mathcal{R}\left(\frac{1}{n}\sum_{\mathbf{x} \in \mathcal{D}} |\nabla_{\mathbf{x}} F_{\mathcal{W}}(\mathbf{x})|\right)}$ **then**
      $\lambda^* = \lambda$
    **end if**
  **end for**

---

# F ALGORITHM DETAILS

The detailed algorithm of our proposed NRDE is illustrated in Algorithm 1.

The synthetic data is generated using Algorithm 3. We primarily use Gaussian or uniform distributions to generate data, where the variances of the data sources are significantly larger than those of the noise sources. Specifically, $\mathcal{S}_D = \text{Unif}\left([-10, 50]^d\right), \mathcal{S}_N = \text{Unif}\left([-40, -20]^d\right), \hat{\mathcal{S}}_N = \text{Unif}\left([-10, 10]^d\right), \hat{\mathcal{S}}_D = \text{Unif}\left([10, 30]^d\right)$. Both the training and testing normal data are generated using the normal data generative process. For noisy normal data, the data sources are distributed according to $\mathcal{S}_D$, while the noise sources are distributed according to variables distributed in $\mathbf{S}_N$ and are perturbed by $\hat{\mathbf{S}}_N$, introducing anomalies in the noise sources. In the case of anomalous data, the generative process closely resembles that of noisy normal data, where the noise sources are distributed in $\mathcal{S}_N$. However, the data sources are perturbed by variables distributed in $\mathbf{S}_D$ and perturbed by $\hat{\mathbf{S}}_D$, leading to anomalies in the data sources. Moreover, $\text{Var}(\hat{\mathbf{S}}_N) < \text{Var}(\hat{\mathbf{S}}_D)$. Data

---

**Algorithm 3** Data Generative Process

---

**Normal data generation:**

**Input:** data source distribution $\mathcal{S}_D$, noise source distribution $\mathcal{S}_N$, number of data sources $m$, data dimension $d$, mixing matrix $W \in \mathbb{R}^{d \times d}$

    **Output:** normal data $\mathbf{x}_i$

    Obtain data sources $\mathbf{s}_D \in \mathbb{R}^m$ by $\mathbf{s}_D \sim \mathcal{S}_D$, obtain noise sources $\mathbf{s}_N \in \mathbb{R}^{d-m}$ by $\mathbf{s}_N \sim \mathcal{S}_N$

    Generate data using $\mathbf{x} = W[\mathbf{s}_D; \mathbf{s}_N]^T$

**Noisy normal data generation:**

**Input:** number of data $\hat{n}$, data source distribution $\mathcal{S}_D$, noise source distribution $\mathcal{S}_N$, noise perturbation distribution $\hat{\mathcal{S}}_N$, number of data sources $m$, data dimension $d$, mixing matrix $W \in \mathbb{R}^{d \times d}$

    **Output:** noisy normal data $\tilde{\mathbf{x}}$

    Obtain data sources $\mathbf{s}_D \in \mathbb{R}^m$ by $\mathbf{s}_D \sim \mathcal{S}_D$, obtain noise sources $\mathbf{s}_N \in \mathbb{R}^{d-m}$ by $\mathbf{s}_N \sim \mathcal{S}_N$, obtain noise perturbation $\hat{\mathbf{s}}_N \in \mathbb{R}^{d-m}$ by $\hat{\mathbf{s}}_N \sim \hat{\mathcal{S}}_N$

    Generate data using $\tilde{\mathbf{x}} = W[\mathbf{s}_D; \mathbf{s}_N + \hat{\mathbf{s}}_N]^T$

**Anomalous data generation:**

**Input:** number of data $\hat{n}$, data source distribution $\mathcal{S}_D$, data perturbation distribution $\hat{\mathcal{S}}_D$, noise source distribution $\mathcal{S}_N$, number of data sources $m$, data dimension $d$, mixing matrix $W \in \mathbb{R}^{d \times d}$

    **Output:** anomalous data $\hat{\mathbf{x}}$

    Obtain data sources $\mathbf{s}_D \in \mathbb{R}^m$ by $\mathbf{s}_D \sim \mathcal{S}_D$, obtain data perturbation $\hat{\mathbf{s}}_D \in \mathbb{R}^m$ by $\hat{\mathbf{s}}_D \sim \hat{\mathcal{S}}_D$, obtain noise sources $\mathbf{s}_N \in \mathbb{R}^{d-m}$ by $\mathbf{s}_N \sim \mathcal{S}_N$

    Generate data using $\hat{\mathbf{x}} = W[\mathbf{s}_D + \hat{\mathbf{s}}_D; \mathbf{s}_N]^T$

---

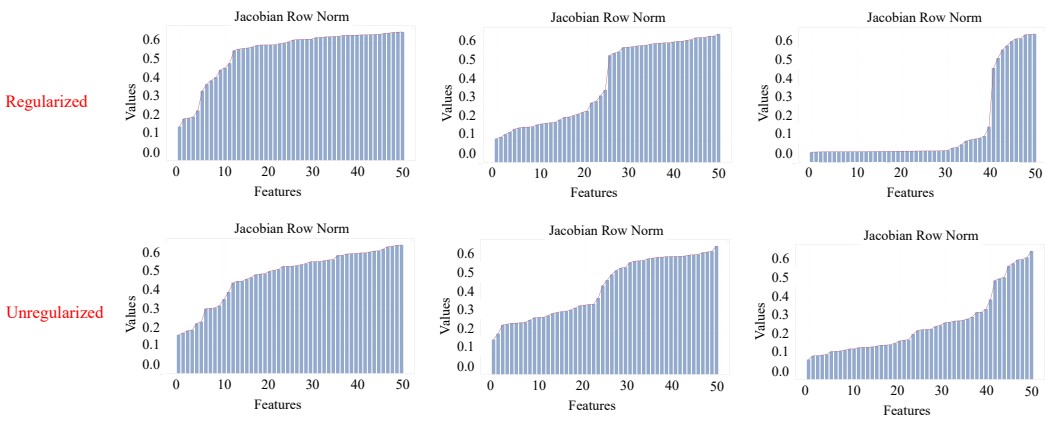

Figure 6: Visualization of the Jacobian matrix row norms on several synthetic datasets with 50 total sources. From left to right, the number of noise dimensions is 40, 25, and 10, respectively. The top row corresponds to the regularized case, whereas the bottom row illustrates the unregularized case.

shown in Figure 1 is generated using $d = 100, m = 10$. Note that here $W$ is an orthogonal matrix. All models are trained using 10000 normal data, and tested using 1000 normal data, 1000 noisy normal data, and 1000 anomalous data.

# G  EXPERIMENTAL RESULTS

## G.1  JACOBIAN ROW NORM VISUALIZATION

In this subsection, we present visualizations of the Jacobian row norms on both synthetic and real-world datasets. Figures 6 and 7 illustrate these results. Notably, even without regularization, the row norms already exhibit clear separability; this distinction becomes even more pronounced when the regularizer is applied.

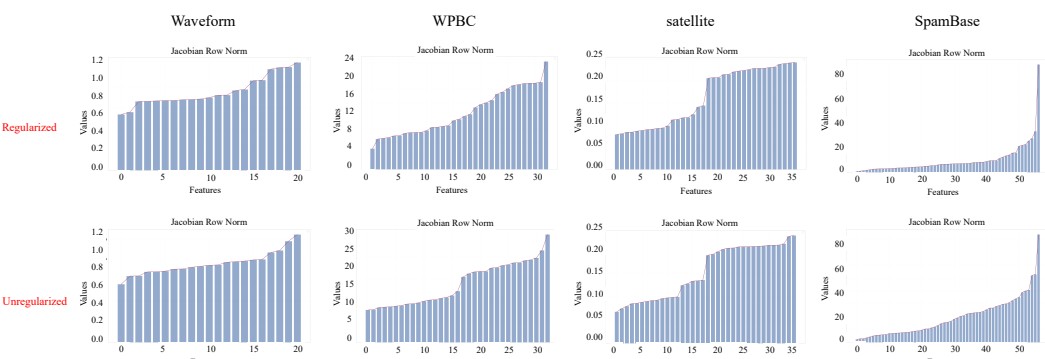

Figure 7: Visualization of the Jacobian matrix row norms on several real datasets. The top row corresponds to the regularized case, whereas the bottom row illustrates the unregularized case.

## G.2 STANDARD ANOMALY DETECTION

In this subsection, we provide the detailed experimental results of AD on 47 real-world datasets. Table 6 and Table 7 show the detailed AUROC and AUPRC results on 47 datasets.

Table 6: Average AUROC (%) with the standard deviation of each method on 47 tabular datasets of ADBench. The best results are marked in **bold**.

| Methods | KDE | KNN | LOF | OC-SVM | IF | AE | DSVDD | RealNVP | NeutralAD | ECOD | ICL | SLAD | DPAD | MCM | DTE-C | Ours |
|---|---|---|---|---|---|---|---|---|---|---|---|---|---|---|---|---|
| ALOI | 56.4 ± 0.0 | 63.4 ± 0.0 | **74.6 ± 0.0** | 55.0 ± 0.0 | 54.4 ± 0.1 | 55.8 ± 0.1 | 51.8 ± 0.2 | 56.2 ± 2.5 | 55.1 ± 0.9 | 53.0 ± 0.0 | 50.2 ± 2.5 | 54.8 ± 0.4 | 51.7 ± 0.6 | 63.2 ± 0.2 | 54.2 ± 0.0 | 56.7 ± 0.4 |
| annthyroid | 91.4 ± 0.0 | 94.1 ± 0.0 | 92.9 ± 0.0 | 90.9 ± 0.0 | 91.8 ± 1.1 | 83.4 ± 2.0 | 79.4 ± 3.2 | 96.1 ± 0.5 | 78.9 ± 2.8 | 78.7 ± 0.0 | 64.0 ± 6.1 | 90.4 ± 2.9 | 91.2 ± 4.7 | 83.9 ± 0.6 | 97.8 ± 0.0 | **98.4 ± 0.0** |
| backdoor | 90.5 ± 0.0 | 93.7 ± 0.0 | **95.7 ± 0.0** | 62.6 ± 0.0 | 75.1 ± 3.2 | 93.5 ± 0.3 | 92.5 ± 1.0 | 91.8 ± 0.4 | 87.1 ± 4.7 | 84.6 ± 0.0 | 95.2 ± 1.9 | 92.5 ± 0.2 | 94.6 ± 0.7 | 96.6 ± 0.1 | 91.8 ± 0.0 | 94.5 ± 2.0 |
| breastw | 98.9 ± 0.0 | 99.1 ± 0.0 | 96.7 ± 0.0 | 99.0 ± 0.0 | **99.5 ± 0.0** | 98.4 ± 0.3 | 99.1 ± 0.1 | 98.0 ± 0.0 | 81.4 ± 3.9 | 99.3 ± 0.0 | 90.2 ± 1.3 | 99.2 ± 0.1 | 98.9 ± 0.2 | 99.0 ± 0.0 | 96.3 ± 0.0 | 99.4 ± 0.0 |
| campaign | 77.3 ± 0.0 | 78.3 ± 0.0 | 69.8 ± 0.0 | 77.1 ± 0.0 | 72.0 ± 1.5 | 80.8 ± 0.7 | 1.1 ± 0.5 | 79.6 ± 0.3 | 62.3 ± 2.6 | 78.3 ± 0.0 | **80.5 ± 1.4** | 76.3 ± 0.6 | 64.2 ± 6.1 | 78.5 ± 0.2 | 76.7 ± 0.0 | 76.9 ± 0.1 |
| cardio | 95.7 ± 0.0 | 93.4 ± 0.0 | 93.0 ± 0.0 | 96.4 ± 0.0 | 94.9 ± 1.1 | 92.4 ± 3.2 | **96.1 ± 0.3** | 94.1 ± 0.4 | 81.0 ± 1.9 | 93.4 ± 0.0 | 83.9 ± 1.5 | 88.7 ± 3.0 | 90.0 ± 3.3 | 90.4 ± 0.8 | 93.6 ± 0.0 | 95.8 ± 0.6 |
| Cardiotocography | 75.0 ± 0.0 | 71.3 ± 0.0 | 72.7 ± 0.0 | 80.7 ± 0.0 | 79.3 ± 2.5 | 73.4 ± 2.4 | 83.4 ± 2.5 | 77.9 ± 1.8 | 58.2 ± 1.9 | 78.5 ± 0.0 | 54.7 ± 11.9 | 58.4 ± 2.1 | 68.0 ± 3.1 | 70.0 ± 0.9 | 72.4 ± 0.0 | **86.1 ± 2.4** |
| celeba | 70.5 ± 0.0 | 68.0 ± 0.0 | 44.9 ± 0.0 | 79.0 ± 0.0 | 70.3 ± 0.7 | 70.9 ± 1.0 | 48.4 ± 0.8 | 79.4 ± 0.6 | 66.4 ± 9.4 | 75.7 ± 0.0 | 69.5 ± 0.0 | 65.2 ± 2.1 | 56.3 ± 6.1 | 56.3 ± 3.3 | 82.7 ± 0.0 | **87.9 ± 0.7** |
| census | 72.0 ± 0.0 | 71.9 ± 0.0 | 60.5 ± 0.0 | 70.2 ± 0.0 | 62.7 ± 1.9 | 71.8 ± 0.1 | 51.9 ± 3.1 | 72.8 ± 0.3 | 72.9 ± 2.6 | 65.9 ± 0.0 | 66.8 ± 0.0 | 68.9 ± 0.6 | 50.7 ± 0.8 | 68.1 ± 0.2 | 69.6 ± 0.0 | **76.7 ± 3.0** |
| cover | 95.5 ± 0.0 | **98.5 ± 0.0** | 98.9 ± 0.0 | 96.2 ± 0.0 | 14.0 ± 1.8 | 98.3 ± 0.6 | 47.6 ± 1.6 | 83.8 ± 0.1 | 85.1 ± 3.8 | 92.0 ± 1.1 | 67.4 ± 3.4 | 79.2 ± 9.9 | 87.9 ± 5.9 | 96.4 ± 0.4 | 96.7 ± 0.0 | 84.1 ± 3.2 |
| donors | 97.4 ± 0.0 | **99.8 ± 0.0** | 98.2 ± 0.0 | 92.0 ± 0.0 | 88.4 ± 1.6 | 93.6 ± 2.1 | 97.7 ± 0.7 | 96.6 ± 0.4 | 95.8 ± 1.7 | 88.8 ± 0.0 | 94.0 ± 0.0 | 96.9 ± 0.6 | 98.7 ± 1.9 | **100.0 ± 0.0** | 98.9 ± 0.0 | 97.6 ± 1.2 |
| fault | **81.2 ± 0.0** | 76.9 ± 0.0 | 67.2 ± 0.0 | 61.4 ± 0.0 | 65.2 ± 0.9 | 73.3 ± 0.4 | 71.9 ± 1.1 | 50.8 ± 0.1 | 73.2 ± 0.3 | 47.4 ± 0.0 | 77.9 ± 1.0 | 79.2 ± 0.3 | 73.5 ± 2.5 | 71.2 ± 0.2 | 71.8 ± 0.0 | 62.7 ± 4.1 |
| fraud | 95.8 ± 0.0 | **96.0 ± 0.0** | 78.3 ± 0.0 | 95.6 ± 0.0 | 95.0 ± 0.2 | 95.7 ± 0.2 | 50.6 ± 12.1 | 54.4 ± 0.1 | 94.9 ± 3.0 | 94.9 ± 0.0 | 83.4 ± 0.0 | 94.6 ± 0.3 | 64.1 ± 13.9 | 95.8 ± 0.2 | 94.4 ± 0.0 | 95.9 ± 0.3 |
| glass | 83.5 ± 0.0 | 89.3 ± 0.0 | 74.0 ± 0.0 | 71.1 ± 0.0 | 81.3 ± 1.0 | 77.9 ± 2.3 | 79.8 ± 2.7 | 81.5 ± 1.3 | **92.1 ± 2.3** | 69.8 ± 0.0 | 90.8 ± 1.1 | 83.5 ± 1.0 | 88.8 ± 2.0 | 86.5 ± 0.1 | 78.4 ± 0.0 | 91.2 ± 0.9 |
| Hepatitis | 79.4 ± 0.0 | 85.0 ± 0.0 | 84.6 ± 0.0 | 84.2 ± 0.0 | 77.8 ± 1.9 | 83.9 ± 1.4 | 80.3 ± 3.2 | 59.4 ± 0.8 | 62.2 ± 5.8 | 71.5 ± 0.0 | 60.1 ± 4.1 | 77.6 ± 3.0 | 83.7 ± 2.5 | 81.2 ± 1.5 | 80.8 ± 0.0 | **85.3 ± 1.7** |
| http | **100.0 ± 0.0** | 99.9 ± 0.0 | 93.0 ± 0.0 | 100.0 ± 0.0 | 99.1 ± 0.3 | 99.8 ± 0.1 | 99.8 ± 0.1 | 99.6 ± 0.0 | 99.0 ± 1.9 | 97.8 ± 0.0 | 50.0 ± 0.0 | **99.9 ± 0.0** | 99.8 ± 0.2 | 99.9 ± 0.1 | 99.5 ± 0.0 | 99.8 ± 0.0 |
| InternetAds | 85.7 ± 0.0 | 73.8 ± 0.0 | 78.7 ± 0.0 | 73.8 ± 0.0 | 43.6 ± 3.2 | **88.2 ± 0.0** | 52.4 ± 1.1 | 81.2 ± 0.3 | 81.6 ± 0.9 | 68.9 ± 0.0 | 78.9 ± 1.0 | 86.7 ± 0.3 | 83.2 ± 0.6 | 82.3 ± 0.1 | 87.1 ± 0.0 | 79.1 ± 0.7 |
| Ionosphere | 97.4 ± 0.0 | **97.0 ± 0.0** | 94.7 ± 0.0 | 96.5 ± 0.0 | 93.6 ± 0.5 | 94.3 ± 1.0 | 78.7 ± 3.4 | 93.8 ± 0.2 | 96.4 ± 0.5 | 75.5 ± 0.0 | 94.1 ± 0.4 | 86.3 ± 0.2 | 97.3 ± 0.4 | 95.4 ± 0.1 | 95.4 ± 0.0 | 87.7 ± 2.0 |
| landsat | 72.7 ± 0.0 | 73.9 ± 0.0 | **75.4 ± 0.0** | 45.7 ± 0.0 | 59.9 ± 1.5 | 57.0 ± 1.8 | 58.1 ± 4.4 | 59.7 ± 2.0 | 70.7 ± 1.4 | 36.6 ± 0.0 | 73.8 ± 2.8 | 69.9 ± 0.1 | 69.7 ± 3.2 | 62.2 ± 0.1 | 58.8 ± 0.0 | 71.5 ± 4.1 |
| letter | 91.8 ± 0.0 | 84.1 ± 0.0 | 86.1 ± 0.0 | 60.9 ± 0.0 | 61.7 ± 2.0 | 80.1 ± 0.6 | 34.2 ± 2.3 | 83.1 ± 0.6 | **92.5 ± 0.8** | 56.0 ± 0.0 | 87.5 ± 1.4 | 90.3 ± 0.6 | 80.2 ± 5.5 | 89.0 ± 0.4 | 89.6 ± 0.0 | 70.2 ± 0.0 |
| Lymphography | 98.6 ± 0.0 | 98.6 ± 0.0 | 98.6 ± 0.0 | 98.4 ± 0.0 | 97.7 ± 0.5 | 98.5 ± 0.1 | 98.5 ± 0.2 | 94.3 ± 0.5 | 82.9 ± 3.3 | **98.5 ± 0.0** | 92.9 ± 5.1 | 98.5 ± 0.2 | 98.3 ± 0.2 | 98.5 ± 0.1 | 97.7 ± 0.0 | **98.7 ± 0.3** |
| magic.gamma | 75.7 ± 0.0 | 82.2 ± 0.0 | **83.2 ± 0.0** | 73.5 ± 0.0 | 77.3 ± 0.5 | 81.8 ± 0.8 | 76.1 ± 1.6 | 79.6 ± 0.5 | 77.5 ± 0.6 | 63.4 ± 0.0 | 71.7 ± 1.7 | 72.4 ± 1.2 | 79.8 ± 2.9 | 82.6 ± 0.3 | 85.8 ± 0.0 | 81.9 ± 1.1 |
| mammography | 88.1 ± 0.0 | 87.6 ± 0.0 | 83.8 ± 0.0 | 88.9 ± 0.0 | 88.3 ± 0.8 | 87.5 ± 2.3 | 81.3 ± 1.4 | 89.5 ± 0.4 | 70.9 ± 4.6 | 90.7 ± 0.0 | 57.3 ± 10.7 | 72.6 ± 3.2 | 84.8 ± 2.2 | 90.7 ± 0.4 | 87.8 ± 0.0 | **91.2 ± 0.3** |
| mnist | 94.8 ± 0.0 | 93.8 ± 0.0 | 92.6 ± 0.0 | 91.0 ± 0.0 | 86.6 ± 1.1 | 93.4 ± 0.2 | 85.1 ± 1.7 | 92.4 ± 0.4 | 77.8 ± 2.8 | 74.7 ± 0.0 | 86.7 ± 1.8 | 91.2 ± 0.6 | 85.8 ± 3.1 | 90.3 ± 0.4 | 84.9 ± 0.0 | 93.2 ± 4.3 |
| musk | 100.0 ± 0.0 | 100.0 ± 0.0 | 1.0 ± 0.0 | 100.0 ± 0.0 | 95.8 ± 3.3 | 1.0 ± 0.0 | 99.9 ± 0.1 | 99.4 ± 0.3 | 100.0 ± 0.0 | 95.8 ± 0.0 | 100.0 ± 0.0 | **100.0 ± 0.0** | 99.9 ± 0.9 | **100.0 ± 0.0** | **100.0 ± 0.0** | 99.8 ± 0.2 |
| optdigits | 97.4 ± 0.0 | 92.7 ± 0.0 | 97.8 ± 0.0 | 60.9 ± 0.0 | 79.6 ± 4.8 | 88.7 ± 0.8 | 32.6 ± 13.0 | 94.2 ± 0.7 | **95.9 ± 0.5** | 60.4 ± 0.0 | 91.7 ± 2.5 | 91.4 ± 1.6 | 75.5 ± 10.8 | 87.8 ± 2.8 | 89.2 ± 0.0 | 94.5 ± 1.4 |
| PageBlocks | 95.0 ± 0.0 | **95.8 ± 0.0** | 96.7 ± 0.0 | 94.4 ± 0.0 | 92.8 ± 0.1 | 94.8 ± 0.6 | 94.7 ± 0.5 | 92.0 ± 0.9 | 93.7 ± 0.5 | 91.4 ± 0.0 | 79.5 ± 4.1 | 87.7 ± 0.0 | 95.4 ± 1.5 | 96.3 ± 0.2 | 96.2 ± 0.0 | 92.1 ± 2.2 |
| pendigits | 99.8 ± 0.0 | **99.8 ± 0.0** | 98.8 ± 0.0 | 96.1 ± 0.0 | 97.0 ± 0.5 | 98.0 ± 0.2 | 88.6 ± 4.5 | 98.5 ± 0.6 | 93.9 ± 1.5 | 92.7 ± 0.0 | 93.8 ± 4.6 | 92.7 ± 2.9 | 94.7 ± 2.8 | 98.6 ± 0.4 | 97.6 ± 0.0 | 98.1 ± 0.7 |
| Pima | 78.1 ± 0.0 | 78.0 ± 0.0 | 73.6 ± 0.0 | 73.1 ± 0.0 | 76.5 ± 0.5 | 73.9 ± 1.7 | 73.0 ± 2.6 | 79.3 ± 0.6 | 56.0 ± 2.6 | 61.5 ± 0.0 | 63.1 ± 3.6 | 54.6 ± 2.1 | 70.2 ± 3.4 | 74.1 ± 1.2 | 70.7 ± 0.0 | **81.7 ± 0.7** |
| satellite | 86.9 ± 0.0 | **87.3 ± 0.0** | 85.1 ± 0.0 | 75.4 ± 0.0 | 80.0 ± 1.2 | 80.3 ± 0.4 | 81.7 ± 2.9 | 83.6 ± 1.3 | 78.4 ± 2.5 | 58.3 ± 0.0 | 83.9 ± 1.8 | 87.0 ± 0.4 | 86.5 ± 1.6 | 82.0 ± 0.1 | 86.1 ± 0.0 | 86.8 ± 0.6 |
| satimage-2 | 99.9 ± 0.0 | 99.9 ± 0.0 | 99.6 ± 0.0 | 99.7 ± 0.0 | 99.4 ± 0.0 | 99.8 ± 0.1 | 98.5 ± 0.6 | 98.9 ± 0.0 | 84.9 ± 1.4 | 96.6 ± 0.0 | 96.2 ± 1.7 | **99.7 ± 0.0** | 98.7 ± 1.3 | 98.9 ± 0.2 | 98.9 ± 0.0 | 99.7 ± 0.1 |
| shuttle | 99.8 ± 0.0 | 99.9 ± 0.0 | 99.9 ± 0.0 | 99.7 ± 0.0 | 99.6 ± 0.1 | 99.8 ± 0.1 | 99.2 ± 0.1 | 99.8 ± 0.0 | 99.9 ± 0.0 | 99.3 ± 0.0 | 99.4 ± 0.4 | 99.9 ± 0.0 | 99.9 ± 178.7 | 100.0 ± 0.0 | 99.7 ± 0.0 | **99.9 ± 0.0** |
| skin | 89.1 ± 0.0 | **99.8 ± 0.0** | 92.5 ± 0.0 | 90.3 ± 0.0 | 88.8 ± 0.4 | 83.9 ± 2.3 | 69.6 ± 1.6 | 90.0 ± 0.1 | 84.4 ± 1.1 | 48.8 ± 0.0 | 50.0 ± 0.0 | 91.2 ± 1.8 | 99.3 ± 0.2 | 79.4 ± 3.3 | 92.4 ± 0.0 | 92.5 ± 0.4 |
| smtp | 88.2 ± 0.0 | 93.5 ± 0.0 | 94.2 ± 0.0 | 85.5 ± 0.0 | 90.1 ± 0.2 | 91.4 ± 3.4 | 89.5 ± 1.2 | 93.4 ± 0.1 | 91.0 ± 2.4 | 87.9 ± 0.0 | 53.0 ± 6.0 | 91.9 ± 3.2 | 93.4 ± 1.1 | 83.5 ± 3.2 | 95.2 ± 0.0 | **95.6 ± 0.1** |
| SpamBase | 85.7 ± 0.0 | 83.0 ± 0.0 | 81.7 ± 0.0 | 81.6 ± 0.0 | 83.6 ± 1.4 | 82.0 ± 0.1 | 79.4 ± 1.7 | 80.1 ± 0.3 | 79.2 ± 0.8 | 66.0 ± 0.0 | 81.4 ± 0.9 | 85.3 ± 0.8 | 76.8 ± 3.8 | 81.3 ± 0.3 | 84.5 ± 0.0 | **87.4 ± 0.5** |
| speech | 45.8 ± 0.0 | 48.5 ± 0.0 | 48.9 ± 0.0 | 45.9 ± 0.0 | 46.7 ± 1.2 | 46.8 ± 0.2 | 45.2 ± 1.2 | 50.0 ± 0.0 | 54.3 ± 4.2 | 46.1 ± 0.0 | 49.1 ± 2.9 | 50.7 ± 3.2 | 54.8 ± 4.6 | 49.9 ± 0.3 | 56.1 ± 0.0 | **64.7 ± 1.9** |
| Stamps | 95.1 ± 0.0 | 90.8 ± 0.0 | 87.2 ± 0.0 | 91.2 ± 0.0 | 91.9 ± 0.5 | 89.2 ± 1.3 | 91.9 ± 0.9 | 93.6 ± 1.4 | 74.2 ± 1.8 | 86.7 ± 0.0 | 88.1 ± 4.4 | 73.0 ± 5.2 | 90.9 ± 2.9 | 88.6 ± 1.3 | 67.5 ± 0.0 | 95.9 ± 1.5 |
| thyroid | 98.3 ± 0.0 | 98.5 ± 0.0 | 94.6 ± 0.0 | 98.2 ± 0.0 | 99.0 ± 0.2 | 98.0 ± 0.3 | 97.5 ± 0.5 | 98.6 ± 0.1 | 65.2 ± 7.9 | 97.7 ± 0.0 | 82.2 ± 5.2 | 94.8 ± 1.8 | 96.1 ± 1.8 | 97.9 ± 0.3 | 99.2 ± 0.0 | **99.2 ± 0.1** |
| vertebral | 43.5 ± 0.0 | 42.5 ± 0.0 | 40.0 ± 0.0 | 52.7 ± 0.0 | 42.6 ± 4.5 | 48.0 ± 4.3 | 43.7 ± 4.5 | 53.6 ± 4.8 | 53.9 ± 3.0 | 41.8 ± 0.0 | 54.2 ± 5.8 | 44.4 ± 4.3 | 46.4 ± 3.5 | 47.2 ± 1.4 | 59.2 ± 0.0 | **72.7 ± 6.0** |
| vowels | 96.5 ± 0.0 | 97.3 ± 0.0 | 96.8 ± 0.0 | 83.0 ± 0.0 | 77.7 ± 1.9 | 95.3 ± 0.9 | 41.3 ± 9.1 | 90.4 ± 1.1 | **98.7 ± 0.1** | 59.5 ± 0.0 | 98.2 ± 0.4 | 97.2 ± 0.6 | 93.4 ± 2.5 | 91.5 ± 1.6 | 97.3 ± 0.0 | 87.8 ± 4.3 |
| Waveform | 76.0 ± 0.0 | 76.2 ± 0.0 | 76.6 ± 0.0 | 69.0 ± 0.0 | 72.5 ± 1.4 | 65.8 ± 2.5 | 69.6 ± 3.8 | 72.5 ± 1.6 | 71.5 ± 0.5 | 60.0 ± 0.0 | 59.8 ± 1.1 | 92.5 ± 4.0 | 61.0 ± 3.5 | 69.6 ± 1.2 | 65.6 ± 0.0 | **91.6 ± 1.1** |
| WBC | 98.1 ± 0.0 | 99.4 ± 0.0 | 97.9 ± 0.0 | 99.0 ± 0.0 | **99.7 ± 0.1** | 99.0 ± 0.3 | 99.3 ± 0.4 | 98.8 ± 0.2 | 78.6 ± 4.7 | 99.0 ± 0.0 | 80.0 ± 6.7 | 98.8 ± 0.8 | 98.3 ± 0.7 | 99.1 ± 0.3 | 98.2 ± 0.0 | 99.9 ± 0.1 |
| WDBC | 99.4 ± 0.0 | 99.1 ± 0.0 | 99.4 ± 0.0 | 99.3 ± 0.0 | 99.3 ± 0.3 | 99.4 ± 0.3 | 98.9 ± 1.1 | 95.0 ± 0.3 | 32.5 ± 6.3 | 97.8 ± 0.0 | 83.7 ± 9.5 | 97.8 ± 0.3 | 97.2 ± 2.6 | 97.9 ± 0.2 | 38.9 ± 0.0 | **99.9 ± 0.0** |
| Wilt | 37.1 ± 0.0 | 60.8 ± 0.0 | 70.8 ± 0.0 | 33.9 ± 0.0 | 50.4 ± 1.6 | 56.2 ± 7.5 | 49.9 ± 2.3 | 59.9 ± 0.5 | **80.3 ± 3.9** | 40.3 ± 0.0 | 78.2 ± 3.7 | 66.8 ± 6.7 | 73.1 ± 2.0 | 66.0 ± 5.2 | 86.8 ± 0.0 | 77.9 ± 2.5 |
| wine | 92.2 ± 0.0 | 93.2 ± 0.0 | 92.2 ± 0.0 | 91.2 ± 0.0 | 88.5 ± 1.4 | 85.6 ± 2.7 | 90.2 ± 3.9 | 92.6 ± 1.6 | 78.5 ± 5.2 | 73.0 ± 0.0 | 82.6 ± 8.2 | 92.7 ± 4.5 | 85.3 ± 4.1 | 95.8 ± 1.2 | 92.3 ± 0.0 | **99.1 ± 1.0** |
| WPBC | 52.5 ± 0.0 | 51.3 ± 0.0 | 50.5 ± 0.0 | 49.1 ± 0.0 | 52.5 ± 0.6 | 49.6 ± 0.6 | 50.0 ± 1.8 | 58.0 ± 1.0 | 59.1 ± 2.1 | 47.0 ± 0.0 | 53.4 ± 5.9 | 50.4 ± 0.8 | 52.1 ± 2.5 | 52.3 ± 0.8 | 48.3 ± 0.0 | **65.3 ± 4.1** |
| yeast | 43.2 ± 0.0 | 46.6 ± 0.0 | 46.7 ± 0.0 | 44.9 ± 0.0 | 42.9 ± 0.7 | 47.9 ± 1.3 | 42.5 ± 1.6 | 51.0 ± 1.4 | 60.1 ± 1.8 | 45.3 ± 0.0 | 55.9 ± 2.1 | 52.7 ± 0.8 | 51.8 ± 1.9 | 45.7 ± 0.7 | 50.5 ± 0.0 | **61.1 ± 2.7** |
| AVG | 84.3 | 85.1 | 81.0 | 79.8 | 78.2 | 81.3 | 74.3 | 83.6 | 77.9 | 73.7 | 76.1 | 79.2 | 81.5 | 83.4 | 83.3 | **86.8** |

## G.3 ANOMALY DETECTION WITH ANOMALY CONTAMINATION

Under this experimental setting, we conduct experiments on five datasets: Cardiotocography, Satellite, SpamBase, Pima, and WPBC. The average AUPRC results are shown in Figure 8. Detailed results for each dataset are shown in Figure 9. As the anomaly ratio increases, the performance variation of our proposed method remains minimal, demonstrating its robustness to anomalous data in the training set. We observe that when the anomaly ratio increases, the performance of some methods does not decrease or even improves. The reason for this may be that, as the anomaly ratio increases, the number of anomalies in the test set decreases, leading to different test sets for experiments at varying anomaly ratios.

Table 7: Average AUPRC (%) with the standard deviation of each method on 47 tabular datasets of ADBench. The best results are marked in **bold**.

| Methods | KDE | KNN | LOF | OC-SVM | IF | AE | DSVDD | RealNVP | NeutralAD | ECOD | ICL | SLAD | DPAD | MCM | DTE-C | Ours |
|---|---|---|---|---|---|---|---|---|---|---|---|---|---|---|---|---|
| ALOI | 10.5±0.0 | 9.8±0.0 | **15.9±0.0** | 7.5±0.3 | 6.6±0.0 | 7.6±0.0 | 7.2±0.2 | 7.1±0.0 | 7.5±0.3 | 6.4±0.0 | 6.1±0.5 | 7.1±0.0 | 7.1±0.4 | 10.9±0.2 | 6.8±0.0 | 7.8±0.3 |
| annthyroid | 66.2±0.0 | 72.0±0.0 | 66.7±0.0 | 65.2±0.0 | 63.8±2.8 | 60.7±2.1 | 54.8±2.2 | 77.0±2.8 | 29.4±4.0 | 40.8±0.0 | 31.3±9.5 | 63.1±5.8 | 64.5±9.5 | 55.0±0.6 | **84.1±0.0** | 79.7±2.0 |
| backdoor | 44.7±0.0 | 45.0±0.0 | 59.9±0.0 | 7.8±0.0 | 9.6±1.8 | 86.8±0.1 | 71.4±1.1 | 77.9±5.8 | 55.8±2.3 | 16.7±0.0 | 89.6±1.2 | **86.0±0.6** | 65.1±3.0 | 81.8±0.3 | 63.2±0.0 | 83.5±1.4 |
| breastw | 98.8±0.0 | 99.1±0.0 | 93.7±0.0 | 98.8±0.0 | **99.5±0.0** | 98.1±0.5 | 99.1±0.1 | 96.7±0.1 | 71.2±3.0 | 99.3±0.0 | 86.3±3.0 | 99.2±0.2 | 98.7±0.2 | 99.0±0.1 | 92.1±0.0 | 99.4±0.0 |
| campaign | 47.7±0.0 | 48.8±0.0 | 39.6±0.0 | 48.9±0.0 | 43.7±1.8 | 49.2±0.8 | 42.5±0.6 | **50.3±0.1** | 28.9±3.2 | 50.0±0.0 | 49.5±1.3 | 48.4±0.4 | 33.0±6.5 | 50.0±0.2 | 48.7±0.0 | 49.4±1.6 |
| cardio | 84.0±0.0 | 76.8±0.0 | 69.3±0.0 | 82.8±0.0 | 78.4±4.0 | 74.7±5.9 | **83.0±0.9** | 71.0±2.6 | 48.9±4.3 | 70.9±0.0 | 60.7±3.2 | 72.7±3.0 | 73.5±6.6 | 73.1±1.0 | 69.5±0.0 | 75.9±4.3 |
| Cardiotocography | 68.1±0.0 | 62.4±0.0 | 59.9±0.0 | 71.0±0.0 | 67.6±2.4 | 65.0±2.5 | **75.1±2.5** | 62.6±2.1 | 40.3±2.3 | 65.7±0.0 | 45.4±10.1 | 54.7±1.5 | 61.5±2.6 | 61.3±1.0 | 61.1±0.0 | 74.3±2.7 |
| celeba | 8.9±0.0 | 9.8±0.0 | 3.7±0.0 | 20.4±0.0 | 12.5±0.7 | 9.5±0.2 | 4.0±0.1 | 13.1±0.6 | 6.6±1.5 | 17.2±0.0 | 8.9±0.0 | **76.1±0.4** | 5.8±1.3 | 7.3±1.0 | 15.7±0.0 | 20.1±1.6 |
| census | 21.6±0.0 | 21.2±0.0 | 14.3±0.0 | 20.5±0.0 | 14.2±0.8 | 21.6±0.1 | 11.9±0.8 | 20.5±0.6 | 23.3±0.0 | 15.5±0.0 | 17.4±0.0 | 19.8±0.2 | 12.1±0.5 | 18.8±0.2 | 18.0±0.0 | **24.7±1.6** |
| cover | 34.2±0.0 | 72.0±0.0 | **83.7±0.0** | 22.0±0.0 | 1.1±0.0 | 52.8±9.6 | 2.0±0.1 | 9.3±1.8 | 29.1±14.8 | 18.4±0.0 | 9.0±3.7 | 9.2±7.8 | 37.4±12.7 | 60.7±1.7 | 67.9±0.0 | 12.8±2.4 |
| donors | 70.9±0.0 | 95.3±0.0 | 76.3±0.0 | 42.4±0.0 | 37.6±3.4 | 49.8±8.2 | 82.3±3.4 | 66.2±1.1 | 64.3±0.0 | 41.2±0.0 | 88.5±2.3 | 65.6±4.9 | **96.6±3.5** | 99.7±0.0 | 77.8±0.0 | 75.0±5.1 |
| fault | **79.8±0.0** | 76.0±0.0 | 64.0±0.0 | 65.2±0.0 | 63.9±1.0 | 72.9±0.3 | 69.3±2.2 | 51.5±0.0 | 70.5±0.5 | 49.4±0.0 | 75.4±1.2 | 78.5±0.6 | 72.5±2.4 | 69.4±0.3 | 72.2±0.0 | 63.6±3.1 |
| fraud | 33.8±0.0 | 31.3±0.0 | 1.1±0.0 | 31.7±0.0 | 21.2±4.8 | 60.5±3.2 | 11.5±13.6 | 3.8±0.0 | 58.8±2.0 | 31.5±0.0 | 30.3±0.0 | 33.3±3.6 | 13.5±11.8 | **80.1±0.4** | 72.8±0.0 | 26.1±0.5 |
| glass | 27.9±0.0 | 29.9±0.0 | 18.8±0.0 | 19.2±0.0 | 22.6±1.8 | 20.8±0.2 | 21.2±1.2 | 24.5±1.4 | **49.7±1.9** | 25.1±0.0 | 38.7±6.8 | 26.1±0.0 | 33.3±3.6 | 26.1±0.2 | 23.4±0.0 | 31.8±1.5 |
| Hepatitis | 59.7±0.0 | 62.1±0.0 | 62.0±0.0 | 61.6±0.0 | 45.9±2.4 | 59.8±2.8 | 54.4±3.3 | 32.0±0.4 | 34.5±4.0 | 40.2±0.0 | 34.3±3.1 | 53.4±6.0 | 59.6±3.6 | 52.4±1.6 | 55.9±0.0 | **69.0±1.0** |
| http | 99.2±0.0 | 99.9±0.0 | 9.6±0.0 | 99.5±0.0 | 45.3±7.7 | 86.8±15.2 | **99.7±0.1** | 55.1±0.0 | 43.3±0.3 | 25.2±0.0 | 92.9±3.4 | 97.4±1.1 | 87.5±10.7 | 57.7±0.0 | 65.2±1.1 |
| InternetAds | 80.7±0.0 | 65.3±0.0 | 67.3±0.0 | 64.4±0.0 | 27.2±1.6 | **86.1±0.0** | 42.3±1.2 | 56.5±0.5 | 69.6±2.3 | 62.8±0.0 | 62.2±1.8 | 79.6±0.9 | 77.8±1.7 | 73.8±0.1 | 78.4±0.0 | 64.2±1.3 |
| Ionosphere | 97.9±0.0 | **98.3±0.0** | 95.2±0.0 | 97.3±0.0 | 94.0±0.4 | 95.5±0.1 | 80.6±3.5 | 90.9±0.3 | 96.9±0.5 | 77.9±0.0 | 94.4±0.9 | 97.1±0.1 | 97.7±0.4 | 96.6±0.1 | 96.1±0.0 | 90.8±1.9 |
| landsat | 54.8±0.0 | 58.0±0.0 | **70.4±0.0** | 32.1±0.0 | 43.2±2.4 | 40.0±1.5 | 42.5±5.0 | 46.4±0.4 | 49.4±1.3 | 27.8±0.0 | 68.9±2.8 | 48.3±1.2 | 50.6±3.8 | 44.0±0.2 | 39.4±0.0 | 55.6±8.4 |
| letter | 59.9±0.0 | 44.4±0.0 | 49.5±0.0 | 20.7±0.0 | 15.5±0.9 | 36.2±1.3 | 8.6±0.4 | 41.2±3.4 | 69.1±2.4 | 13.8±0.0 | 52.1±2.5 | 57.2±2.6 | 40.6±8.6 | 46.2±0.6 | **64.9±0.0** | 21.0±2.8 |
| Lymphography | 80.0±0.0 | 80.0±0.0 | 80.0±0.0 | 72.0±0.0 | 80.5±3.3 | 76.6±4.0 | 80.0±5.2 | 42.9±2.1 | 28.7±1.2 | 89.7±0.0 | 66.6±22.6 | 77.9±5.9 | 75.6±4.2 | 77.2±3.9 | 65.0±0.0 | **80.2±2.1** |
| magic.gamma | 80.4±0.0 | 85.4±0.0 | 86.2±0.0 | 78.5±0.0 | 80.2±0.3 | 84.9±0.6 | 80.3±1.3 | 83.2±0.0 | 80.5±0.3 | 67.6±0.0 | 77.6±1.4 | 77.9±1.2 | 84.0±1.9 | 86.4±0.2 | **87.9±0.0** | 84.7±0.8 |
| mammography | 43.7±0.0 | 41.9±0.0 | 32.7±0.0 | 41.2±0.0 | 38.8±5.0 | 37.2±9.0 | 34.9±2.7 | 44.8±2.2 | 10.0±0.7 | **54.0±0.0** | 15.9±5.3 | 15.4±3.6 | 36.9±5.6 | 41.2±1.7 | 39.3±0.0 | 46.0±2.6 |
| mnist | **78.7±0.0** | 77.1±0.0 | 72.3±0.0 | 69.4±0.0 | 55.8±2.4 | 74.0±0.3 | 58.1±1.8 | 67.1±1.2 | 49.7±2.0 | 29.9±0.0 | 63.0±2.6 | 74.0±0.1 | 64.8±4.7 | 69.1±1.0 | 58.7±0.0 | 76.7±0.3 |
| musk | 100.0±0.0 | 100.0±0.0 | 99.0±0.0 | 100.0±0.0 | 60.8±22.5 | 99.9±0.0 | 99.2±1.5 | 84.9±6.7 | 100.0±0.0 | 63.2±0.0 | 100.0±0.0 | **100.0±0.0** | 99.9±0.1 | 100.0±0.0 | 100.0±0.0 | 98.2±2.1 |
| optdigits | 49.7±0.0 | 33.6±0.0 | 5.3±0.0 | 6.5±0.0 | 14.1±2.6 | 19.8±1.1 | 4.0±0.8 | 40.5±3.6 | 55.5±6.0 | 7.0±0.0 | 34.0±7.1 | 26.0±3.4 | 16.7±7.9 | 19.9±2.8 | 22.4±0.0 | **57.4±1.3** |
| PageBlocks | 84.8±0.0 | 86.6±0.0 | **87.9±0.0** | 80.0±0.0 | 70.2±0.5 | 82.6±1.7 | 84.4±0.7 | 74.5±1.8 | 78.2±1.1 | 66.4±0.0 | 68.0±5.1 | 71.2±1.4 | 86.9±1.8 | 85.5±0.5 | 84.9±0.0 | 74.3±3.6 |
| pendigits | 96.7±0.0 | **95.9±0.0** | 69.7±0.0 | 47.4±0.0 | 48.9±3.7 | 56.7±3.1 | 27.5±9.1 | 68.0±8.1 | 30.2±2.8 | 38.5±0.0 | 48.3±10.4 | 32.6±4.5 | 58.1±16.3 | 66.6±5.8 | 45.9±0.0 | 79.5±8.1 |
| Pima | 77.0±0.0 | 76.9±0.0 | 73.0±0.0 | 74.3±0.0 | 77.0±0.4 | 73.9±2.0 | 73.4±2.2 | 77.6±0.8 | 56.8±3.2 | 65.7±0.0 | 63.0±4.1 | 58.7±1.9 | 71.0±2.9 | 74.1±1.4 | 70.2±0.0 | **79.7±0.6** |
| satellite | 89.2±0.0 | **89.3±0.0** | 88.5±0.0 | 82.3±0.0 | 84.3±0.6 | 85.8±0.1 | 84.4±2.5 | 86.8±1.1 | 74.5±2.2 | 65.7±0.0 | 87.2±1.3 | 87.8±0.3 | 87.6±1.4 | 85.7±0.1 | 87.7±0.0 | 88.0±0.6 |
| satimage-2 | 98.3±0.0 | 97.9±0.0 | **99.6±0.0** | 97.4±0.0 | 93.9±1.0 | 94.2±2.5 | 93.7±2.2 | 57.2±2.5 | 7.3±0.5 | 77.3±0.0 | 82.5±4.4 | 91.8±3.0 | 89.0±5.8 | 61.0±3.1 | 52.4±0.0 | 96.3±1.1 |
| shuttle | 98.1±0.0 | 97.5±0.0 | 99.4±0.0 | 97.5±0.0 | 98.5±0.5 | 97.3±0.0 | 97.0±0.1 | 98.1±0.2 | **99.9±0.3** | 94.3±0.0 | 98.3±0.5 | 97.5±0.0 | 99.2±0.1 | 99.2±0.1 | 94.0±0.0 | 98.7±0.1 |
| skin | 65.0±0.0 | **99.5±0.0** | 73.0±0.0 | 66.3±0.0 | 63.5±0.9 | 64.0±6.4 | 48.6±0.7 | 65.7±2.5 | 67.4±2.4 | 30.3±0.0 | 34.4±0.0 | 80.3±5.0 | 98.3±0.5 | 63.3±8.5 | 70.7±0.0 | 73.3±2.4 |
| smtp | 58.8±0.0 | 42.0±0.0 | 29.3±0.0 | 60.5±0.0 | 0.9±0.0 | 28.9±12.4 | 16.0±6.8 | 32.0±2.0 | **62.8±4.3** | 52.6±0.0 | 6.1±12.0 | 52.1±9.0 | 58.5±5.6 | 43.7±0.1 | 44.1±0.0 | 51.5±0.0 |
| SpamBase | 87.7±0.0 | 86.6±0.0 | 82.9±0.0 | 84.9±0.0 | 87.3±1.2 | 84.8±0.1 | 80.8±0.9 | 80.5±0.4 | 68.9±0.0 | 86.3±0.8 | **88.9±0.4** | 82.3±2.9 | 84.7±0.3 | 86.2±0.0 | 88.7±0.4 | |
| speech | 3.7±0.0 | 3.7±0.0 | 4.5±0.0 | 3.6±0.0 | 3.5±0.2 | 3.6±0.4 | 3.0±0.2 | 3.2±0.0 | 4.0±0.0 | 3.8±0.0 | 3.3±0.2 | 3.8±0.5 | 4.4±1.1 | 4.4±0.2 | 4.9±0.0 | **5.3±0.7** |
| Stamps | 63.7±0.0 | 54.2±0.0 | 44.2±0.0 | 51.0±0.0 | 50.9±1.3 | 47.4±2.4 | 52.8±2.6 | 61.8±6.1 | 26.8±7.1 | 45.2±0.0 | 56.2±10.8 | 33.6±4.3 | 52.8±6.1 | 19.9±2.8 | 22.4±0.0 | **72.7±8.3** |
| thyroid | 73.8±0.0 | 77.4±0.0 | 58.8±0.0 | 73.9±0.0 | 83.7±1.6 | 78.3±5.5 | 78.9±2.1 | 76.4±1.9 | 6.2±3.2 | 62.9±0.0 | 28.8±12.6 | 67.6±7.7 | 60.6±5.1 | 71.9±2.8 | 86.4±0.0 | **86.8±1.4** |
| vertebral | 19.7±0.0 | 20.3±0.0 | 19.6±0.0 | 23.1±0.0 | 19.4±1.6 | 21.6±2.7 | 20.1±1.7 | 25.3±2.2 | 29.8±1.4 | 19.5±0.0 | 26.2±4.0 | 21.4±3.3 | 21.2±1.3 | 20.9±0.1 | 27.1±0.0 | **41.0±6.6** |
| vowels | 77.7±0.0 | 76.3±0.0 | 74.3±0.0 | 44.2±0.0 | 25.4±3.0 | 69.9±6.3 | 6.0±1.2 | 63.6±3.6 | **87.6±2.0** | 14.2±0.0 | 84.5±2.2 | 76.6±6.9 | 64.1±11.0 | 56.4±3.3 | 79.6±0.0 | 51.0±4.5 |
| Waveform | 27.6±0.0 | 27.0±0.0 | 31.7±0.0 | 10.7±0.0 | 10.8±0.5 | 11.1±1.3 | 9.5±1.3 | 11.3±0.6 | 47.4±2.5 | 7.6±0.0 | 29.6±2.2 | 5.7±1.0 | 12.0±2.1 | 20.0±0.8 | 10.3±0.0 | **34.8±3.5** |
| WBC | 85.5±0.0 | 95.7±0.0 | 82.3±0.0 | 91.2±0.0 | 97.8±0.7 | 92.1±3.7 | 94.3±2.9 | 83.2±4.1 | 26.7±2.2 | 99.0±0.0 | 24.3±6.3 | 91.7±5.4 | 86.0±6.8 | 93.7±1.5 | 77.2±0.0 | **99.2±1.0** |
| WDBC | 90.9±0.0 | 81.5±0.0 | 89.9±0.0 | 87.7±0.0 | 86.5±6.2 | 90.0±6.4 | 80.8±15.5 | 36.0±1.4 | 4.0±0.3 | 73.9±0.0 | 34.3±23.2 | 70.3±2.8 | 76.4±11.5 | 66.8±2.4 | 17.1±0.0 | **99.4±1.1** |
| Wilt | 7.4±0.0 | 12.9±0.0 | 17.0±0.0 | 7.0±0.0 | 9.2±0.3 | 10.8±2.2 | 10.1±0.4 | 11.3±0.1 | **51.9±8.0** | 8.1±0.0 | 38.6±4.3 | 17.8±4.3 | 17.3±0.9 | 13.7±2.0 | 29.5±0.0 | 19.4±1.9 |
| wine | 58.2±0.0 | 60.8±0.0 | 52.7±0.0 | 55.0±0.0 | 57.3±2.6 | 42.4±6.0 | 56.5±11.4 | 56.6±7.1 | 33.8±1.5 | 30.5±0.0 | 44.4±14.6 | 64.0±18.8 | 39.8±6.6 | 80.2±6.5 | 59.6±0.0 | **95.2±4.0** |
| WPBC | 38.3±0.0 | 37.5±0.0 | 37.5±0.0 | 36.9±0.0 | 37.8±0.3 | 37.1±0.0 | 37.2±1.1 | 43.8±1.0 | 50.6±1.6 | 35.4±0.0 | 41.3±4.5 | 39±0.9 | 39.2±1.7 | 39.0±0.7 | 36.5±0.0 | **52.5±4.6** |
| yeast | 48.2±0.0 | 49.5±0.0 | 49.8±0.0 | 48.6±0.0 | 48.10.4 | 40.3±0.4 | 47.0±1.1 | 51.7±0.6 | 57.5±1.5 | 50.0±0.0 | 55.0±2.0 | 53.1±0.7 | 52.3±0.9 | 49.0±0.8 | 51.2±0.0 | **58.8±2.3** |
| AVG | 62.3 | 62.2 | 57.6 | 54.6 | 48.9 | 58.2 | 51.5 | 53.6 | 48.0 | 45.0 | 49.9 | 58.4 | 58.2 | 59.9 | 57.0 | **63.3** |

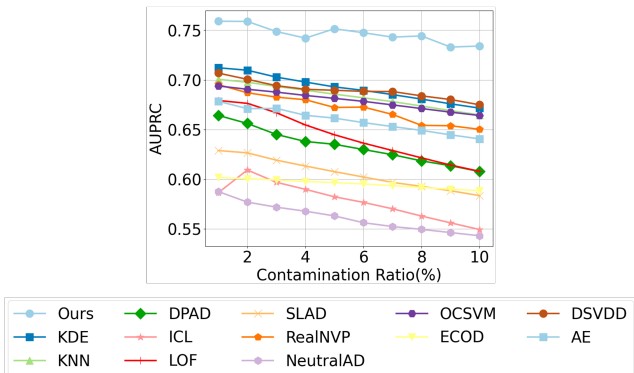

| Ours | DPAD | SLAD | OCSVM | DSVDD |
|---|---|---|---|---|
| KDE | ICL | RealNVP | ECOD | AE |
| KNN | LOF | NeutralAD | | |

Figure 8: The average AUPRC performance of 13 methods across 5 datasets of AD experiments with anomaly contamination, with anomaly ratio ranging from 1% to 10%.

## G.4 OUTLIER DETECTION

To evaluate the effectiveness of our method in outlier detection (transductive learning), we conduct experiments on several datasets where all data are used for both training and testing, and compare our method with other outlier detection methods. We provide the detailed experimental results for outlier detection on 5 datasets: Cardiotocography, Satellite, SpamBase, Pima, and WPBC. We compare our proposed method with traditional density-based methods and state-of-the-art outlier detection methods. The AUROC and AUPRC results are shown in Table 8.

## G.5 ABLATION STUDIES

In this subsection, we investigate how each component of our proposed method affects its anomaly detection performance and analyze the impact of different values of the hyperparameter $\lambda$ on detection performance. The two main components of our method are the regularizer $\mathcal{R}$ and the weighted log density $u(\mathbf{x})$. Notably, when these two components are ablated, the method reduces to a basic

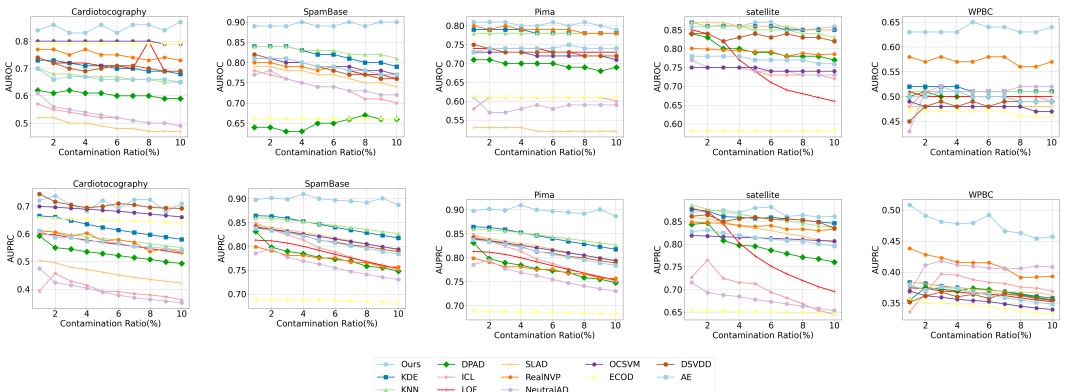

Figure 9: The detailed AUROC and AUPRC performance of 13 methods across 5 datasets of AD experiments with anomaly contamination, with anomaly ratio ranging from $1\%$ to $10\%$.

Table 8: Detailed AUROC performance of outlier detection on 5 datasets. The best results are marked in **bold**.

| AUROC | Cardiotocography | SpamBase | Satellite | Pima | WPBC |
|---|---|---|---|---|---|
| IF | 68.8 | 65.5 | 67.9 | 67.0 | 49.1 |
| ECOD | 78.5 | 65.5 | 58.2 | 59.4 | 48.1 |
| OC-SVM | 69.5 | 53.3 | 66.3 | 62.3 | 48.4 |
| KPCA | 53.4 | 52.1 | 48.2 | 53.8 | 45.5 |
| LOF | 52.3 | 45.6 | 54.1 | 60.1 | 52.0 |
| KDE | 50.2 | 49.5 | 76.0 | 72.2 | 49.9 |
| $k$NN | 57.9 | 52.9 | 65.0 | 65.1 | 47.2 |
| RealNVP | 62.7 | 56.5 | 74.6 | 70.7 | 59.1 |
| COPOD | 66.2 | 68.7 | 63.3 | 65.4 | 52.3 |
| DeepIF | 63.0 | 37.9 | 74.3 | 61.3 | 49.4 |
| Ours | **80.5** | **77.7** | **81.9** | **80.1** | **62.2** |

normalizing flow, i.e., RealNVP. Table 9 and 10 present the average performance results of different components across five datasets. We observe that both components contribute positively to overall performance. Specifically, the regularizer $\mathcal{R}$ primarily enhances the separability of sources, thus exerting minimal influence on the basic normalizing flow.

We also investigate the impact of different values of the hyperparameter $\lambda$ and learning rate on the performance of standard AD. The detailed experimental results are shown in Table 11 and 12. We observe that the method is not highly sensitive to changes in $\lambda$ and learning rate; however, in some datasets, large values of $\lambda$ may affect the training process and lead to a decrease in performance.

### G.6 MORE EXPERIMENTS ON SYNTHETIC AND REAL DATASETS TO VERIFY ASSUMPTIONS AND MOTIVATIONS

In this subsection, we include several experiments on both synthetic and real datasets to further verify our assumptions and motivations.

#### G.6.1 PERFORMANCE RESULTS WHEN VARIANCE DIFFERENCE IS NOT SATISFIED

Here, we analyze the performance of NRDE on synthetic datasets where the variance difference is not satisfied. Suppose the variance of pure data sources is $\sigma_d^2$, and the variance of noise sources is $\sigma_n^2$, we now report the performance results on synthetic datasets with different $\frac{\sigma_d^2}{\sigma_n^2}$ in Table 13. The performance decline of NRDE verifies our assumptions and motivations.

Table 9: Average AUROC and AUPRC performance of the proposed method containing different components.

| AUROC | Cardiotocography | SpamBase | Satellite | Pima | WPBC |
|---|---|---|---|---|---|
| **w/o** $u$, $\mathcal{R}$ | 77.9 | 80.1 | 83.6 | 79.3 | 58.0 |
| **w/o** $u$ | 77.2 | 80.7 | 82.3 | 78.2 | 60.2 |
| **w/o** $\mathcal{R}$ | 84.6 | 84.7 | 84.1 | 80.8 | 61.9 |
| Ours | **86.1** | **87.4** | **86.8** | **81.7** | **65.3** |
| AUPRC | Cardiotocography | SpamBase | Satellite | Pima | WPBC |
| **w/o** $u$, $\mathcal{R}$ | 62.6 | 80.5 | 86.8 | 77.6 | 43.8 |
| **w/o** $u$ | 60.9 | 80.8 | 85.1 | 76.5 | 45.4 |
| **w/o** $\mathcal{R}$ | 73.2 | 86.8 | 85.0 | 79.1 | 47.8 |
| Ours | **74.3** | **88.7** | **88.0** | **81.7** | **52.5** |

Table 10: Average AUROC and AUPRC performance of the proposed method containing different components across 5 datasets.

| Components | AUROC | AUPRC |
|---|---|---|
| **w/o** $u(\mathbf{x})$, $\mathcal{R}$ | 75.7 | 70.2 |
| **w/o** $u(\mathbf{x})$ | 75.7 | 69.7 |
| **w/o** $\mathcal{R}$ | 79.2 | 74.3 |
| Ours | **81.4** | **76.6** |

Table 11: Average AUROC and AUPRC performance of the proposed method with different values of learning rate $lr$.

| AUROC | Cardiotocography | SpamBase | Satellite | Pima | WPBC |
|---|---|---|---|---|---|
| $lr = 0.001$ | 62.9 | 82.9 | 82.4 | 81.1 | 59.3 |
| $lr = 0.005$ | 78.0 | 87.4 | 80.2 | 80.5 | 64.1 |
| $lr = 0.01$ | 86.1 | 86.3 | 83.5 | 81.0 | 62.9 |

Table 12: Average AUROC and AUPRC performance of the proposed method with different values of hyperparameter $\lambda$.

| AUROC | Cardiotocography | SpamBase | Satellite | Pima | WPBC |
|---|---|---|---|---|---|
| $\lambda = 0$ | 84.6 | 84.7 | 84.1 | 80.8 | 61.9 |
| $\lambda = 0.01$ | 85.1 | 86.3 | 83.5 | 81.0 | 62.9 |
| $\lambda = 0.1$ | 86.1 | 87.4 | 82.4 | 81.7 | 61.1 |
| $\lambda = 1$ | 79.4 | 82.0 | 86.8 | 80.3 | 65.3 |
| AUPRC | Cardiotocography | SpamBase | Satellite | Pima | WPBC |
| $\lambda = 0$ | 73.2 | 86.8 | 85.0 | 79.1 | 47.8 |
| $\lambda = 0.01$ | 74.3 | 86.9 | 85.2 | 79.2 | 50.9 |
| $\lambda = 0.1$ | 74.3 | 88.7 | 84.7 | 79.7 | 47.7 |
| $\lambda = 1$ | 67.1 | 83.8 | 88.0 | 79.0 | 52.5 |

Table 13: AUROC performance of NRDE on synthetic datasets with different $\frac{\sigma_d^2}{\sigma_n^2}$ ratios.

| $\sigma_d^2/\sigma_n^2$ | 9 | 6 | 4 | 2 | 1 | 0.5 |
|---|---|---|---|---|---|---|
| NRDE | 87.5 | 82.9 | 80.4 | 77.9 | 71.2 | 68.6 |

### G.6.2 PERFORMANCE RESULTS COMPARISON WITH IDEAL BASELINES.

In synthetic dataset where $m$ the number of data sources is known, we compare the performance of NRDE with KDE-C, DSVDD-C and KNN-C which are evaluated on datasets without noise components and NRDE$-m$, where only the $m$ sources with largest variance from set $A$ are used for computing anomaly score: $u_m(x_{new}) = \log|\det(\nabla_{x_{new}} F_\mathcal{W}(x_{new}))| - \frac{d}{2}\log 2\pi - \frac{1}{2}\sum_{i \in A} w_i F_\mathcal{W}(\mathbf{x}_{new})_i^2$. The results are shown in Table 14. Since NRDE is an approximation of NRDE$-m$, its performance being close but not as good as NRDE$-m$ and other ideal baselines supports our claim and motivation.

Table 14: AUROC (%) performance of NRDE and other baselines on the synthetic dataset.

| Method | NRDE | NRDE-m | KDE-C | KNN-C | DSVDD-C |
|--------|------|--------|-------|-------|---------|
| AUROC | 83.1 | 86.3 | 87.2 | 90.2 | 87.5 |

### G.6.3 EXPERIMENTAL RESULTS USING CONTRADICTORY ASSUMPTION

If we make a contradictory assumption that the variances of data sources should be smaller, then the weight for each source should be defined as:

$$w_i = \exp\left(||(\frac{1}{n}\sum_{\mathbf{x} \in \mathcal{D}} |\nabla_\mathbf{x} F_\mathcal{W}(\mathbf{x})|)_i||\right) / \sum_{j=1}^{d} \exp\left(||(\frac{1}{n}\sum_{\mathbf{x} \in \mathcal{D}} |\nabla_\mathbf{x} F_\mathcal{W}(\mathbf{x})|)_j||\right)$$

where sources with smaller variances obtain larger weights. This method is denoted as NRDE-CON. The performance of NRDE-CON and NRDE on several datasets is shown in Table 15, where the results support the assumption in our paper.

Table 15: AUROC (%) performance of NRDE-CON and NRDE.

| Method | WPBC | Thyroid | Musk | Annthyroid | Wilt |
|--------|------|---------|------|------------|------|
| NRDE-CON | 60.1 | 59.6 | 76.5 | 53.3 | 63.1 |
| NRDE | 65.3 | 99.2 | 99.8 | 98.4 | 77.9 |

## H MORE EXPERIMENTAL RESULTS DURING THE REBUTTAL PHASE

To facilitate the review process, this section consolidates all supplementary experiments conducted and added during the rebuttal phase.

### H.1 MORE EXPERIMENTAL COMPARISON BETWEEN $\lambda = 0$ AND $\lambda \neq 0$

In this subsection, we provide a performance comparison of NRDE ($\lambda \neq 0$)and NRDE ($\lambda = 0$) on 47 datasets in Table 16. Setting $\lambda \neq 0$ results in performance improvement in most datasets.

### H.2 HYPERPARAMETER CONFIGURATION OF NRDE ON 47 DATASETS

In this subsection, we provide the hyperparameter configuration of NRDE on all the 47 datasets in Table 17.

### H.3 DYNAMICS OF TRAINING LOSS AND AUROC ACROSS THE TRAINING PROCEDURE

In this subsection, we provide the dynamics of training loss and AUROC across the training procedure for several datasets in Figure 10. As shown in the figure, the decrease in loss is consistent with the improvement in performance.

### H.4    Experiments using symmetric design for anomaly detection on noisy data

In this subsection, we consider a symmetric design in which stronger noise is added to the anomalous test samples. We conduct experiments on the same five datasets, and the corresponding results are presented in Table 18. In this scenario, NRDE consistently outperforms competing methods, underscoring its robustness to noise.

### H.5    A practical diagnostic to verify variance difference on real data.

In this subsection, we provide a simple way to empirically verify the variance difference underlying our assumption is as follows. We first train an *unregularized* normalizing flow $F_{\mathcal{W}}$ on the real dataset $\mathcal{D}$ and compute the average absolute Jacobian

$$J = \frac{1}{n} \sum_{\mathbf{x} \in \mathcal{D}} \left| \nabla F_{\mathcal{W}}(\mathbf{x}) \right|.$$

We then take the row norms $\{\|J_i\|\}_{i=1}^{d}$ and sort them in ascending order, denoted by $\{\|J_{i*}\|\}_{i=1}^{d}$.

One simple way to measure the gap is to compute the gap $\Delta_i = \frac{J_{(i+1)*} - J_{i*}}{J_{(i+1)*}}$, since the number of data sources $m$ can not be obtained, we use the expectation of such gap $\mathbb{E}(\Delta) = \sum_{i=1}^{d-1} \Delta_i / (d-1)$ to measure the variance difference where a large value of $\mathbb{E}(\Delta)$ indicates that the variance difference is pronounced and that NRDE is particularly appropriate in such cases.

We conducted experiments to measure $\mathbb{E}(\Delta)$ on datasets where NRDE shows performance improvements and on datasets where it exhibits performance drops compared to other density-based methods. The results in Table 19 show that datasets with performance improvements tend to have larger values of $\mathbb{E}(\Delta)$, illustrating that $\mathbb{E}(\Delta)$ is an effective diagnostic for measuring the variance difference.

### H.6    Sensitivity analysis of NRDE to architectural choices

In this subsection, we include additional experiments on five representative datasets to evaluate the sensitivity of our method to architectural choices in Table 20 and Table 21. Overall, NRDE remains robust across different architectures in most cases.

### H.7    Performance comparison between NRDE and other hyperparameter-tuned baselines

For all baseline methods, We follow the widely-used setting in recent papers Yin et al. (2024); Xu et al. (2023b); Livernoche et al. (2023); Shenkar & Wolf (2022) to use the recommended or best-performing hyperparameter configuration given in their original paper. To further eliminate any concerns regarding insufficient tuning, we perform grid search over the hyperparameters on several recent methods: MCM, DTE-C and SLAD based on their original papers and report their best-performing results on each datasets. For MCM, learning rate $lr \in \{0.001, 0.05, 0.01\}$ and $\lambda \in \{0.1, 1, 10\}$. For DTE-C, learning rate $lr \in \{0.001, 0.05, 0.01\}$ and time stamps $T \in \{100, 400, 1000\}$. For SLAD, $lr \in \{0.001, 0.05, 0.01\}$ and hidden dimension $\hat{d} \in \{64, 128, 256\}$. Experimental results in Table 22 show that NRDE still outperforms these tuned baselines in most cases.

### H.8    Outlier detection on more datasets

In this subsection, we conduct outlier detection experiments on the other 5 datasets. As shown in Table 23, NRDE still outperforms these baselines in the transductive setting, demonstrating its robustness to contamination by anomalies in the training set.

### H.9    More Statistical reporting for standard AD

In this subsection, we now include box plots illustrating the performance distributions of different methods across the 47 datasets in Figure 11, as well as the corresponding p-values for each comparison with the baselines in Table 24. While the p-values indicate that the performance improvement of

NRDE over simple methods such as KNN and KDE is not statistically significant, NRDE exhibits statistically significant gains over the other deep learning–based baselines.

### H.10 TRAINING TIME COST COMPARISON BETWEEN NRDE AND REALNVP (NORMALIZING FLOW)

Since NRDE and RealNVP share the same inference procedure, we report only their training times across datasets of varying dimensionality in Table 25. The main time consumption of NRDE compared to RealNVP is the time for Jacobian matrix computation for each training data. The results indicate that, even for large-scale or high-dimensional datasets, NRDE's training time remains comparable to RealNVP with no substantial increase.

### H.11 PERFORMANCE COMPARISON BETWEEN NRDE, $p_{\mathcal{X}}(\mathbf{x}_{\text{PURE}})$ AND $p_{\bar{\mathcal{X}}}(\mathbf{x}_{\text{PURE}})$

In this subsection, we compare the performance of NRDE, $p_{\mathcal{X}}(\mathbf{x}_{\text{pure}})$ and $p_{\bar{\mathcal{X}}}(\mathbf{x}_{\text{pure}})$ on several datasets. Since estimating $p_{\mathcal{X}}(\mathbf{x}_{\text{pure}})$ and $p_{\bar{\mathcal{X}}}(\mathbf{x}_{\text{pure}})$ requires explicitly measuring $m$, which is the number of data sources, here we estimate $m$ using a simple strategy. First, we compute the average absolute Jacobian

$$J = \frac{1}{n} \sum_{\mathbf{x} \in \mathcal{D}} \left| \nabla F_{\mathcal{W}}(\mathbf{x}) \right|.$$

We then take the row norms $\{J_i\}_{i=1}^d$ and sort them in ascending order, denoted by $\{J_{i*}\}_{i=1}^d$. Then we measure the variance gap by computing $\Delta_i = \frac{J_{(i+1)*} - J_{i*}}{J_{(i+1)*}}$, and find $m$ by

$$m = \arg\max_i \Delta_i \tag{44}$$

As shown in Table 26, the performance of using $p_{\mathcal{X}}(\mathbf{x}_{\text{pure}})$ and $p_{\bar{\mathcal{X}}}(\mathbf{x}_{\text{pure}})$ is very close to that of NRDE. A paired t-test on 10 datasets shows that the performance differences between NRDE and these two pure-data-based baselines are not statistically significant ($p > 0.05$).

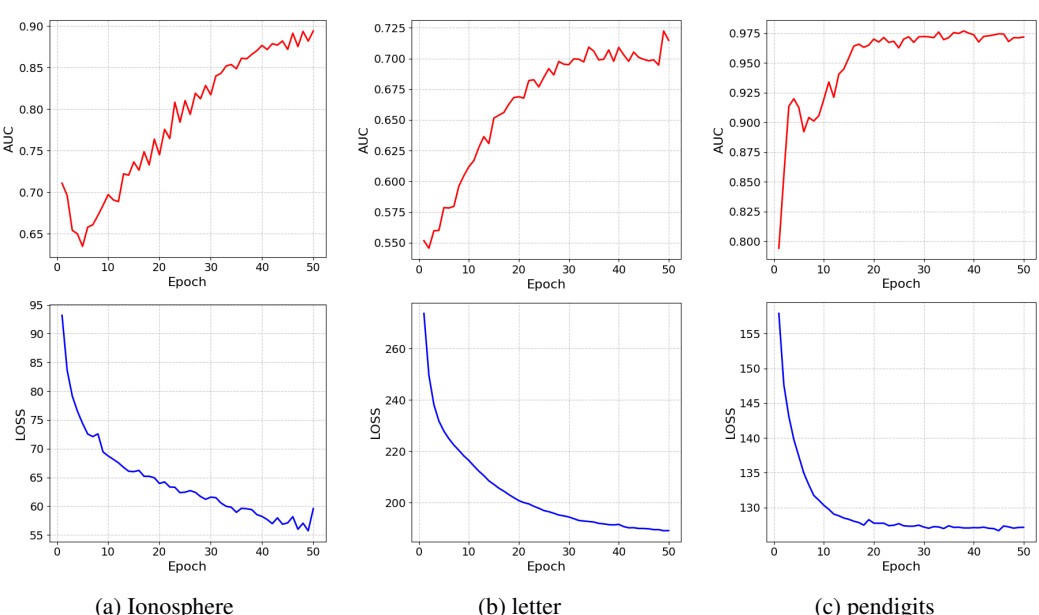

(a) Ionosphere        (b) letter        (c) pendigits

Figure 10: Dynamics of training loss and AUROC across the training procedure for several datasets.

Table 16: AUROC performance comparison of NRDE on $47$ real-world datasets in ADBench with or without $\mathcal{R}$ ($\lambda$ is set to 0).

| Data | $\lambda \neq 0$ | $\lambda = 0$ |
|---|---|---|
| ALOI | 55.1 | 56.7 |
| annthyroid | 98.1 | 98.4 |
| backdoor | 86.7 | 94.5 |
| breastw | 99.3 | 99.4 |
| campaign | 76.4 | 76.9 |
| cardio | 76.1 | 95.8 |
| Cardiotocography | 84.6 | 86.1 |
| celeba | 84.3 | 87.9 |
| census | 73.4 | 76.7 |
| cover | 69.2 | 84.1 |
| donors | 97.3 | 97.6 |
| fault | 58.3 | 62.7 |
| fraud | 92.4 | 95.9 |
| glass | 89.0 | 91.2 |
| Hepatitis | 82.3 | 85.3 |
| http | 99.7 | 99.8 |
| InternetAds | 76.2 | 79.1 |
| Ionosphere | 87.6 | 87.7 |
| landsat | 71.1 | 71.5 |
| letter | 67.0 | 70.2 |
| Lymphography | 98.6 | 98.7 |
| magic.gamma | 80.6 | 81.9 |
| mammography | 89.7 | 91.2 |
| mnist | 91.8 | 93.2 |
| musk | 99.9 | 99.8 |
| optdigits | 94.2 | 94.5 |
| PageBlocks | 92.1 | 92.1 |
| pendigits | 96.2 | 98.1 |
| Pima | 80.8 | 81.7 |
| satellite | 84.0 | 86.8 |
| satimage-2 | 99.6 | 99.7 |
| shuttle | 99.8 | 99.9 |
| skin | 92.1 | 92.5 |
| smtp | 95.7 | 95.6 |
| SpamBase | 86.6 | 87.4 |
| speech | 63.6 | 64.7 |
| Stamps | 93.7 | 95.9 |
| thyroid | 99.3 | 99.2 |
| vertebral | 67.8 | 72.7 |
| vowels | 83.3 | 87.8 |
| Waveform | 91.7 | 91.6 |
| WBC | 99.7 | 99.9 |
| WDBC | 99.6 | 99.9 |
| Wilt | 79.4 | 77.9 |
| wine | 97.2 | 99.1 |
| WPBC | 63.6 | 65.3 |
| yeast | 59.2 | 61.1 |

Table 17: Hyperparameter configuration of NRDE on 47 real-world datasets in ADBench.

| Data | lr | $\lambda$ |
|---|---|---|
| ALOI | 0.005 | 1 |
| annthyroid | 0.001 | 0.1 |
| backdoor | 0.001 | 0.1 |
| breastw | 0.001 | 0.1 |
| campaign | 0.005 | 0.01 |
| cardio | 0.005 | 0.1 |
| Cardiotocography | 0.01 | 0.01 |
| celeba | 0.001 | 0.01 |
| census | 0.005 | 0.01 |
| cover | 0.01 | 1 |
| donors | 0.001 | 0.1 |
| fault | 0.01 | 1 |
| fraud | 0.001 | 1 |
| glass | 0.005 | 0.1 |
| Hepatitis | 0.01 | 0.1 |
| http | 0.01 | 0.01 |
| InternetAds | 0.001 | 0.1 |
| Ionosphere | 0.001 | 0.01 |
| landsat | 0.005 | 0.1 |
| letter | 0.001 | 0.01 |
| Lymphography | 0.005 | 0.1 |
| magic.gamma | 0.01 | 0.1 |
| mammography | 0.005 | 1 |
| mnist | 0.001 | 1 |
| musk | 0.005 | 1 |
| optdigits | 0.005 | 1 |
| PageBlocks | 0.01 | 0.01 |
| pendigits | 0.01 | 0.1 |
| Pima | 0.001 | 0.1 |
| satellite | 0.001 | 1 |
| satimage-2 | 0.001 | 0.1 |
| shuttle | 0.01 | 0.1 |
| skin | 0.001 | 0.1 |
| smtp | 0.01 | 0.1 |
| SpamBase | 0.01 | 0.1 |
| speech | 0.001 | 0.1 |
| Stamps | 0.01 | 0.1 |
| thyroid | 0.005 | 0.01 |
| vertebral | 0.005 | 0.1 |
| vowels | 0.005 | 1 |
| Waveform | 0.001 | 0.1 |
| WBC | 0.01 | 0.1 |
| WDBC | 0.01 | 0.1 |
| Wilt | 0.01 | 1 |
| wine | 0.005 | 1 |
| WPBC | 0.005 | 0.1 |
| yeast | 0.01 | 1 |

Table 18: AUROC results (%) of the best-performing 5 methods on anomaly detection with noisy data, where stronger noise is added to the anomalous test samples. The best results per dataset are in **bold**.

| Dataset | DSVDD | KPCA | IF | kNN | NRDE (ours) |
|---------|-------|------|------|------|-------------|
| Cardiotocography | **83.7** | 75.8 | 80.7 | 71.3 | 82.1 |
| Pima | 72.5 | 77.0 | 75.8 | 78.1 | **79.6** |
| Satellite | 81.5 | 84.1 | 79.6 | **86.9** | 85.1 |
| SpamBase | 80.3 | **86.3** | 82.4 | 83.0 | 79.1 |
| WPBC | 47.5 | 52.2 | 51.7 | 51.5 | **62.9** |
| AVG | 73.1 | 75.1 | 74.0 | 74.2 | **77.2** |

Table 19: Average gap values on datasets showing significant performance improvement or drop compared to other density-based methods.

| Datasets (Improvement) | average gap |
|------------------------|-------------|
| annthyroid | 0.23 |
| smtp | 0.58 |
| vertebral | 0.16 |
| Pima | 0.35 |
| Cardiotocography | 0.14 |
| **Datasets (Drop)** | average gap |
| Ionosphere | 0.04 |
| landsat | 0.03 |
| letter | 0.03 |
| optdigits | 0.07 |
| pendigits | 0.06 |

Table 20: Average AUROC performance of the proposed method with different numbers of coupling layers (T).

| AUROC | SpamBase | Satellite | Pima | WPBC | Cardiotocography |
|-------|----------|-----------|------|------|------------------|
| $T = 2$ | 87.4 | 86.8 | 81.7 | 65.3 | 86.1 |
| $T = 3$ | 84.2 | 87.4 | 83.0 | 63.9 | 75.7 |
| $T = 4$ | 86.3 | 88.3 | 82.3 | 62.7 | 78.3 |

Table 21: Average AUROC performance of the proposed method with different width of coupling layers (b).

| AUROC | SpamBase | Satellite | Pima | WPBC | Cardiotocography |
|-------|----------|-----------|------|------|------------------|
| $b = 512$ | 86.2 | 85.1 | 80.2 | 65.5 | 84.5 |
| $b = 1024$ | 86.1 | 85.0 | 81.0 | 64.6 | 85.3 |
| $b = 2048$ | 87.4 | 86.8 | 81.7 | 65.3 | 86.1 |

Table 22: AUROC (%) comparison between tuned MCM, SLAD, DTE-C and our proposed method on tabular datasets of different dimensionalities from ADBench.

| Dataset | SLAD | MCM | DTE-C | Ours |
|---|---|---|---|---|
| **Low-dimensional (<10 features)** | | | | |
| annthyroid | 91.3 | 98.5 | 98.0 | 98.4 |
| glass | 83.5 | 86.5 | 81.6 | 91.2 |
| mammography | 79.5 | 90.7 | 89.8 | 91.2 |
| Pima | 62.1 | 75.2 | 71.1 | 81.7 |
| vertebral | 46.7 | 56.0 | 67.9 | 72.7 |
| **Middle-dimensional (10–100 features)** | | | | |
| Cardiotocography | 61.1 | 79.5 | 75.1 | 86.1 |
| fraud | 94.8 | 95.8 | 95.8 | 95.9 |
| satellite | 88.1 | 85.7 | 83.0 | 86.8 |
| satimage-2 | 99.7 | 99.9 | 99.0 | 99.7 |
| shuttle | 99.9 | 99.9 | 99.7 | 99.9 |
| **High-dimensional (>100 features)** | | | | |
| backdoor | 92.5 | 97.2 | 92.3 | 94.5 |
| census | 70.0 | 72.0 | 71.1 | 76.7 |
| mnist | 91.2 | 95.3 | 91.8 | 93.2 |
| musk | 100.0 | 100.0 | 100.0 | 99.8 |
| speech | 55.4 | 49.9 | 53.7 | 64.7 |

Table 23: Detailed AUROC performance of outlier detection on 10 datasets. The best results are marked in **bold**.

| AUROC | Cardiotocography | SpamBase | Satellite | Pima | WPBC | glass | optdigits | PageBlocks | pendigits | Waveform | AVG |
|---|---|---|---|---|---|---|---|---|---|---|---|
| IF | 68.8 | 65.5 | 67.9 | 67.0 | 49.1 | 78.2 | 74.1 | 89.1 | **95.5** | 72.8 | 72.8 |
| ECOD | 78.5 | 65.5 | 58.2 | 59.4 | 48.1 | 70.4 | 60.4 | 91.3 | 92.7 | 60.3 | 68.4 |
| OC-SVM | 69.5 | 53.3 | 66.3 | 62.3 | 48.4 | 59.9 | 50.7 | **91.4** | 93.1 | 67.1 | 66.2 |
| KPCA | 53.4 | 52.1 | 48.2 | 53.8 | 45.5 | 49.9 | 52.2 | 64.3 | 57.2 | 56.0 | 53.2 |
| LOF | 52.3 | 45.6 | 54.1 | 60.1 | 52.0 | 77.0 | 53.7 | 71.5 | 49.9 | 70.5 | 58.6 |
| KDE | 50.2 | 49.5 | 76.0 | 72.2 | 49.9 | 82.0 | 32.2 | 90.6 | 89.0 | 75.1 | 66.6 |
| kNN | 57.9 | 52.9 | 65.0 | 65.1 | 47.2 | **86.7** | 37.2 | 88.8 | 75.8 | 73.4 | 65.0 |
| RealNVP | 62.7 | 56.5 | 74.6 | 70.7 | 59.1 | 79.6 | 72.3 | 86.4 | 91.1 | 69.8 | 72.2 |
| COPOD | 66.2 | 68.7 | 63.3 | 65.4 | 52.3 | 75.5 | 68.2 | 87.5 | 90.4 | 73.3 | 71.0 |
| DeepIF | 63.0 | 37.9 | 74.3 | 61.3 | 49.4 | 84.5 | 56.3 | 87.5 | 95.3 | 78.6 | 68.8 |
| Ours | **80.5** | **77.7** | **81.9** | **80.1** | **62.2** | 85.0 | **75.3** | 82.7 | 88.3 | **91.8** | **80.5** |

Table 24: Paired t-test between NRDE and each baseline over 47 tabular datasets.

| | KDE | KNN | LOF | OC-SVM | IF | AE | DSVDD | RealNVP | NeutralAD | ECOD | ICL | SLAD | DPAD | MCM | DTE-C |
|---|---|---|---|---|---|---|---|---|---|---|---|---|---|---|---|
| AUROC $p$ | 0.0547 | 0.0990 | 0.0184 | 2.3e-05 | 9.4e-06 | 0.0091 | 1.1e-06 | 0.0003 | 5.3e-05 | 2.1e-09 | 3.7e-06 | 0.0006 | 0.0004 | 0.0041 | 0.0269 |
| AUPRC $p$ | 0.6429 | 0.8984 | 0.0419 | 0.0011 | 1.6e-07 | 0.0422 | 6.2e-06 | 5.0e-06 | 0.0018 | 2.1e-08 | 4.3e-04 | 0.0762 | 0.0360 | 0.1833 | 0.0674 |

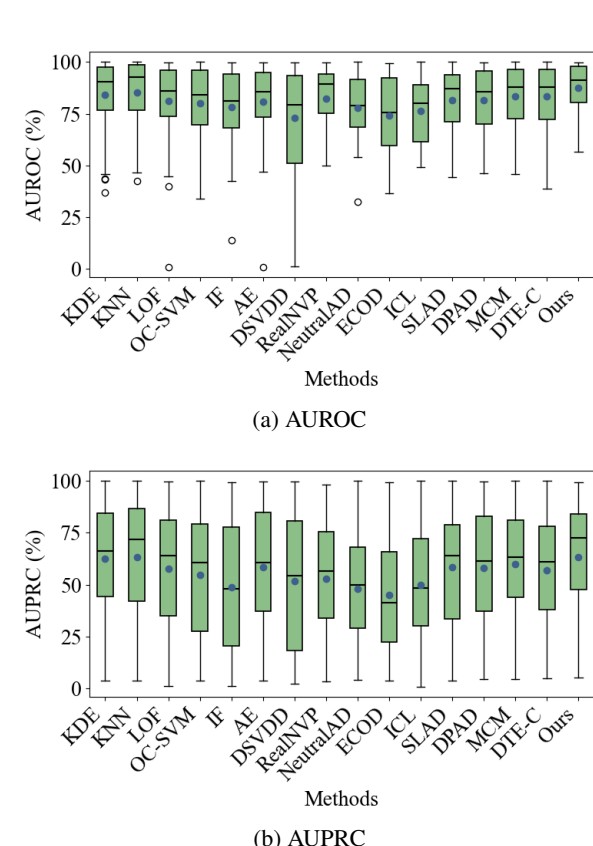

(a) AUROC

(b) AUPRC

Figure 11: Box plots comparing the performance distributions of different methods across the 47 datasets

Table 25: 100 epoch training time cost ($s$) comparison between NRDE and RealNVP on different dimensional datasets from ADBench.

| Dataset | NRDE | RealNVP |
|---|---|---|
| annthyroid | 4.12 | 2.03 |
| glass | 1.58 | 0.59 |
| mammography | 5.46 | 2.72 |
| Pima | 1.68 | 0.61 |
| vertebral | 1.45 | 0.59 |
| Cardiotocography | 4.29 | 0.90 |
| fraud | 388.80 | 75.21 |
| satellite | 30.31 | 18.8 |
| satimage-2 | 31.40 | 19.04 |
| shuttle | 46.26 | 30.05 |
| mnist | 71.34 | 20.84 |
| musk | 70.41 | 18.03 |

Table 26: Performance comparison between NRDE, $p_{\mathcal{X}}\left(\mathbf{x}_{\text{pure}}\right)$ and $p_{\bar{\mathcal{X}}}\left(\mathbf{x}_{\text{pure}}\right)$.

|  | NRDE | $p_{\mathcal{X}}\left(\mathbf{x}_{\text{pure}}\right)$ | $p_{\bar{\mathcal{X}}}\left(\mathbf{x}_{\text{pure}}\right)$ |
|---|---|---|---|
| Satellite | 86.1 | 86.6 | **88.0** |
| WPBC | 65.3 | **71.4** | 65.6 |
| Cardio | 86.1 | **87.2** | 86.9 |
| Pima | **81.7** | 79.7 | 80.4 |
| SpamBase | 87.4 | 85.5 | **87.9** |
| annthyroid | **98.1** | 97.5 | 97.2 |
| smtp | **95.6** | 95.2 | 95.5 |
| glass | 91.2 | 91.5 | **92.6** |
| mammography | 91.2 | **91.4** | 90.9 |
| vertebral | 72.7 | 77.4 | **81.0** |
| AVG | 85.5 | 86.3 | 86.6 |
| $p-$value | - | 0.36 | 0.24 |