# OpenReview forum: "Noise-Robust Density Estimation for Tabular Data Anomaly Detection"
_ICLR.cc/2026/Conference — Submitted to ICLR 2026_

### Official Review · Reviewer_BvpA · 2025-10-23

**Soundness:** 2
**Presentation:** 1
**Contribution:** 2
**Rating:** 2
**Confidence:** 4

**Summary:**

a normalizing flow method for anomaly detection that separates the signal part from the noise part.

**Strengths:**

The idea to separate noise from "pure data" in the modeling process seems as a good basis for research.

**Weaknesses:**

The assumption in Eq.6 drives this work. However, this is not validated nor justified. For me, it looks more like a matter of defining what is considered data and what is considered noise. Also, what determines the scale of the components in the vector s. They can be arbitrary scaled and still independent. On the other hand, if the scale is influenced by their influence on the data, then low variance implies little effect, so the difference they made can be negligible.

The writing in general is very poor, making the paper hard to understand. This is true for all sections, starting from the abstract. I had to stop reading the methodology section early since it was becoming frustrating to keep up. For example, p_X is not defined, and then it is used in Eq.9. Also, the denominator of Eq.9 is clearly zero.

I have little trust in the results of Figure 4. Although ADBench is a public benchmark, the authors run on it using their own code. Also, the uploaded code is partial and does not reproduce the results.

**Questions:**

see above

---

> ### Author Response · Authors · 2025-11-25
> **Response to Reviewer BvpA**
>
> Dear Reviewer BvpA,
>
> We sincerely appreciate your valuable comments and suggestions. Additional experimental results conducted during the rebuttal phase are provided in **Appendix H of our paper** for the reviewers’ convenience. Please refer to it for details. Our responses are as follows.
>
> **Weaknesses**
>
> **Response to W1**
>
> Thank you for this insightful question.
>
> For assumption verification, we would like to clarify that the assumption is verified in the following ways:
>
> (i) visualization of Jacobian row norms on real and synthetic datasets from Figure 6 and Figure 7 of Appendix G.1 illustrates the relationship between Jacobian row norms and variance of different sources.
>
> (ii) In Appendix G.6 of our paper, we conducted several experiments on both real and synthetic datasets to verify our assumption. Please refer to it.
>
> (iii) As discussed in the paper, directly verifying the variance-related assumption on real datasets is inherently challenging because the underlying generative process $G$ or its inverse  $F_{\mathcal{W}}^*$ is not accessible during training.
> A practical way to indirectly verify our assumptions is to compare NRDE’s performance with that of other baselines on the 47 datasets, particularly with RealNVP (normalizing flow). The substantial performance gains of NRDE over RealNVP on most datasets suggest that the assumptions reasonably hold in the majority of cases.
>
> (iv) During rebuttal, we conduct another experiments on real data to verify our assumption. Please refer to it in **Appendix H.5 and H.11 of our paper** for details.
>
> As for the scale of components in $s$, we do not explicitly impose a scale on the components of
> $s$. In general, components drawn from distributions $\mathcal{N}(0,\sigma)$ with larger variance $\sigma$ naturally exhibit larger scale. Importantly, low-variance components can still exert substantial influence on the observed data depending on the underlying (and unknown) generative process $G$. For example, in the simplified linear case  $x=G(s)=Ws$, the effect of each component is governed jointly by its scale and the corresponding transformation weights in $W$.
>
> **Response to W2**
>
> Thank you for pointing out the writing issues. We have divided the methodology into 3 parts and substantially revised the methodology section to address the specific problems you mentioned (highlighted in orange). In the original Eq.9, all $p$ denotes the probability density rather than probability and we used $p(\mathbf{s}\_N=0)$ (the denominator) to denote the density of $\mathbf{s}\_N$ at $0$, meaning that denominator is not zero; a formal notation should be $p\_{S_N}(\mathbf{0})$. $\mathcal{X}$ is the distribution of $\mathbf{x}$, which was defined in line 175 of the old version of the paper. Therefore, $p_\mathcal{X}$ is density function of $\mathbf{x}$. Since Eq.9 is a little confusing and does not provide a concrete result, we replace it with the following formula
>
> $p\_{\mathcal{X}}(\mathbf{x}\_{{\text{pure} }})=p\_{\mathcal{Z}}(F(\mathbf{x}\_{\text {pure }}))|\operatorname{det}(\nabla\_{\mathbf{x}\_{\text{pure}}}F\_\mathcal{W}(\mathbf{x}\_{\text{pure}}))|$
>
> where $p_\mathcal{Z}$ denotes the density function of $\mathbf{z}$.
>
> We kindly invite you to refer to the updated version of the paper for the clarified and corrected presentation. We have also revised other sections to further improve readability and ensure that the overall exposition is clear and easy to follow.
>
> **Response to W3**
>
> Thank you for pointing out the reproducibility issue. Although the implementations of several baseline methods are not included directly in our code release, we rely on either the official code provided by the original authors or the implementations available in PyOD, a widely used Python library for anomaly detection. Regarding the reproducibility of our own method, the released code corresponds to the outlier detection (transductive anomaly detection) setting. To reproduce the results under the standard anomaly detection setting, one only needs to replace the data-loading function read\_OD\_data with read\_data. In addition, we provide the full hyperparameter configurations for each dataset in Table 17 from **Appendix H.2 of our paper** to further facilitate reproducibility.
>
> **We are grateful for the reviewer’s thoughtful evaluation of our work and rebuttal. Should any issues remain unclear, we would be glad to provide further explanation.**

---

> > ### Comment · Reviewer_BvpA · 2025-11-26
> >
> > After reading the rebuttal I did not find the work to be better motivated. My concerns regarding the experiments are unchanged.

---

> ### Author Response · Authors · 2025-11-27
>
> Dear Reviewer BvpA,
>
> Thanks for your feedback. Our responses are as follows:
>
> **Q1: After reading the rebuttal I did not find the work to be better motivated.**
>
> R:
>
> - We would like to clarify our motivation more explicitly. As illustrated in **line 91 and Figure 1 of our paper**, density-based anomaly detectors—including normalizing flows—are highly sensitive to the intrinsic noise present in data, which can lead to high false-positive rates.
>
> - To address this limitation, we propose the regularized normalizing flow, which divides the latent variables into two groups: signal (data) sources and noise sources. By identifying and isolating the noise sources, we can compute the density of pure data which is generated only by signal sources. This design ensures that the anomaly score (density of pure data) remains uninfluenced by noise factors, thereby providing robustness against inherent noise and significantly reducing noise-induced false alarms. We hope this further clarifies the underlying motivation of our method.
>
> **Q2: Concerns regarding the experiments are unchanged.**
>
> R:
>
> - For experiments, the reason why code of ADbench is not directly used is that in the original paper,  the authors  use 70\% data for training and the remaining 30\% as a testing set, whereas we follow another widely used UAD setting in recent papers ([1]-[4]) mentioned in **line 409 of our paper**: 50\% of normal samples are randomly used for training, while the remaining 50\% are merged with all anomalies to form the test set.
>
> - All baseline methods in our evaluation are implemented using publicly available official or open-source codebases and their best-performing hyperparameter configurations are used, as stated in line 417 of our paper. This ensures that comparisons between NRDE and other baselines across different experimental scenarios are fair and unbiased.
>
> - We also provide an additional baseline tuning protocol, where grid search is applied individually on each dataset to determine the optimal hyperparameters for all baselines (see **Appendix H.7**).
>
> - Moreover, we conduct comprehensive ablation studies to validate the contribution of each core component in NRDE, covering hyperparameters, network architecture, and other design choices, demonstrating that performance gains stem from principled model design rather than isolated factors.
>
> - To further support reproducibility, we have now provided complete runnable code for all baseline models and our method in the revised **Supplementary Material**, which fully regenerates the results presented in Figure 4.
>
> - If any concerns still remain regarding our experimental implementation or reproducibility, we would truly appreciate your guidance in specifying the exact aspects that require revision or clarification, so that we can address them accurately and openly.
>
> [1] Yin, Jiaxin, et al. "MCM: Masked cell modeling for anomaly detection in tabular data." The Twelfth International Conference on Learning Representations. 2024.
>
> [2] Livernoche, Victor, et al. "On diffusion modeling for anomaly detection." arXiv preprint arXiv:2305.18593 (2023).
>
> [3] Xu, Hongzuo, et al. "Fascinating supervisory signals and where to find them: Deep anomaly detection with scale learning." International Conference on Machine Learning. PMLR, 2023.
>
> [4] Shenkar, Tom, and Lior Wolf. "Anomaly detection for tabular data with internal contrastive learning." International conference on learning representations. 2022.

---

### Official Review · Reviewer_Xjrr · 2025-10-25

**Soundness:** 3
**Presentation:** 2
**Contribution:** 2
**Rating:** 4
**Confidence:** 4

**Summary:**

The paper proposes a noise-robust density estimator for tabular anomaly detection. A normalizing flow is trained with a Jacobian-based regularizer so that latent coordinates separate into signal and nuisance groups. After training, test points are scored by a weighted log density that emphasizes high-variance latent directions, which is meant to approximate the density of the noise-free component and reduce false alarms on normal but noisy data. The method is clearly motivated and the pipeline from the generative view to the practical scoring rule is coherent. Experiments on forty-seven ADBench datasets and several stress settings report strong average ranks against classical and modern baselines, with ablations that attribute gains to both the regularizer and the weighted density. Figures and tables make the intuition tangible.

I like the idea and the careful framing, but some choices limit the strength of the claims. The approach assumes that signal sources have much larger variance than noise sources and that certain smoothness conditions hold. These are plausible but not verified on real datasets. Weight computation from average Jacobian norms can be sensitive to feature scaling and to architectural choices. Several baselines look under-tuned, and search spaces differ; some figures lack uncertainty intervals. The noisy data protocol perturbs normal points more strongly than anomalies, which may advantage methods that explicitly discount noise. The method is also costlier than a plain flow during training, and the work would benefit from an end-to-end latency profile. With additional diagnostics for the assumptions, more uniform tuning, and fuller statistics, the contribution would be quite solid in practice.

**Strengths:**

The paper addresses a real pain point of density-based anomaly detection on tabular data, namely false alarms caused by nuisance variation that is not semantically relevant for the task. The proposed Jacobian regularized flow with a weighted log density is conceptually simple, implementable with common flow libraries, and well-documented. The modeling story is coherent from the generative view through the practical scoring rule. The empirical section spans a broad range of datasets and settings, and the authors provide extensive tables and visualizations. I especially liked the clear diagrams, as well as the Jacobian row norm plots, which make the mechanism tangible. The ablations show that each component adds value, and the paper includes a time complexity discussion and implementation details that will help reproduction.

**Weaknesses:**

The core assumption that signal sources have markedly larger variance than noise sources is strong and domain-dependent, and the paper does not provide a quantitative diagnostic to verify it on real data before applying the method. The weighting relies on average Jacobian norms, which are known to be sensitive to scaling and to the architecture of the flow. The study hints at robustness, but a more systematic sensitivity analysis would be welcome. Several baselines appear under-tuned or constrained, and the tuning budgets differ, which may inflate gains. Statistical reporting is thin for some figures, and the noisy data protocol perturbs normal test points more strongly than anomalies, which may bias the comparison. The theoretical section assumes reasonable conditions for analysis, but these conditions are not verified for all datasets. Finally, practicality at scale is mixed. The method computes Jacobian norms and adds a regularizer during training, which is more costly than a plain flow. The paper compares testing complexity in Appendix B, but an end-to-end latency and memory profile for large datasets would provide a clearer picture of the practical implications.

**Questions:**

1) Could you provide a simple diagnostic that estimates the strength of variance separation between signal and noise sources on a given dataset, for example, a gap statistic over Jacobian row norms or a permutation-based calibration, and advise a user when NRDE is appropriate?

2) How sensitive are the results to feature scaling and to the choice of flow architecture, for instance, the number of coupling layers and the width?

3) In the noisy data experiment, normal test points receive stronger perturbation than anomalies. Would you consider a symmetric design and report the same table.

4) Can you add confidence intervals for AUROC and AUPRC in the main figures and expand the transductive study to more datasets?

5) Could you include a small empirical audit of the modeling assumptions, such as independence proxies or partial correlation summaries, and report how often the assumptions seem to hold across the forty-seven datasets?

---

> ### Author Response · Authors · 2025-11-25
> **Response to Reviewer Xjrr(1/6)**
>
> Dear Reviewer Xjrr：
>
> We sincerely appreciate your recognition of the motivation and effectiveness of our method. Thank you very much for your valuable suggestions, which will help us further improve the quality of our work. Additional experimental results conducted during the rebuttal phase are provided in **Appendix H of our paper** for the reviewers’ convenience. Please refer to it for details. Our responses are as follows.
>
> **Response to W1 (a quantitative explicit diagnostic to verify variance difference)**
>
> We thank the reviewer for this insightful question. A simple way to empirically verify the variance difference underlying our assumption is as follows. We first train an unregularized normalizing flow $F_{\mathcal{W}}$ on the real dataset $\mathcal{D}$ and compute the average absolute Jacobian
> $
> {J} = \frac{1}{n} \sum_{\mathbf{x}\in \mathcal{D}} \bigl|\nabla F_{\mathcal{W}}(\mathbf{x})\bigr|.
> $
> We then take the row norms $\\{{J}\_i\\}\_{i=1}^d$ and sort them in ascending order, denoted by  $\\{{J}\_{i^\ast}\\}\_{i=1}^d$.
>
> One simple way to measure the gap is to compute the gap  $ \Delta\_i=(J\_{(i+1)^\ast}-J_{i^\ast}) / J\_{(i+1)^\ast}$, since the number of data sources $m$ can not be obtained, we use the expectation of such gap $\mathbb{E}(\Delta)=\sum_{i=1}^{d-1}\Delta_i/(d-1)$ to measure the variance difference where a large value of $\mathbb{E}(\Delta)$ indicates that the variance difference is pronounced and that NRDE is particularly appropriate in such cases.
>
> We conducted experiments to measure $\mathbb{E}(\Delta)$ on datasets where NRDE shows performance improvements and on datasets where it exhibits performance drops compared to other density-based methods. The results in the following table show that datasets with performance improvements tend to have larger values of $\mathbb{E}(\Delta)$, illustrating that $\mathbb{E}(\Delta)$ is an effective diagnostic for measuring the variance difference.  Additional experiments on both synthetic and real datasets that further validate our assumptions and motivation can be found in Appendix G.6.
>
> **Table 1.** Average gap values on datasets showing significant performance improvement or drop compared to other density-based methods.
>
> | **Datasets (Improvement)** | **$\mathbb{E}(\Delta)$** |
> |-|-|
> | annthyroid | 0.23|
> | smtp| 0.58|
> | vertebral| 0.16|
> | Pima| 0.35|
> | Cardiotocography| 0.14|
> | **Datasets (Drop)**| **$\mathbb{E}(\Delta)$** |
> | Ionosphere| 0.04|
> | landsat| 0.03 |
> | letter| 0.03|
> | optdigits| 0.07|
> | pendigits| 0.06|
>
> **Response to W2 (weighting can be sensitive to scaling and to the architecture of the flow.)**
>
> Thank you for raising this insightful question. For scaling,  each dataset is preprocessed using z-score normalization—ensuring that every feature has zero mean and unit variance, which cancels out scaling differences.
>
> Regarding the influence of the flow architecture (e.g., the number of coupling layers and network width), we now include additional experiments on five representative datasets to evaluate the sensitivity of our method to architectural choices. Overall, NRDE remains robust across different architectures in most cases.
>
> **Table 2.** Average AUROC performance of the proposed method with different number of coupling layers ($T$).
>
> | AUROC   | SpamBase | Satellite | Pima  | WPBC
> |-|:-:|:-:|:-:|:-:
> | $T = 2$ | 87.4     | 86.8      | 81.7  | 65.3
> | $T = 3$ | 84.2     | 87.4      | 83.0  | 63.9
> | $T = 4$ | 86.3     | 88.3      | 82.3  | 62.7
>
> **Table 3.** Average AUROC performance of the proposed method with different width of coupling layers ($b$).
>
> | AUROC| SpamBase | Satellite | Pima  | WPBC
> |-|:-:|:-:|:-:|:-:
> | $b=512$  | 86.2     | 85.1      | 80.2  | 65.5
> | $b=1024$ | 86.1     | 85.0      | 81.0  | 64.6
> | $b=2048$ | 87.4     | 86.8      | 81.7  | 65.3

---

> > ### Author Response · Authors · 2025-11-25
> > **Response to Reviewer Xjrr(2/6)**
> >
> > **Response to W3 (A more systematic sensitivity analysis)**
> >
> > Thank you for pointing this out. We summarize below the sensitivity analyses already included in the paper and in our rebuttal:
> >
> > - Sensitivity to different AD scenarios: NRDE is evaluated under multiple settings, including standard AD, AD on noisy data, AD with anomaly contamination, and transductive AD.
> >
> > - Sensitivity to injected noise: Experiments on noisy data assess the robustness of NRDE to artificial noise perturbations.
> >
> > - Sensitivity to anomalies in training data: Experiments involving anomaly contamination and transductive AD evaluate how NRDE behaves when training data include anomalous samples.
> >
> > - Sensitivity to hyperparameters: The ablation studies investigate the impact of hyperparameter choices.
> >
> > - Sensitivity to variance assumptions: Appendix G.6 evaluates NRDE’s behavior under different variance configurations and under contradictory assumptions.
> >
> > In addition, during the rebuttal period, we conducted further experiments analyzing NRDE’s sensitivity to architectural changes, injected noise, and anomaly contamination, providing a more systematic view of its robustness.
> >
> > If there are additional sensitivity aspects that should be considered, we would be glad to investigate them, as we share your interest in providing a comprehensive and systematic sensitivity analysis.
> >
> > **Response to W4 (several baselines appear under-tuned)**
> >
> > Thanks for pointing out. For these baseline methods, We follow the widely-used setting in recent papers ([1]-[4]) to use the recommended or best-performing hyperparameter configuration  given in their original paper.
> >
> > To further eliminate any concerns regarding insufficient tuning, we perform grid search over the hyperparameters on several recent methods: MCM, DTE-C and SLAD based on their original papers and report their best-performing results on each datasets. For MCM,  learning rate $lr \in \{0.001,0.05,0.01\}$ and $\lambda \in \{0.1,1,10\}$. For DTE-C, learning rate $lr \in \{0.001,0.05,0.01\}$ and time stamps $T \in \{100,400,1000\}$. For SLAD, $lr \in \{0.001,0.05,0.01\}$ and hidden dimension $\hat{d} \in \{64,128,256\}$. Experimental results in the following table show that NRDE still outperforms these tuned baselines in most cases.
> >
> > [1] Yin, Jiaxin, et al. *"MCM: Masked cell modeling for anomaly detection in tabular data."* The Twelfth International Conference on Learning Representations. 2024.
> >
> > [2] Livernoche, Victor, et al. *"On diffusion modeling for anomaly detection."* arXiv preprint arXiv:2305.18593 (2023).
> >
> > [3] Xu, Hongzuo, et al. *"Fascinating supervisory signals and where to find them: Deep anomaly detection with scale learning."* International Conference on Machine Learning. PMLR, 2023.
> >
> > [4] Shenkar, Tom, and Lior Wolf. *"Anomaly detection for tabular data with internal contrastive learning."* International Conference on Learning Representations. 2022.
> >
> > **Table 4.** AUROC (%) comparison between tuned MCM, SLAD, DTE-C and our proposed method on tabular datasets of different dimensionalities from ADBench.
> >
> > | Dataset| SLAD | MCM  | DTE-C | Ours |
> > |-|:-:|:-:|:-:|:-:|
> > | **Low-dimensional ($<$10 features)** |||||
> > | annthyroid| 91.3 | **98.5** | 98.0  | 98.4 |
> > | glass| 83.5 | 86.5 | 81.6  | **91.2** |
> > | mammography | 79.5 | 90.7 | 89.8  | **91.2** |
> > | Pima| 62.1 | 75.2 | 71.1  | **81.7** |
> > | vertebral| 46.7 | 56.0 | 67.9  | **72.7** |
> > | **Middle-dimensional (10–100 features)** |||||
> > | Cardiotocography  | 61.1 | 79.5 | 75.1  | **86.1** |
> > | fraud| 94.8 | 95.8 | 95.8  | **95.9** |
> > | satellite| **88.1** | 85.7 | 83.0  | 86.8 |
> > | satimage-2| 99.7 | **99.9** | 99.0  | 99.7 |
> > | shuttle| 99.9 | 99.9 | 99.7  | **99.9** |
> > | **High-dimensional ($>$100 features)** |||||
> > | backdoor | 92.5 | **97.2** | 92.3  | 94.5 |
> > | census | 70.0 | 72.0 | 71.1  | **76.7** |
> > | mnist| 91.2 | **95.3** | 91.8  | 93.2 |
> > | musk| **100.0**| 100.0| 100.0 | 99.8 |
> > | speech| 55.4 | 49.9 | 53.7  | **64.7** |

---

> > > ### Author Response · Authors · 2025-11-25
> > > **Response to Reviewer Xjrr(3/6)**
> > >
> > > **Response to W5 (Statistical reporting is thin for some figures)**
> > >
> > > Thank you for pointing out this oversight. We have now included box plots illustrating the performance distributions of different methods across the 47 datasets in **Figure 11 from Appendix H of our paper**, as well as the corresponding p-values for each comparison with the baselines. While the p-values indicate that the performance improvement of NRDE over simple methods such as KNN  is not statistically significant, NRDE exhibits statistically significant gains over the other deep learning–based baselines.
> > >
> > > **Table 5.** Paired t-test between NRDE and each baseline over 47 tabular datasets.
> > >
> > > | | KDE| KNN| LOF | OC-SVM | IF| AE| DSVDD| RealNVP | NeutralAD | ECOD| ICL| SLAD| DPAD| MCM  | DTE-C|
> > > |-|-|-|-|-|-|-|-|-|-|-|-|-|--|-|-|
> > > | AUROC p-value | 0.0547 | 0.0990 | 0.0184 | 2.3e-05 | 9.4e-06 | 0.0091 | 1.1e-06 | 0.0003  | 5.3e-05  | 2.1e-09 | 3.7e-06 | 0.0006 | 0.0004 | 0.0041 | 0.0269 |
> > > | AUPRC p-value | 0.6429 | 0.8984 | 0.0419 | 0.0011 | 1.6e-07 | 0.0422 | 6.2e-06 | 5.0e-06  | 0.0018   | 2.1e-08 | 4.3e-04 | 0.0762 | 0.0360 | 0.1833 | 0.0674
> > >
> > >
> > > **Response to W6 (noisy data protocol perturbs normal test points more strongly than anomalies)**
> > >
> > > Thank you for this insightful question. Adding stronger noise to normal test samples than to anomalous ones reduces the separability between the two classes, making the detection task more challenging than under equal-noise perturbations. Consequently, this protocol provides a stricter evaluation of robustness to artificially injected noise.
> > >
> > > For completeness, we also consider a symmetric design in which stronger noise is added to the anomalous test samples. We conduct experiments on the same five datasets, and the corresponding results are presented in the table below. These experiments further demonstrate the robustness of NRDE under different artificial-noise-injection protocols.
> > >
> > > **Table 6.** AUROC results (%) of the best-performing 5 methods on anomaly detection with noisy data. The best results per dataset is in **bold**.
> > >
> > > | Dataset| DSVDD | KPCA  | IF    | kNN   | NRDE (ours) |
> > > |-|:-----:|:-:|:-:|:--:|:-:|
> > > | Cardiotocography  | **83.7** | 75.8 | 80.7 | 71.3 | 82.1|
> > > | Pima| 72.5 | 77.0 | 75.8 | 78.1 | **79.6** |
> > > | Satellite| 81.5 | 84.1 | 79.6 | **86.9** | 85.1 |
> > > | SpamBase| 80.3 | **86.3** | 82.4 | 83.0 | 79.1 |
> > > | WPBC  | 47.5 | 52.2 | 51.7 | 51.5 | **62.9** |
> > > | **AVG**| 73.1 | 75.1 | 74.0 | 74.2 | **77.2** |
> > >
> > > **Response to W7 (theoretical conditions verification)**
> > >
> > > Thank you for pointing this out. We summarize below the assumptions made in the theoretical section and how they are verified:
> > >
> > > **(i)** Assumption in Eq. (6): variance difference between data sources and noise sources.
> > > Appendix G.6 provides several experiments on both synthetic and real datasets that support this assumption:
> > >
> > > (1) Performance results on datasets where variance difference is not satisfied: We analyze the performance of NRDE on synthetic datasets where the variance difference is not satisfied. Suppose the variance of pure data sources is $\sigma_d^2$, and the variance of noise sources is $\sigma^2_n$, we now report the performance results on synthetic datasets with different $\frac{\sigma_d^2}{\sigma_n^2}$ in the following table . The performance decline of NRDE in datasets with smaller variance difference verifies our assumptions and motivations.
> > > **Table 7.** AUROC performance of NRDE on synthetic datasets with different $\frac{\sigma_d^2}{\sigma_n^2}$ ratios.
> > >
> > > | $\frac{\sigma_d^2}{\sigma_n^2}$ | 9    | 6    | 4    | 2    | 1    | 0.5  |
> > > |-|-|-|-|-|-|-|
> > > | NRDE| 87.5 | 82.9 | 80.4 | 77.9 | 71.2 | 68.6 |
> > >
> > > (2) Performance results comparison with ideal baselines: In synthetic dataset where $m$ the number of data sources is known, we compare the performance of NRDE with KDE-C, DSVDD-C and KNN-C which are evaluated on datasets without noise components and NRDE$-m$, where only the $m$ sources with the largest variance from set $A$ are used for computing anomaly score: $u_m(x_{new})=\log | \det (\nabla_{x_{new}} F_\mathcal{W}(x_{new}))|-\frac{d}{2}\log2\pi\ -\frac{1}{2}\sum_{i\in A}w_iF_\mathcal{W}(\mathbf{x}_\text{new})_i^2$. The results are shown in the following table. Since NRDE is an approximation of NRDE$-m$, its performance being close but not as good as NRDE$-m$ and other ideal baselines supports our claim and motivation.
> > >
> > > **Table 8.** AUROC (%) performance of NRDE and other ideal baselines on the synthetic dataset.
> > >
> > > | Method | NRDE | NRDE-m | KDE-C | KNN-C | DSVDD-C |
> > > |-|-|--|-|-|--|
> > > | AUROC| 83.1 | 86.3   | 87.2  | 90.2  | 87.5    |

---

> ### Author Response · Authors · 2025-11-25
> **Response to Reviewer Xjrr(4/6)**
>
> (3) Experimental results using a contradictory assumption: If we make a contradictory assumption that the variances of data sources should be smaller, then the weight for each source should be defined as:
>
> $w_i ={\exp\Big({||(\frac{1}{n}\sum\_{\mathbf{x}\in\mathcal{D}}\vert\nabla\_{\mathbf{x}}F\_\mathcal{W}(\mathbf{x})\vert)\_i||}\Big)} / \sum\_{j=1}^d\exp\Big(||(\frac{1}{n}\sum\_{\mathbf{x}\in\mathcal{D}}\vert\nabla\_{\mathbf{x}}F\_\mathcal{W}(\mathbf{x})\vert)\_j|| \Big)$
>
> where sources with smaller variances obtain larger weights. This method is denoted as NRDE-CON. The performance of NRDE-CON and NRDE on several datasets is shown in the following table, where the results that NRDE significantly outperforms NRDE-CON support the assumption in our paper.
>
> **Table 9.** AUROC (%) performance of NRDE-CON and NRDE.
>
> | Method| WPBC | Thyroid | Musk | Annthyroid | Wilt |
> |--|-|-|-|--|-|
> | NRDE-CON   | 60.1 | 59.6| 76.5 | 53.3| 63.1 |
> | NRDE | 65.3 | 99.2| 99.8 | 98.4| 77.9 |
>
>
> **(ii)** Smoothness assumption in Eq. (11).
> As noted in the paper, this assumption is commonly used in linear ICA. Because the true generative process may be nonlinear, we adopt a more general smoothness condition. However, directly verifying this assumption on real datasets is inherently difficult, since the underlying generative process is not observable.
>
> **(iii)** Theorem 3.2 provides the error bound between our weighted log density and the density of pure data $\mathbf{x}\_{\text {pure }}$. Here, we compare the performance between NRDE, $p\_{\mathcal{X}}\left(\mathbf{x}\_{\text {pure }}\right)$ and $p\_{\bar{\mathcal{X}}}\left(\mathbf{x}\_{\text {pure }}\right)$, which are the estimated density of pure data $\mathbf{x}\_{\text {pure }}$. Since  estimating $p\_{\mathcal{X}}\left(\mathbf{x}\_{\text {pure }}\right)$ and $p\_{\bar{\mathcal{X}}}\left(\mathbf{x}\_{\text {pure }}\right)$ requires explicitly measuring $m$, which is the number of data sources, here we estimate $m$ using a simple strategy: we first compute the average absolute Jacobian
> $J = \frac{1}{n} \sum\_{\mathbf{x} \in \mathcal{D}}|\nabla F\_{\mathcal{W}}(\mathbf{x})|$.
> We then take the row norms $\\{{J}\_i\\}\_{i=1}^d$ and sort them in ascending order, denoted by $\\{{J}\_{i^*}\\}_{i=1}^d$.
>
> then we  measure the variance gap by computing $ \Delta\_i=(J\_{(i+1)^\ast}-J_{i^\ast}) / J\_{(i+1)^\ast}$, and find $m=\arg \max_{i}\Delta_i$.
>
>  As shown in the following table, the performance of using $p\_{\mathcal{X}}\left(\mathbf{x}\_{\text{pure}}\right)$ and $p\_{\bar{\mathcal{X}}}\left(\mathbf{x}\_{\text{pure}}\right)$ is very close to that of NRDE. A paired t-test on 10 datasets shows that the performance differences between NRDE and these two pure-data-based baselines are not statistically significant ($p > 0.05$)
>
>  **Table 10.** Performance comparison between NRDE, $p\_{\mathcal{X}}\left(\mathbf{x}\_{\text{pure}}\right)$ and $p\_{\bar{\mathcal{X}}}\left(\mathbf{x}\_{\text{pure}}\right)$.
>
> | Dataset| NRDE | $p_{\mathcal{X}}\left(\mathbf{x}_{\text{pure}}\right)$ | $p_{\bar{\mathcal{X}}}\left(\mathbf{x}_{\text{pure}}\right)$ |
> |--|-|-|-|
> | Satellite| 86.1 | 86.6| **88.0**|
> | WPBC| 65.3 | **71.4**     | 65.6|
> | Cardio| 86.1 | **87.2**     | 86.9|
> | Pima| **81.7** | 79.7     | 80.4 |
> | SpamBase | 87.4 | 85.5| **87.9** |
> | annthyroid   | **98.1** | 97.5     | 97.2 |
> | smtp| **95.6** | 95.2     | 95.5 |
> | glass | 91.2 | 91.5| **92.6**|
> | mammography  | 91.2 | **91.4**| 90.9|
> | vertebral    | 72.7 | 77.4 | **81.0**|
> | **AVG**      | 85.5 | 86.3| 86.6 |
> | **p-value**  | -    | 0.36| 0.24|
>
>
> **(iv)** The assumption in Theorem 3.1 considers situations where certain anomalies  $\hat{\mathbf{x}}$ receive the same density under the normalizing flow as normal samples $\mathbf{x}$. In such cases, if the flow outputs for these anomalies share the same norm as those of normal data, they can still be correctly identified by our proposed score $u^\ast(\hat{\mathbf{x}})$.
>
> **(v)** Regarding Assumption 3.3 and Theorem 3.4, these conditions characterize easy anomalies, namely anomalies that  receive lower anomaly scores than the maximum anomaly score among normal samples, and lie sufficiently far from the normal data manifold.
>
> for **(iv)** and **(v)**, we believe such conditions naturally arise in real datasets.

---

> > ### Author Response · Authors · 2025-11-25
> > **Response to Reviewer Xjrr(5/6)**
> >
> > **Response to W8 (an end-to-end latency profile)**
> >
> > We appreciate your suggestion regarding runtime comparison between NRDE and RealNVP. Because both methods share the same inference procedure, we report only their training times across datasets of varying dimensionality. The main time consumption of NRDE compared to RealNVP is the time for jacobian matrix computation for each training data. The results indicate that, even for large-scale or high-dimensional datasets, NRDE’s training time remains comparable to RealNVP with no substantial increase.
> >
> > **Table 11.** 100-epoch training time cost (seconds) comparison between NRDE and RealNVP on different dimensional datasets from ADBench.
> >
> > | Dataset          | NRDE  | RealNVP |
> > |------------------|:-----:|:-------:|
> > | annthyroid       | 4.12  | 2.03    |
> > | glass            | 1.58  | 0.59    |
> > | mammography      | 5.46  | 2.72    |
> > | Pima             | 1.68  | 0.61    |
> > | vertebral        | 1.45  | 0.59    |
> > | Cardiotocography | 4.29  | 0.90    |
> > | fraud            | 388.80| 75.21   |
> > | satellite        | 30.31 | 18.80   |
> > | satimage-2       | 31.40 | 19.04   |
> > | shuttle          | 46.26 | 30.05   |
> > | mnist            | 71.34 | 20.84   |
> > | musk             | 70.41 | 18.03   |

---

> ### Author Response · Authors · 2025-11-25
> **Response to Reviewer Xjrr(6/6)**
>
> **Questions**
>
> **Response to Q1**
>
> Please refer to our response to W1.
>
> **Response to Q2**
>
> Please refer to our response for W2.
>
> **Response to Q3**
>
> Please refer to our response for W6.
>
> **Response to Q4**
>
> Please refer to our response for W5. As for more experiments for the transductive study, we now conduct experiments on the other $5$ datasets. As shown in the following table, NRDE still outperforms these baselines in transductive setting, demonstrating its robustness to anomaly contamination in training set.
>
> **Table 8.** Detailed AUROC performance of outlier detection on 10 datasets. The best results are marked in **bold**.
>
> | AUROC   | Cardiotocography | SpamBase | Satellite | Pima | WPBC | glass | optdigits | PageBlocks | pendigits | Waveform | **Avg** |
> |---------|------------------|----------|-----------|------|------|-------|-----------|------------|-----------|----------|--------:|
> | IF      | 68.8 | 65.5 | 67.9 | 67.0 | 49.1 | 78.2 | 74.1 | 89.1 | **95.5** | 72.8 | **72.8** |
> | ECOD    | 78.5 | 65.5 | 58.2 | 59.4 | 48.1 | 70.4 | 60.4 | 91.3 | 92.7 | 60.3 | **68.4** |
> | OC-SVM  | 69.5 | 53.3 | 66.3 | 62.3 | 48.4 | 59.9 | 50.7 | **91.4** | 93.1 | 67.1 | **66.2** |
> | KPCA    | 53.4 | 52.1 | 48.2 | 53.8 | 45.5 | 49.9 | 52.2 | 64.3 | 57.2 | 56.0 | **53.2** |
> | LOF     | 52.3 | 45.6 | 54.1 | 60.1 | 52.0 | 77.0 | 53.7 | 71.5 | 49.9 | 70.5 | **58.6** |
> | KDE     | 50.2 | 49.5 | 76.0 | 72.2 | 49.9 | 82.0 | 32.2 | 90.6 | 89.0 | 75.1 | **66.6** |
> | $k$NN   | 57.9 | 52.9 | 65.0 | 65.1 | 47.2 | **86.7** | 37.2 | 88.8 | 75.8 | 73.4 | **65.0** |
> | RealNVP | 62.7 | 56.5 | 74.6 | 70.7 | 59.1 | 79.6 | 72.3 | 86.4 | 91.1 | 69.8 | **72.2** |
> | COPOD   | 66.2 | 68.7 | 63.3 | 65.4 | 52.3 | 75.5 | 68.2 | 87.5 | 90.4 | 73.3 | **71.0** |
> | DeepIF  | 63.0 | 37.9 | 74.3 | 61.3 | 49.4 | 84.5 | 56.3 | 87.5 | 95.3 | 78.6 | **68.8** |
> | **Ours**| **80.5** | **77.7** | **81.9** | **80.1** | **62.2** | 85.0 | **75.3** | 82.7 | 88.3 | **91.8** | **80.5** |
>
> **Response to Q5**
>
> We thank the reviewer for this insightful question but we may not fully capture the exact intent of the question. If the intent is to assess the independence between the components of the latent sources $s$, we clarify the following: according to the objective of the normalizing flow in Eq.~(1), the output
> $z = F_\mathcal{W}(x)$ is optimized to follow $\mathcal{N}(0, I)$. The latent sources $s$ are then defined as
> $s = \Sigma^{\frac{1}{2}} z$, where $\Sigma$ is a diagonal covariance matrix. Therefore, by construction, the components of $s$ are statistically independent.
>
> Verifying the variance assumption directly on real data is challenging because the underlying generative process $G$ or its inverse  $F_{\mathcal{W}}^*$ is not accessible during training. To partially address this,
> - we conducted experiments on real datasets under a contradictory assumption, and the corresponding results are reported in **Table 17 of Appendix G.6.3** verify our assumption.
> - Another practical way to indirectly assess how often our assumptions hold across the 47 datasets is to compare NRDE’s performance with that of other density-based methods, particularly with RealNVP (normalizing flow). The substantial performance gains of NRDE over RealNVP on most datasets suggest that the assumptions reasonably hold in the majority of cases. However, on several datasets such as landsat, optdigits, and letter, NRDE shows limited or no improvement—indicating that the assumptions may not hold for these datasets.
>
> **We sincerely thank the reviewer for taking the time to evaluate our paper and this rebuttal. We look forward to your further feedback, and please feel free to let us know if any concerns or questions remain unaddressed.**

---

> > ### Comment · Reviewer_Xjrr · 2025-11-28
> >
> > Thank you for your thorough answers. I will increase my score to reflect your thoughful rebuttal.

---

> ### Author Response · Authors · 2025-11-28
>
> We are very grateful for your feedback on our rebuttal and your recognition of our work.

---

### Official Review · Reviewer_SWAH · 2025-10-30

**Soundness:** 3
**Presentation:** 3
**Contribution:** 3
**Rating:** 4
**Confidence:** 4

**Summary:**

The authors introduce Noise-Robust Density Estimation (NRDE), an unsupervised anomaly-detection method for tabular data that estimates the density of clean samples while reducing the influence of noise. The method builds on a normalizing flow architecture (similar to RealNVP) and uses a Jacobian-based regularization that separates signal from noise in the latent space (in an ICA fashion). It is evaluated on 47 AD-BENCH datasets, showing that it outperforms multiple baselines. They also provide an ablation and theoretical analysis to support its ability to isolate the true data density. Finally, the method is demonstrated to be robust to noise.

**Strengths:**

The authors address an important well well-studied problem.

Overall, the paper is well written and mostly easy to follow.

I like the idea presented in the paper, and
The method demonstrates good results on a wide range of datasets.

The method is more robust to noise compared with baselines.

**Weaknesses:**

The main weakness of the paper is that some technical details are obscure and written in a vague way in the appendix. For example, the discussion about \lambda and the learning rate does not provide a complete picture on how these hyperparameters are selected.

For instance:
“Fixing λ = 0, decrease learning rate from 0.01 to 0.001 until training becomes stable (i.e., no loss explosion); (ii) Then, based on (15), viewed as minW L(λ, W), select λ ∈ 1, 0.1, 0.01 such that the regularization term λR(·) is on a comparable scale with 0.1 · L(0, W).”

-If lambda is first initialized as 0, how much effect does it have? I see an ablation and sensitivity analysis, but only for 5  datasets.

-What are the final values of all these parameters? How exactly were they selected? The explanation above does not provide a procedure based on which I can reproduce this selection (“comparable scale with”,” decrease the learning rate”, how?)
It would be valuable to show the dynamics of the training loss and performance across the training procedure.

*I would be happy to raise my score if these issues could be clarified in the rebuttal.

**Questions:**

Why were these five datasets selected for all the extended evaluations (contamination, ablation…)
For the sensitivity analysis, suddenly one more dataset is removed, why?

The number of baselines, as stated in the paper, varies (13, 14, 15). You actually compare to 15.
In 4.4 why is the noise added to normal test samples stronger?

Citation format doesn’t fit the writing. Namely:
Anomaly detection Chandola et al. (2009); Pang et al. (2021); Ruff et al. (2021)-> should be
Anomaly detection (Chandola et al. (2009); Pang et al. (2021); Ruff et al. (2021))
Only if the name is part of the sentence, it should be without brackets.

Commas are missing after some equations, for example, 15.

Typos:
rangeing- ranging

---

> ### Author Response · Authors · 2025-11-25
> **Response to Reviewer SWAH (1/3)**
>
> Dear Reviewer SWAH：
>
> We sincerely appreciate your recognition of our contributions and the effectiveness of our method. Thank you very much for your valuable suggestions, which will help us further improve the quality of our work. Additional experimental results conducted during the rebuttal phase are provided in **Appendix H of our paper** for the reviewers’ convenience. Please refer to it for details. Our responses are as follows.
>
> **Weaknesses**
>
> **Response to W1 (Clarity of hyperparameter tuning strategy)**
>
> Thank you for highlighting this source of confusion, which indeed impacts the reproducibility of our method. Now we provide a detailed algorithm for our hyperparameter tuning strategy. Please refer to **Algorithm 2 in Appendix D of our paper** for details.
>
> Once the hyperparameters are determined using this algorithm, it will not be changed during the training process. Meanwhile, the hyperparameter configuration for all $47$ datasets is also provided.
>
> **Table 1.** Hyperparameter configuration of NRDE on 47 real-world datasets in ADBench.
>
> | Data             | $\text{lr}$ | $\lambda$ | Data             | $\text{lr}$ | $\lambda$ |
> |------------------|------------:|----------:|------------------|------------:|----------:|
> | ALOI             | 0.005       | 1         | annthyroid       | 0.001       | 0.1       |
> | backdoor         | 0.001       | 0.1       | breastw          | 0.001       | 0.1       |
> | campaign         | 0.005       | 0.01      | cardio           | 0.005       | 0.1       |
> | Cardiotocography | 0.01        | 0.01      | celeba           | 0.001       | 0.01      |
> | census           | 0.005       | 0.01      | cover            | 0.01        | 1         |
> | donors           | 0.001       | 0.1       | fault            | 0.01        | 1         |
> | fraud            | 0.001       | 1         | glass            | 0.005       | 0.1       |
> | Hepatitis        | 0.01        | 0.1       | http             | 0.01        | 0.01      |
> | InternetAds      | 0.001       | 0.1       | Ionosphere       | 0.001       | 0.01      |
> | landsat          | 0.005       | 0.1       | letter           | 0.001       | 0.01      |
> | Lymphography     | 0.005       | 0.1       | magic.gamma      | 0.01        | 0.1       |
> | mammography      | 0.005       | 1         | mnist            | 0.001       | 1         |
> | musk             | 0.005       | 1         | optdigits        | 0.005       | 1         |
> | PageBlocks       | 0.01        | 0.01      | pendigits        | 0.01        | 0.1       |
> | Pima             | 0.001       | 0.1       | satellite        | 0.001       | 1         |
> | satimage-2       | 0.001       | 0.1       | shuttle          | 0.01        | 0.1       |
> | skin             | 0.001       | 0.1       | smtp             | 0.01        | 0.1       |
> | SpamBase         | 0.01        | 0.1       | speech           | 0.001       | 0.1       |
> | Stamps           | 0.01        | 0.1       | thyroid          | 0.005       | 0.01      |
> | vertebral        | 0.005       | 0.1       | vowels           | 0.005       | 1         |
> | Waveform         | 0.001       | 0.1       | WBC              | 0.01        | 0.1       |
> | WDBC             | 0.01        | 0.1       | Wilt             | 0.01        | 1         |
> | wine             | 0.005       | 1         | WPBC             | 0.005       | 0.1       |
> | yeast            | 0.01        | 1         |                  |             |           |

---

> > ### Author Response · Authors · 2025-11-25
> > **Response to Reviewer SWAH (2/3)**
> >
> > **Response to W2 (Effectiveness of $\lambda=0$ on more datasets)**
> >
> > Thanks for pointing out ablation study experiments on more datasets to verify the effectiveness of $\lambda=0$. Now we provide performance comparison of NRDE ($\lambda\neq0$)and NRDE ($\lambda=0$)  on $47$ datasets. Setting $\lambda\neq0$ results in performance improvement in most datasets.
> >
> > **Table 2.** AUROC performance comparison of NRDE on 47 real-world datasets in ADBench with or without $\mathcal{R}$ ($\lambda$ is set to 0).
> >
> > | Data             | $\lambda = 0$ | $\lambda \neq 0$ | Data             | $\lambda = 0$ | $\lambda \neq 0$ |
> > |------------------|-------------------|-----------------|------------------|-------------------|-----------------|
> > | ALOI             | 55.1              | 56.7            | annthyroid       | 98.1              | 98.4            |
> > | backdoor         | 86.7              | 94.5            | breastw          | 99.3              | 99.4            |
> > | campaign         | 76.4              | 76.9            | cardio           | 76.1              | 95.8            |
> > | Cardiotocography | 84.6              | 86.1            | celeba           | 84.3              | 87.9            |
> > | census           | 73.4              | 76.7            | cover            | 69.2              | 84.1            |
> > | donors           | 97.3              | 97.6            | fault            | 58.3              | 62.7            |
> > | fraud            | 92.4              | 95.9            | glass            | 89.0              | 91.2            |
> > | Hepatitis        | 82.3              | 85.3            | http             | 99.7              | 99.8            |
> > | InternetAds      | 76.2              | 79.1            | Ionosphere       | 87.6              | 87.7            |
> > | landsat          | 71.1              | 71.5            | letter           | 67.0              | 70.2            |
> > | Lymphography     | 98.6              | 98.7            | magic.gamma      | 80.6              | 81.9            |
> > | mammography      | 89.7              | 91.2            | mnist            | 91.8              | 93.2            |
> > | musk             | 99.9              | 99.8            | optdigits        | 94.2              | 94.5            |
> > | PageBlocks       | 92.1              | 92.1            | pendigits        | 96.2              | 98.1            |
> > | Pima             | 80.8              | 81.7            | satellite        | 84.0              | 86.8            |
> > | satimage-2       | 99.6              | 99.7            | shuttle          | 99.8              | 99.9            |
> > | skin             | 92.1              | 92.5            | smtp             | 95.7              | 95.6            |
> > | SpamBase         | 86.6              | 87.4            | speech           | 63.6              | 64.7            |
> > | Stamps           | 93.7              | 95.9            | thyroid          | 99.3              | 99.2            |
> > | vertebral        | 67.8              | 72.7            | vowels           | 83.3              | 87.8            |
> > | Waveform         | 91.7              | 91.6            | WBC              | 99.7              | 99.9            |
> > | WDBC             | 99.6              | 99.9            | Wilt             | 79.4              | 77.9            |
> > | wine             | 97.2              | 99.1            | WPBC             | 63.6              | 65.3            |
> > | yeast            | 59.2              | 61.1            |                  |                   |                 |
> >
> >
> > **Response to W3 (dynamics of the training loss and performance)**
> >
> > Thank you for suggesting this additional experiment to improve clarity. We now include the dynamics of both the training loss and the performance metrics over the course of training on several datasets in **Figure 10 from Appendix H of our paper**, where the decrease in loss is consistent with the improvement in performance.

---

> > > ### Author Response · Authors · 2025-11-25
> > > **Response to Reviewer SWAH (3/3)**
> > >
> > > **Questions**
> > >
> > > **Response to Q1:**
> > >
> > > We thank the reviewer for this helpful question. Since our main focus is on standard anomaly detection, we evaluate NRDE and 15 baseline methods on all 47 datasets to demonstrate the overall effectiveness of NRDE. For the extended evaluations—robustness to artificially injected noise, anomaly contamination in the training data, and hyperparameter sensitivity—we use 5 randomly selected datasets. These robustness studies are not the primary focus of the paper, and conducting all evaluations across all 47 datasets under multiple settings for every baseline would be computationally prohibitive.
> > >
> > > Outlier detection results on the other 5 datasets are already provided in the table below and further demonstrate the robustness of NRDE under anomaly contamination in the training data. We would be happy to provide additional results under these extended settings upon request.
> > >
> > > Regarding sensitivity analysis, we thank the reviewer for pointing out this oversight. In the revised version, we now include the complete sensitivity analysis results on all 5 datasets in Table 14 of our paper.
> > >
> > > **Table 3.** AUROC  of different methods for outlier detection on the other 5 datasets. The best results are marked in **bold**.
> > >
> > > | Method  | glass | optdigits | PageBlocks | pendigits | Waveform | Avg    |
> > > |---------|-------|-----------|------------|-----------|----------|--------|
> > > | IF      | 78.2  | 74.1      | 89.1       | **95.5**  | 72.8     | 81.94  |
> > > | ECOD    | 70.4  | 60.4      | 91.3       | 92.7      | 60.3     | 75.02  |
> > > | OC-SVM  | 59.9  | 50.7      | **91.4**   | 93.1      | 67.1     | **72.44** |
> > > | KPCA    | 49.9  | 52.2      | 64.3       | 57.2      | 56.0     | 55.92  |
> > > | LOF     | 77.0  | 53.7      | 71.5       | 49.9      | 70.5     | 64.52  |
> > > | KDE     | 82.0  | 32.2      | 90.6       | 89.0      | 75.1     | 73.78  |
> > > | kNN     | **86.7** | 37.2   | 88.8       | 75.8      | 73.4     | 72.38  |
> > > | RealNVP | 79.6  | 72.3      | 86.4       | 91.1      | 69.8     | 79.84  |
> > > | COPOD   | 75.5  | 68.2      | 87.5       | 90.4      | 73.3     | 78.98  |
> > > | DeepIF  | 84.5  | 56.3      | 87.5       | 95.3      | **78.6** | 80.44  |
> > > | **Ours**| 85.0 | **75.3** | 82.7     | 88.3      | **91.8** | **84.62** |
> > >
> > >
> > > **Response to Q2**
> > >
> > > Thanks for pointing out writing issue. We have corrected the number of baselines in our paper.
> > >
> > > Regrading noise injection, thank you for this insightful question. Adding stronger noise to normal test samples than to anomalous ones reduces the separability between the two classes, making the detection task more challenging than under equal-noise perturbations. Consequently, this protocol provides a stricter evaluation of robustness to artificially injected noise.
> > >
> > > For completeness, we also consider a symmetric design in which stronger noise is added to the anomalous test samples. We conduct experiments on the same five datasets, and the corresponding results are presented in the table below. These experiments further demonstrate the robustness of NRDE under different artificial-noise-injection protocols.
> > >
> > > **Table 4.** AUROC results (%) of the best-performing 5 methods on anomaly detection with noisy data. The best results per dataset is in **bold**.
> > >
> > > | Dataset           | DSVDD | KPCA  | IF    | kNN   | NRDE (ours) |
> > > |-------------------|:-----:|:-----:|:-----:|:-----:|:-----------:|
> > > | Cardiotocography  | **83.7** | 75.8 | 80.7 | 71.3 | 82.1        |
> > > | Pima              | 72.5 | 77.0 | 75.8 | 78.1 | **79.6** |
> > > | Satellite         | 81.5 | 84.1 | 79.6 | **86.9** | 85.1 |
> > > | SpamBase          | 80.3 | **86.3** | 82.4 | 83.0 | 79.1 |
> > > | WPBC              | 47.5 | 52.2 | 51.7 | 51.5 | **62.9** |
> > > | **AVG**           | 73.1 | 75.1 | 74.0 | 74.2 | **77.2** |
> > >
> > > **Response to Q3**
> > >
> > > Thanks for pointing out these writing issues. We have corrected them in our revised paper.
> > >
> > > **We thank the reviewer for reviewing our paper and this rebuttal. We are looking forward to your feedback and pleaselet us know if there are any concerns or questions still not properly addressed.**

---

### Author Response · Authors · 2025-12-01
**Brief Summary of Rebuttal (1/2)**

Dear Area Chair and Senior Area Chair,

We sincerely thank you for your time and effort in evaluating our submission. We fully understand that the technical issues encountered on the OpenReview system have considerably increased your workload, and we are very grateful for your patience and dedication.

To help streamline the discussion and support an efficient final assessment, we respectfully summarize the reviewers’ key concerns alongside our corresponding responses in the table below. We hope this concise overview offers clarity and convenience during the rebuttal process.

**Reviewer SWAH:**
| NO. | Reviewer Concerns | Our Responses|
|------------------|-----|-------|
|Weaknesses|||
| W1 |Clarity of hyperparameter tuning strategy| We have provided the detailed algorithm for our hyperparameter tuning strategy with specific hyperparameter configuration.  |
|W2|Effectiveness of $\lambda=0$ on more datasets|We have provided the detailed results on $47$ datasets.|
|W3|Dynamics of the training loss and performance|We have provided  the dynamics in Figure 10 of our paper.|
|Questions|||
|Q1|Only five datasets selected for all the extended evaluations|Our main focus is on standard anomaly detection and evaluation is conducted on 47 datasets. Extended evaluations on more datasets are provided.|
|Q2|In evaluation on noisy data, noise added to normal test samples is stronger| This protocol provides a stricter evaluation of robustness to artificially injected noise. We now provide results under a symmetric design where stronger noise is added to anomalous samples, and NRDE remains robust in this setting.|
|Q3|Several writing issues|We have corrected them in our revised paper.|

The reviewer explicitly noted: **“I would be happy to raise my score if these issues (weaknesses) could be clarified in the rebuttal.”** Since the main concerns are focused on technical details and clarity of presentation, and we have provided detailed responses including the hyperparameter-tuning algorithm, additional experiments, and training dynamics, we believe that Reviewer SWAH’s concerns have been substantively addressed.

**Reviewer Xjrr:**
| NO. | Reviewer Concerns | Our Responses|
|------------------|-----|-------|
|Weaknesses|||
|W1|A quantitative explicit diagnostic to verify variance difference should be provided.|We provided  a simple way to empirically verify the variance difference.|
|W2|Weighting can be sensitive to scaling and to the architecture of the flow.|We provided experiments to evaluate the sensitivity of our method to different architectural choices.|
|W3|A more systematic sensitivity analysis would be welcome|We summarized the sensitivity analysis included in our paper and during rebuttal phase.|
|W4|Several baselines appear under-tuned|We provided results where grid search over the hyperparameters is used on several recent methods to select their best-performing hyperparameter for each datasets.|
|W5|Statistical reporting is thin for some figures|We provided the box plots for each methods and p-values for each comparison with the baselines.|
|W6| Noisy data protocol may bias the comparison.|We provided results using a symmetric design.|
|W7|Theoretical conditions verification|We summarized the theoretical conditions and verifications for them.|
|W8|Running time comparison between NRDE and RealNVP|We provided time consumption of NRDE and RealNVP.|
|Questions|||
|Q1|Same as W1|Same as W1.|
|Q2|Same as W2|Same as W2.|
|Q3|Same as W6|Same as W6.|
|Q4| Expand the transductive study to more datasets|More experimental results for the transductive study are provided.|
|Q5|Include a small empirical audit of the modeling assumptions|We summarized the modeling assumptions and their verifications.|

**The reviewer appreciates our thorough answers and claimed to increase score to reflect our thoughtful rebuttal,** which indicates that the concerns from **Reviewer Xjrr** have been satisfactorily addressed.

---

> ### Author Response · Authors · 2025-12-01
> **Brief Summary of Rebuttal (2/2)**
>
> **Reviewer BvpA:**
> | NO. | Reviewer Concerns | Our Responses|
> |------------------|-----|-------|
> |Weaknesses|||
> |W1|Assumption in Eq.6 is not validated nor justified|We summarized all the verifications for this assumption in our paper and during rebuttal phase.|
> |W2|Writing issues in methodology part|We substantially revised the methodology section to address the specific problems (highlighted in orange).|
> |W3|Code reproducibility issue|We explain the reason for not using code of ADBench directly in detail and provide complete runnable code for all baseline models and our method.|
>
> We also find that several concerns from **Reviewer BvpA** are in conflict with the comments from the other two reviewers:
>
> | Concern from **Reviewer BvpA**                            | Comment from **Reviewer SWAH**                                                                                                    | Comment from **Reviewer Xjrr**                                                                                                                                                               |
> |---------------------------------------------------------|----------------------------------------------------------------------------------------------------------------------------------|---------------------------------------------------------------------------------------------------------------------------------------------------------------------------------------------|
> | “The writing in general is very poor.”                    | “Overall, the paper is well written and mostly easy to follow.”                                                                   | “I like the idea and the careful framing.”  “The proposed Jacobian-regularized flow with a weighted log density is conceptually simple, implementable with common flow libraries, and well-documented.” |
> | “After reading the rebuttal I did not find the work to be better motivated.” | “The authors address an important and well-studied problem.”  “I like the idea presented in the paper.” | “The method is clearly motivated and the pipeline from the generative view to the practical scoring rule is coherent.” “The paper addresses a real pain point of density-based anomaly detection on tabular data, namely false alarms caused by nuisance variation that is not semantically relevant for the task.” |
>
> After reading response from **Reviewer BvpA**, we find that:
> - **W1 and W2 appear to be resolved**, as the reviewer does not reiterate these concerns in their response.
> - The new concern about **motivation** (raised in the reviewer’s response) seems to conflict with the assessments of the other two reviewers, both of whom explicitly state that the problem is important and the method is clearly and carefully motivated. In our response, we provided additional explanations and empirical evidence to further clarify the motivation.
> - Regarding **W3 (code reproducibility)**, the reviewer does not specify which aspects remain unclear or problematic. In our rebuttal, we carefully explained why we did not directly use the original ADBench code and provided complete, runnable code for all baseline methods and our approach to ensure full reproducibility.
>
> **We greatly value the time, effort, and expertise you have dedicated to reviewing our work. Please don't hesitate to contact us if you have any questions or require further clarification.**
>
> Sincerely,
>
> The Authors

---

### Meta-Review · Area_Chair_fCgH · 2026-01-08

**Summary:**

Reviewers SWAH and Xjrr focused on technical clarity, specifically regarding hyperparameter tuning strategies, statistical significance, and the robustness of the variance separation assumption across diverse datasets. Xjrr further questioned the fairness of baseline comparisons and the end-to-end latency of the proposed Jacobian regularizer, while BvpA raised significant concerns about the fundamental motivation and writing quality. BvpA remained highly skeptical of the core logic in Eq. 6, arguing that the definition of noise versus signal is arbitrary and that the experimental results lacked sufficient reproducibility.

The initial version of the paper lacked a lot of technical clariy and lacked sufficient experimentation. The authors did a great job in the rebuttal and included some clarifications and quite a few experiments. But, I still feel two issues remain - the fundamental assumption of Eq 6 is strong and can be domain-specific, and authors need to provide more justification / analysis (some of which was included in the rebuttal). Second, the experiments and analysis section should be improved. Because of these concerns, I am inclined to reject the paper.

**Reviewer Concerns:**

The authors successfully addressed hyperparameter study (SWAH), baseline tuning (Xjrr), and statistical reporting (Xjrr) by providing new algorithms, grid search results, and p-value tables in the updated appendix. They also mitigated reproducibility concerns by releasing the full code and providing a diagnostic for variance separation, which largely satisfied the technical doubts of the two marginal reviewers. However, the conceptual validity of the noise-signal separation remains an outstanding point of contention for BvpA, who feels the motivation for the weighting scheme is still poorly justified.

**Reviewer Scores:**

Reviewer Xjrr explicitly stated they would increase their score to a 6, noting that the authors provided a thorough rebuttal and systematic sensitivity analysis. Reviewer SWAH would retain the score - the authors addes some experiments that the reviewer requested, but the technical clarity can still be improved. Reviewer BvpA explicitly indicated that they retain their score at 2, as they found the revised motivation and experimental validation insufficient to resolve their concerns.

---

### Decision · Program_Chairs · 2026-01-26

Reject